# FraPPE: Fast and Efficient Preference-based Pure Exploration

**Udvas Das**
Univ. Lille, Inria
CNRS, Centrale Lille
UMR 9189 - CRIStAL, Lille, France
udvas.das@inria.fr

**Apurv Shukla**
Department of EECS, University of Michigan
Ann Arbor, MI, USA
apurv.shukla@umich.edu

**Debabrota Basu**
Univ. Lille, Inria
CNRS, Centrale Lille
UMR 9189 - CRIStAL, Lille, France
debabrota.basu@inria.fr

## Abstract

Preference-based Pure Exploration (PrePEx) aims to identify with a given confidence level the set of Pareto optimal arms in a vector-valued (aka multi-objective) bandit, where the reward vectors are ordered via a (given) preference cone $\mathcal{C}$. Though PrePEx and its variants are well-studied, there does not exist a *computationally efficient* algorithm that can *optimally* track the existing lower bound (Shukla and Basu, 2024) for arbitrary preference cones. We successfully fill this gap by efficiently solving the minimisation and maximisation problems in the lower bound. First, we derive three structural properties of the lower bound that yield a computationally tractable reduction of the minimisation problem. Then, we deploy a Frank-Wolfe optimiser to accelerate the maximisation problem in the lower bound. Together, these techniques solve the maxmin optimisation problem in $\mathcal{O}(KL^2)$ time for a bandit instance with $K$ arms and $L$ dimensional reward, which is a significant acceleration over the literature. We further prove that our proposed PrePEx algorithm, FraPPE, asymptotically achieves the optimal sample complexity. Finally, we perform numerical experiments across synthetic and real datasets demonstrating that FraPPE achieves the lowest sample complexities to identify the exact Pareto set among the existing algorithms.

## 1 Introduction

Randomised experiments are at the core of statistically sound evaluation and selection of public policies (Banerjee et al., 2020), clinical trials (Altman and Dore, 1990), material discovery (Raccuglia et al., 2016), and advertising strategies (Kohavi and Longbotham, 2015). They allocate a set of participants to different available choices, observe corresponding outcomes, and choose the optimal one with statistical significance. As these static and randomised experimental designs demand high number of samples, it has invoked a rich line of research in adaptive experiment design (Hu and Rosenberger, 2006; Foster et al., 2021), also known as active (sequential) testing (Yu et al., 2006; Naghshvar and Javidi, 2013; Kossen et al., 2021) or *pure exploration* problem (Even-Dar et al., 2006; Bubeck et al., 2009; Carlsson et al., 2024). Pure exploration problems sequentially allocate participants to different available choices while looking into past allocations and outcomes. The aim is to leverage the previous information gain and structure of the experiments and observations to identify the optimal choice with as few number of interactions as possible. Even in this age of data,

39th Conference on Neural Information Processing Systems (NeurIPS 2025).

pure exploration draws significant attention in diverse settings, such as material design (Gopakumar et al., 2018), clinical trials (Villar et al., 2018), medical treatments (Murphy, 2005), where each observation involves a human participant or is costly due to the involved experimental infrastructure.

In this paper, we focus on multi-armed bandit formulation of pure exploration (Thompson, 1933; Lattimore and Szepesvári, 2020), which is a theoretically-sound and popular framework for sequential decision-making under uncertainty, and the archetypal setup of Reinforcement Learning (RL) (Sutton et al., 1998). In bandits, a learner encounters an instance of $K$ decisions (or arms). At each time step $t$, a learner chooses one of these $K$-arms $A_t$, and obtains a noisy feedback $R_t$ (or reward or outcome) from the reward distribution corresponding to that arm, i.e. $\nu_{A_t}$. Note that each reward distribution has a fixed mean $\boldsymbol{\mu}_a$, which is unknown to the learner. The goal of the learner is to adaptively select the arms in order to identify the arm with the highest mean reward with a statistical confidence level and the least number of interactions. This is popularly known as the fixed-confidence Best Arm Identification (BAI) problem (Jamieson and Nowak, 2014; Garivier and Kaufmann, 2016; Soare et al., 2014; Degenne et al., 2020; Wang et al., 2021), which is a special case of pure exploration.

**PrePEx: Motivation.** The classical BAI and pure exploration literature, like most of the traditional RL literature, focuses on scalar reward, i.e. a single or scalarised objective. But this does not reflect the reality as often decisions have multiple and often conflicting outcomes (Gopakumar et al., 2018; Wei et al., 2023), and we might need to consider all of them before selecting the 'optimal' one. For example, in clinical trial, the goal is not only to choose the most effective dosage of a drug but also to ensure it is below a certain toxicity level (Réda et al., 2020). Another example is a phase II vaccine clinical trial, known as COV-BOOST (Munro et al., 2021), that measures the immunogenicity indicators (e.g. cellular response, anti-spike IgG and NT50) of different Covid-19 vaccines while applied as a booster (third dose). It is hard for a computer scientist to design a scalarised reward out of these indicators (known as the reward design problem), and different experts might have different preferences over them. Even in high-data regimes, proper reward modelling has emerged as a hard problem in Reinforcement Learning under Human Feedback (RLHF) literature (Scheid et al., 2024).

These evidences motivates us to study a multi-objective (aka vector-valued) bandit problem, where the reward feedback at every step is an $L$-dimensional vector corresponding to $L$-objectives of the learner. Additionally, the learner has access to a set of incomplete preferences over these objectives that together form a preference cone $\mathcal{C}$ (Jahn et al., 2009; Löhne, 2011). For example, polyhedral cones, positive orthant cone ($\mathbb{R}_+^L$), and customised asymmetric cones are used in aircraft design optimisation (Mavrotas, 2009), portfolio optimization balancing return, risk, and liquidity (Ehrgott, 2005), and climate policy optimization with multiple national goals (Keeney and Raiffa, 1993), respectively. In these cases, there might not exist an optimal arm but a *set of Pareto optimal arms* $\mathcal{P}^* \subseteq \{1, \dots, K\}$ (Drugan and Nowe, 2013; Auer et al., 2016).

Given the preference cone $\mathcal{C}$ and sampling access to the bandit instance, *the learner aims to exactly identify the whole Pareto optimal set of arms with a confidence level at least $1 - \delta \in [0, 1)$ while hoping to use as less samples as possible.* This problem is known as the *Preference-based Pure Exploration* (**PrePEx**) (Shukla and Basu, 2024), or Pareto set identification (Auer et al., 2016) or vector optimisation with bandit feedback (Ararat and Tekin, 2023).

**PrePEx: Related Works.** There are mainly three types of algorithms proposed for PrePEx: *successive arm elimination*, *lower bound tracking*, and *posterior sampling-based*. Successive arm elimination algorithms use allocations to collect samples from arms and eliminate them one by one when enough evidence regarding their suboptimality is gathered (Auer et al., 2016; Ararat and Tekin, 2023; Korkmaz et al., 2023; Karagözlü et al., 2024). But all of them can identify only an approximation of the Pareto optimal set. The same limitation applies for confidence bound based algorithms (Kone et al., 2023a). In contrast, the lower bound tracking algorithms first derive a lower bound on the number of samples required to solve PrePEx as a max-min optimisation problem. Then following the Track-and-Stop framework (Kaufmann et al., 2016), at every step, these algorithms plug in the empirical estimates of means of the objectives (obtained via previous samples) in the lower bound optimisation problem and obtains a candidate allocation policy. The allocation leads to a choice of the arm at every step till the algorithm is confident enough to identify the correct set of arms. But this optimisation problem is challenging, and only recently, Shukla and Basu (2024) proposed an explicit lower bound for any arbitrary preference cone. Since their lower bound contains optimisation over a non-convex set, they construct a convex hull of the non-convex variables and use it to propose a Track-and-Stop algorithm, PreTS. But it is computationally infeasible to run PreTS for

| Methods | Computational Complexity[1] | Preference Cone | Beyond Gaussian |
|---|---|---|---|
| Crepon et al. (2024) | $\mathcal{O}\left(K^{L+1}L^3\right)$ | Positive orthant | ✗ |
| Kone et al. (2024) | $\mathcal{O}\left(K^2L\min\{K,L\}\right)$ | Positive orthant | ✗ |
| Shukla and Basu (2024) | Intractable | Arbitrary cone | ✓ |
| FraPPE (This paper) | $\mathcal{O}\left(KL\min\{K,L\}\right)$ | Arbitrary cone | ✓ |

Table 1: Comparison of asymptotically optimal algorithms for solving PrePEx.

the benchmarks in the literature. Crepon et al. (2024) focuses only on the right-orthant as the cone and propose a specific optimization method by adding and removing points from the Pareto set. Still, it is computationally inefficient ($\mathcal{O}(K^L)$ runtime) to run it on existing benchmarks (Kone et al., 2024). The complexity of this optimisation problem has motivated Kone et al. (2024) to avoid it and propose a posterior sampling-based algorithm. Further discussion on related works is in Appendix A.1.

**Contributions.** We affirmatively address (Table 1) an extension of the open problem (Crepon et al., 2024): *Can we design a computationally efficient (polynomial in both $K$ and $L$) and statistically optimal PrePEx algorithms beyond Gaussian rewards and independent objectives, and for arbitrary preference cones?*

(1) **Tractable Optimisation:** We leverage the structure of the PrePEx problem to reduce the intractable $\sup - \inf - \inf - \inf$ optimisation problem in the existing lower bound to a tractable $\max - \min - \min - \min$ problem. This shows the redundancy of the convex hull approach of PreTS and solve the inner minimisation problems in $\mathcal{O}(KL\min\{K,L\})$ time. In practice, $K \gg L$, and thus, this is a significant improvement over the existing optimisation-based (Crepon et al., 2024) algorithms exhibiting $\mathcal{O}(K^L)$ computational complexity. (2) **Asymptotically Optimal and Efficient Algorithm:** We further leverage the Frank-Wolfe algorithm (Wang et al., 2021) to solve the outer maximisation problem for exponential family distributions and a relaxed stopping criterion to propose FraPPE. We prove a non-asymptotic sample complexity upper bound for FraPPE and shows it to be asymptotically optimal as $\delta \to 0$. (3) **Empirical Performance Gain:** We conduct numerical experiments across synthetic datasets with varying correlations between objectives and a real-life dataset (COV-BOOST). The results show that FraPPE enjoys the lowest empirical stopping time ($\sim$5-6X lower) and the lowest empirical error uniformly over time among all the baselines.

In brief, *we propose the first computationally efficient and asymptotically optimal algorithm, FraPPE, that works for arbitrary preference cones and exponential family distributions while achieving the lowest sample complexity and probability of error for identifying the exact set of Pareto optimal arms.*

## 2 Problem Formulation: Preference-based Pure Exploration (PrePEx)

First, we formally state the preference based pure exploration problem under the fixed-confidence setting and elaborate the relevant notations.

**Notations.** For any $n \in \mathbb{N}$, $[n]$ denotes the set $\{1, 2, \ldots, n\}$. $\mathbf{z}_\ell$ refers to the $\ell^{th}$ component of a vector $\mathbf{z}$. We use $\|\cdot\|_p$ to denote the $\ell_p$-norm of a vector. $\mathrm{vect}(A)$ is the vectorized version of matrix $A$. $\Delta_K$ represents the simplex on $[K]$ and $D_{\mathrm{KL}}(P \parallel Q)$ refers to the KL-divergence between two absolutely continuous distributions $P$ and $Q$. $\mathrm{ch}\{X\}$ means convex hull of a set $X$.

**Problem Formulation.** In PrePEx, we deal with a multi-objective bandit problem. A bandit environment consists of $K$ arms and each arm yields $L$-dimensional reward corresponding to the $L$ objectives. Specifically, each arm $a \in [K]$ has a reward distribution $\nu_a$ over $\mathbb{R}^L$ with unknown mean $\boldsymbol{\mu}_a \in \mathbb{R}^L$. Thus, a bandit environment is represented by the vectors of mean rewards $\{\boldsymbol{\mu}_i\}_{i=1}^K$, or alternatively, a matrix $M \in \mathbb{R}^{L \times K}$ such that its $a^{\mathrm{th}}$ column is $\boldsymbol{\mu}_a$.

At each time $t \in \mathbb{N}$, the learner pulls an arm $A_t \in [K]$ and observes an $L$-dimensional reward vector $\boldsymbol{R}_t$ sampled from $\nu_{A_t}$. In the simplest setting of pure exploration, i.e. Best Arm Identification (BAI), we have $L = 1$ and the learner focuses on finding the best arm, i.e. the arm with highest mean (Garivier and Kaufmann, 2016). In more general settings with $L = 1$, the learner aims to find a policy $\pi \in \Delta_K$ indicating the proportion to choose the arms to maximize the expected reward obtained from the environment (Carlsson et al., 2024).

---

[1]We ommit the complexity of calculating Pareto set, i.e. $\mathcal{O}(K(\log K)^{\max\{1, L-2\}})$, as it is same for every method (Kung et al., 1975).

*Pareto Optimality and Preference Cones.* Since we have mean vectors ($L > 1$), we need a set of preferences over the objectives to compare the means rewards of arms or policies. Thus, following the vector optimization literature (Jahn et al., 2009; Löhne, 2011; Ararat and Tekin, 2023), we assume that the learner has access to an ordering cone $\mathcal{C}$.

**Definition 1** (Ordering Cone). *A set $\mathcal{C} \subseteq \mathbb{R}^L$ is a **cone** if $\mathbf{v} \in \mathcal{C}$ implies that $\alpha\mathbf{v} \in \mathcal{C}$ for all $\alpha \geq 0$. A **solid cone** has a non-empty interior, i.e. $\mathrm{int}(\mathcal{C}) \neq \emptyset$. A **pointed cone** contains the origin. A closed convex cone that is both pointed and solid is called an **ordering cone** (aka a proper cone).*

Following PrePEx literature (Ararat and Tekin, 2023; Karagözlü et al., 2024; Shukla and Basu, 2024), we focus on the **polyhedral ordering cone** that induces a set of partial orders on vectors in $\mathbb{R}^L$.

**Definition 2** (Polyhedral Ordering Cone). *A cone $\mathcal{C}$ is a **polyhedral ordering cone** if $\mathcal{C} \triangleq \{x \in \mathbb{R}^L \,|\, W\mathbf{x} \geq 0\}$, where $W \in \mathbb{R}^{K \times L}$ with row-transposes $W_i^\top$ representing rays spanning the cone.*

$W$ is called the half-space representation of $\mathcal{C}$. An example of polyhedral cone is $\mathcal{C}_{\pi/4} \triangleq \{(r\cos\theta, r\sin\theta) \in \mathbb{R}^2 \mid r \geq 0 \wedge \theta \in [0, \pi/4]\}$, i.e., all the 2-dimensional vectors that makes an angle less than $\pi/4$ with the $x$-axis. The commonly used cone in the Pareto set identification literature (Kone et al., 2023a,b; Crepon et al., 2024) is the positive orthant $\mathbb{R}_+^L$, i.e. $\mathcal{C}_{\pi/2}$.

To avoid any redundancy (Ararat and Tekin, 2023; Shukla and Basu, 2024), we assume that $W$ is full row-rank and normalized, i.e. $\|W_i\|_2 = 1$. Hereafter, we call them *preference cones*, and the vectors in the cone as the *preferences*. For simplicity, in this paper, we consider that the preferences are normalised, i.e. $\mathbf{z} \in \mathcal{C} \cap \mathbb{B}(1) \triangleq \bar{\mathcal{C}}$. We now define the partial orders w.r.t. a preference cone $\bar{\mathcal{C}}$.

**Definition 3** (Partial Order). *For every $\boldsymbol{\mu}, \boldsymbol{\mu}' \in \mathbb{R}^L, \boldsymbol{\mu} \preceq_{\bar{\mathcal{C}}} \boldsymbol{\mu}'$ if $\boldsymbol{\mu} \in \boldsymbol{\mu}' + \bar{\mathcal{C}}$ and $\boldsymbol{\mu} \prec_{\bar{\mathcal{C}}} \boldsymbol{\mu}'$ if $\boldsymbol{\mu} \in \boldsymbol{\mu}' + \mathrm{int}(\bar{\mathcal{C}})$. Alternatively, $\boldsymbol{\mu} \preceq_{\bar{\mathcal{C}}} \boldsymbol{\mu}'$ is equivalent to $\mathbf{z}^\top(\boldsymbol{\mu} - \boldsymbol{\mu}') \leq 0, \forall \mathbf{z} \in \bar{\mathcal{C}}^+$. Here, $\bar{\mathcal{C}}^+$ is the dual cone of $\bar{\mathcal{C}}$.*

The partial order induced by $\bar{\mathcal{C}}$ induces further order over the set of arms $[K]$. Specifically, given any two arms $i, j \in [K]$: (i) arm $j$ *weakly dominates* arm $i$ iff $\boldsymbol{\mu}_i \preceq_{\bar{\mathcal{C}}} \boldsymbol{\mu}_j$, (ii) arm $j$ *dominates* arm $i$ iff $\boldsymbol{\mu}_i \prec_{\bar{\mathcal{C}} \setminus \{0\}} \boldsymbol{\mu}_j$, (iii) arm $j$ *strongly dominates* arm $i$ iff $\boldsymbol{\mu}_i \prec_{\bar{\mathcal{C}}} \boldsymbol{\mu}_j$.

**Definition 4** (Pareto Optimal Set and Policies). *An arm $i \in [K]$ is **Pareto optimal** if it is not dominated by any other arm w.r.t. the cone $\bar{\mathcal{C}}$. The **Pareto optimal set** $\mathcal{P}^*$ is the set of all Pareto optimal arms. The set of **Pareto optimal policies** (also known as **Pareto front**) $\Pi^P \subset \Delta_K$ is the set of non-dominated distributions having support on subsets of Pareto optimal arms.*

In Figure 1, we show the Pareto optimal arms for the SNW dataset with $K = 256$ and $L = 2$ (Zuluaga et al., 2012a). For the preference cone $\mathcal{C}_{\pi/2}$, the blue points represent the mean rewards of the Pareto optimal arms, whereas for $\mathcal{C}_{2\pi/3}$, the Pareto optimal arms correspond to green points (the blue and red lines show respective Pareto fronts). This shows how preference cones affect Pareto optimality.

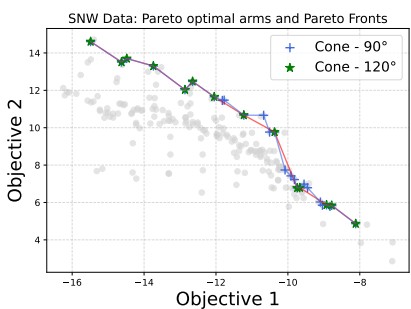

Figure 1: Effect of preference cones on Pareto optimal arms in SNW dataset.

Generally, PrePEx aims to identify all the distributions over the arms, i.e. the policies $\boldsymbol{\pi} \in \Delta_K$, lying on the *Pareto optimal policy set*. Given a preference cone $\bar{\mathcal{C}}$ and the bandit instance $M$, a learner can solve a vector optimization problem to exactly identify a policy on the Pareto front, i.e. a policy whose mean reward is non-dominated by any other policy w.r.t. $\bar{\mathcal{C}}$. Mathematically, a policy $\boldsymbol{\pi}^\star$ belongs to the Pareto front $\Pi^P$ if

$$\boldsymbol{\pi}^\star \in \arg\max_{\boldsymbol{\pi} \in \Delta_K} M\boldsymbol{\pi} \text{ with respect to } \bar{\mathcal{C}}. \quad (1)$$

In PrePEx, we address the problem in Equation (1), when the mean matrix $M$ is unknown *a priori* but bounded, i.e. $\|M\|_{\infty,\infty} \in [-M_{\max}, M_{\max}]$. We denote all such instances by $\mathcal{M}$. Our goal is to exactly identify with as few samples as possible, all the policies lying on the Pareto front using the $L$-dimensional reward feedback from the arms pulled.

**Definition 5** ($(1-\delta)$-correct PrePEx). *An algorithm for Preference-based Pure Exploration (PrePEx) is said to be $(1-\delta)$ correct if with probability at least $1 - \delta$, it returns the Pareto front $\Pi^P$.*

**Lower Bound.** Shukla and Basu (2024) quantify the minimum number of samples that an $(1 - \delta)$-correct PrePEx algorithm requires to identify the Pareto front as an optimisation problem.

**Theorem 1** (Lower Bound (Shukla and Basu, 2024))**.** *Given a bandit model $M \in \mathcal{M}$, a preference cone $\bar{\mathcal{C}}$, and a confidence level $\delta \in (0, 1)$, the expected stopping time of any $(1 - \delta)$-correct PrePEx algorithm, to identify the Pareto optimal policy set is $\mathbb{E}[\tau_\delta] \geq \mathcal{T}_{M, \bar{\mathcal{C}}} \log \left( \frac{1}{2.4\delta} \right)$, where the expectation is taken over the stochasticity of both the algorithm and the bandit instance. Here, $\mathcal{T}_{M, \bar{\mathcal{C}}}$ is called the* ***characteristic time*** *and is expressed as*

$$\left( \mathcal{T}_{M, \bar{\mathcal{C}}} \right)^{-1} \triangleq \sup_{\boldsymbol{\omega} \in \Delta_K} \inf_{\boldsymbol{\pi} \in \Delta_K \setminus \{\boldsymbol{\pi}^\star\}, \boldsymbol{\pi}^\star \in \Pi^P(M, \bar{\mathcal{C}})} \inf_{\tilde{M} \in \partial \Lambda(M)} \inf_{\mathbf{z} \in \bar{\mathcal{C}}} \sum_{k=1}^{K} \boldsymbol{\omega}_k D_{\mathrm{KL}} \left( \mathbf{z}^\top M_k \, \big\| \, \mathbf{z}^\top \tilde{M}_k \right). \quad (2)$$

*Here*, $\partial\Lambda(M) \triangleq \bigcup_{\substack{\boldsymbol{\pi} \in \Delta_K \setminus \{\boldsymbol{\pi}^\star\} \\ \boldsymbol{\pi}^\star \in \Pi^P(M, \bar{\mathcal{C}})}} \left\{ \tilde{M} \in \mathcal{M} \setminus \{M\} : \exists \, \mathbf{z} \in \bar{\mathcal{C}}^+, \, \langle \mathrm{vect} \left( \mathbf{z}(\boldsymbol{\pi} - \boldsymbol{\pi}^\star)^\top \right), \mathrm{vect}(\tilde{M}) \rangle = 0 \right\}.$

The lower bound tests how hard it is to distinguish a given instance $M$ and another instance $\tilde{M} \in \mathcal{M} \setminus \{M\}$ in the direction of a preference $\mathbf{z} \in \bar{\mathcal{C}}^+$ that makes them the most indistinguishable. $\tilde{M}$ is called an alternative instance and the set of alternative instances $\Lambda(M)$ is called the Alt-set. Specifically, Alt-set consists of all such instances in $\mathcal{M}$ which has at least one Pareto optimal policy different than that of $M$. We aim to find out the allocation $\boldsymbol{\omega}$ that allows us to maximise their KL-divergences and render them as distinguishable as possible.

**From Lower Bound to Optimal Algorithms.** A common approach to design asymptotically optimal pure exploration algorithms (e.g. Track-and-Stop (Kaufmann et al., 2016)) is to solve the $\sup - \inf$ optimisation problem (Eq. (2)) at every step with empirical estimates of the means $M$ obtained through bandit interaction, and stop only when one is confident about correctness. In this context, *we derive three structural observations regarding the optimisation problem and Alt-set that allows us to design the first efficient and asymptotically optimal PrePEx algorithm for generic preference cones.*

## 3 Structural Reduction of the Optimisation Problem

We observe that the optimisation problem in the lower bound (Equation (2)) is a combination of three $\inf$ and one $\sup$ problems. It is easy to observe that since $\Delta_K$ and $\bar{\mathcal{C}}^+$ are convex and compact sets. Thus, the optimisation problem reduces to

$$\max_{\boldsymbol{\omega} \in \Delta_K} \inf_{\substack{\boldsymbol{\pi} \in \Delta_K \setminus \{\boldsymbol{\pi}^\star\} \\ \boldsymbol{\pi}^\star \in \Pi^P(M, \bar{\mathcal{C}})}} \underbrace{\inf_{\tilde{M} \in \partial \Lambda(M)} \min_{\mathbf{z} \in \bar{\mathcal{C}}^+} \sum_{k=1}^{K} \boldsymbol{\omega}_k D_{\mathrm{KL}} \left( \mathbf{z}^\top M_k \, \big\| \, \mathbf{z}^\top \tilde{M}_k \right)}_{\triangleq f(\boldsymbol{\omega}, \boldsymbol{\pi}^\star, \boldsymbol{\pi}|M)}. \quad (3)$$

The outer maximisation problem with respect to $\boldsymbol{\omega}$, called **allocation**, is a linear programming (LP) problem solvable by any off-the-shelf LP-solver (e.g. CPLEX (Manual, 1987), HIGHS (Huangfu and Hall, 2018)), if all the inner $\inf$ problems can be reduced to $\min$ problems. The inner minimisation problem with respect to the preference $\mathbf{z}$ can be solved using an off-the-shelf conic programming solver (e.g. CVXOPT (Vandenberghe, 2010), CLARABEL (Towsley et al., 2022)). In the following sections, we further reduce the two $\inf$ problems into $\min$ problems.

### 3.1 Structure of the Pareto Optimal Policy Set

We first observe that the $\inf$ problem over the set of Pareto optimal policies $\Pi^P$ and the complementary set of any other policies $\Delta_K \setminus \{\boldsymbol{\pi}_i^\star\}$ is costly as the sets are continuous and possibly non-convex. It leaves two possibilities: (a) optimising over the convex hull of $\Pi^P$ or (b) reducing the optimisation problem to a tractable smaller set. Constructing the convex hull is computationally expensive and might lead to a looser minimum. Thus, we aim to reduce the $\inf$ problem to a well-behaved subset of $\Pi^P$. First, we observe that $\Pi^P$ is a compact set consisting of stationary policies from $\Delta_K$. Secondly, we prove that $\Pi^P$ is spanned by $p$ pure policies corresponding to the $p$ Pareto optimal arms.

**Theorem 2** (Basis of $\Pi^P$)**.** *Pareto optimal policy set $\Pi^P$ is spanned by $p$ pure policies corresponding to $p$ Pareto optimal arms, i.e. $\{\boldsymbol{\pi}_i^\star\}_{i=1}^p$. Here, $\boldsymbol{\pi}_i^\star$ is the pure policy with support on only arm $i$.*

Detailed proof is in Appendix B.1. Thus, $\Pi^P$ has finite number of extreme points. Given $\Pi^P$ is compact and $f(\boldsymbol{\omega}, \boldsymbol{\pi}^\star, \boldsymbol{\pi}|M)$ is continuous in $\boldsymbol{\pi}^\star$, $\inf_{\boldsymbol{\pi}^\star \in \Pi^P}$ is equal to $\min_{\boldsymbol{\pi}^\star \in \{\boldsymbol{\pi}_i^\star\}_{i=1}^p}$ (Fact 7). Now, we define the notion of neighbouring pure policies.

**Definition 6** (Neighbouring Pure Policies)**.** $\pi_j \in \{\pi_i\}_{i=1}^K$ *is a neighbouring pure policy of* $\pi^\star \in \{\pi_i^\star\}_{i=1}^p$ *if any mixed policy (distribution over actions) with support* $\mathrm{Supp}(\pi) = \{i,j\}$ *is not dominated by any other policy. Formally,* $\mathrm{nbd}(\pi_i^\star) \triangleq \{\pi_j \in \{\pi_k\}_{k=1}^K \setminus \{\pi_i^\star\} : \forall \pi \in \Delta_K, \pi_{ij}$ *with* $\mathrm{Supp}(\pi_{ij}) = \{i,j\}, M^\top \pi_{ij} \npreceq_{\bar{\mathcal{C}}} M^\top \pi\}$.

By Carathéodory's theorem (Leonard and Lewis, 2015), any Pareto optimal pure policy can have up to $\min\{K,L\}$ such neighbouring pure policies. For example, neighbouring pure policies of any $+$ Pareto optimal arm in Figure 1 are the $+$ arms connected by the Pareto front (blue line). We also observe that since SNW dataset[2] has *two objectives*, *each pure Pareto optimal policy (or arm) has two neighbouring pure policies (or arms)*. With this structure, we obtain that $\inf_{\pi \in \Pi^P \setminus \{\pi_i^\star\}}$ is equal to $\min_{\pi \in \mathrm{nbd}(\pi_i^\star)}$ as $f(\omega, \pi^\star, \pi|M)$ is continuous in $\pi$ and the minimum is achieved at one of the extreme points of the polyhedra. Thus, the optimisation problem in Equation (3) reduces to

$$\max_{\omega \in \Delta_K} \min_{\substack{\pi_j \in \mathrm{nbd}(\pi_i^\star) \\ \pi_i^\star \in \{\pi_i^\star\}_{i=1}^p}} \underbrace{\inf_{\tilde{M} \in \bar{\partial}\Lambda(M)} \min_{\mathbf{z} \in \bar{\mathcal{C}}^+} \sum_{k=1}^K \omega_k D_{\mathrm{KL}}\left(\mathbf{z}^\top M_k \,\Big\|\, \mathbf{z}^\top \tilde{M}_k\right)}_{\triangleq f_{ij}(\omega|M)}, \tag{4}$$

while $\bar{\partial}\Lambda(M) \triangleq \bigcup_{\substack{\pi_j \in \mathrm{nbd}(\pi_i^\star) \\ \pi_i^\star \in \{\pi_i^\star\}_{i=1}^p}} \left\{ \tilde{M} \in \mathcal{M} \setminus \{M\} : \exists \mathbf{z} \in \bar{\mathcal{C}}^+, \, \langle \mathrm{vect}\left(\mathbf{z}(\pi_j - \pi_i^\star)^\top\right), \mathrm{vect}(\tilde{M}) \rangle = 0 \right\}$ is a subset of the Alt-set $\partial\Lambda(M)$ considered in Equation (2). Thus, *these observations together yield a significant reduction in the complexity of the* $\inf$ *problem over the set of Pareto optimal policies and their neighbours to* $\mathcal{O}(K \min\{K,L\})$ *evaluation problems of* $f_{ij}(\omega|M)$.

**Remark 1** (Connection to Construction of Alt-set by Crepon et al. (2024))**.** *Crepon et al. (2024) optimise the* $f(\omega, \pi^\star, \pi, M)$ *by constructing a function from the Pareto set to* $[L]$ *that computes the minimum cost of adding and removing any Pareto optimal point of* $M$. *This is a combinatorial mapping with* $\mathcal{O}(K^L)$ *realisations. We accelerate it significantly by leveraging the structure of* $\Pi^P$ *and evaluate* $\mathcal{O}(K \min\{K,L\})$ *functions of Pareto optimal points and their neighbours.*

## 3.2   Structure of the Alternative Set

Now, the only demanding optimisation problem left is finding the infimum over the Alternative instance $\tilde{M}$ in the reduced Alt-set $\bar{\partial}\Lambda(M)$. We observe that for each Pareto optimal pure policy $\pi_i^\star$ and its neighbouring pure policy $\pi_j$, $\Lambda_{ij}(M) \triangleq \left\{ \tilde{M} \in \mathcal{M} \setminus \{M\} : \exists \mathbf{z} \in \bar{\mathcal{C}}^+, \, \langle \mathrm{vect}\left(\mathbf{z}(\pi_j - \pi_i^\star)^\top\right), \mathrm{vect}(\tilde{M}) \rangle = 0 \right\}$ is a convex set. Hence, the reduced Alt-set $\bar{\partial}\Lambda(M)$ is a union of $\mathcal{O}(K \min\{K,L\})$ convex sets $\{\Lambda_{ij}(M)\}_{i,j}$.

This observation removes the need of constructing a convex hull around the original Alt-set defined by Shukla and Basu (2024), and searching for the infimum in it. This procedure is prohibitively expensive. But reducing the Alt-set to a union of convex sets completely eliminates this step and we can reduce the optimisation problem (Fact 8) further:

$$\max_{\omega \in \Delta_K} \min_{\substack{\pi_j \in \mathrm{nbd}(\pi_i^\star) \\ \pi_i^\star \in \{\pi_i^\star\}_{i=1}^p}} \min_{\tilde{M} \in \bar{\Lambda}_{ij}(M)} \min_{\mathbf{z} \in \bar{\mathcal{C}}} \sum_{k=1}^K \omega_k D_{\mathrm{KL}}\left(\mathbf{z}^\top M_k \,\Big\|\, \mathbf{z}^\top \tilde{M}_k\right), \tag{5}$$

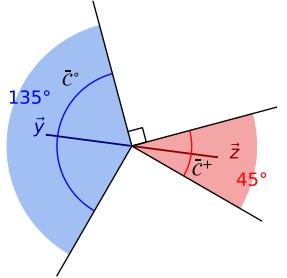

Since both the innermost minimisation problems are over convex sets, the minimisation problem in the lower bound can be solved using $\mathcal{O}(K \min\{K,L\})$ calls to a convex optimisation oracle. For well-behaved distributions, e.g. multi-variate Gaussians, we have closed form solutions and can avoid the minimisation over the Alternative instances $\tilde{M}$ (Appendix B.3).

**Remark 2** (Connection to Pure Exploration with Multiple Answers)**.** *The idea of constructing Alt-set as a union of convex sets was first introduced by Degenne and Koolen (2019) for pure exploration with multiple correct answers and further elaborated in (Wang et al., 2021).*

Figure 2: Preference & polar cones.

---

[2]SNW dataset ($K = 206, L = 2$) is derived from the domain of computational hardware design, specifically concerning the optimization of sorting network configurations (Zuluaga et al., 2012b).

### 3.3 Structure of the Alternative Instance and Polar Cone

Though we have a tractable max-min optimization problem at hand, the elegant structure of the preference cones allow us to reduce the computational complexity further.

**Proposition 1** (Polar Cone Representation of Alternative Instances). *For all $M \in \mathcal{M}$ and given $\pi_i^\star \in \Pi^P$ and $\pi_j \in \mathrm{nbd}(\pi_i^\star)$, the set of Alternative instances takes the explicit form $\Lambda_{ij}(M) = \{\tilde{M} \in \mathcal{M} \setminus \{M\} : \exists \mathbf{y} \in \mathrm{bd}(\bar{\mathcal{C}}^\circ) \text{ such that } \tilde{M}\pi_j = \tilde{M}\pi_i^\star + \mathbf{y}\}$ where $\mathbf{y}$ are defined with the polar cone of $\bar{\mathcal{C}}^+$, i.e. $\bar{\mathcal{C}}^\circ \triangleq \{\mathbf{y} \in \mathbb{R}^L \text{ s.t. } \langle \mathbf{z}, \mathbf{y} \rangle \leq 0, \forall \mathbf{z} \in \bar{\mathcal{C}}^+\}$.*

Proposition 1 shows that though an Alternative instance is a $K \times L$-dimensional matrix, it can be represented by an $L$-dimensional vector $\mathbf{y}$ at the boundary of the polar cone. We illustrate the geometry of this transformation in Figure 2. If an off-the-shelf convex optimiser is used to minimise over Alternative instances and the optimiser is not dimension-free (e.g. mirror descent (Beck and Teboulle, 2003), Vaidya's algorithm (Vaidya, 1996)), then this transformation reduces the computational expense significantly (For most of the real data, $K \gg L$). For multivariate Gaussians with covariance matrix $\Sigma$, we derive the closed form of the polar vector corresponding to the Alternative instance (ref. Lemma 4). Please refer to Appendix B.2 for detailed proof.

## 4 Algorithm Design: Frugal and Fast PrePEx with Frank-Wolfe

Enabled by the structural reductions in Section 3, we are now ready to describe the FraPPE algorithm, which is the first computationally efficient, optimisation-driven algorithm for exact detection of the Pareto optimal set. We begin by stating the required assumption to build FraPPE.

**Assumption 1** ($L$-Parameter Exponential Family). *Let $X = (X_1, , X_L)$ be a $L$-dimensional random vector with a distribution $P_\theta, \theta \in \Theta$ and mean $\boldsymbol{\mu} \in \mathcal{M} \subset \mathbb{R}^L$. Suppose $X_1, \ldots, X_L$ are jointly continuous. Then, the family of distributions $\{P_\theta, \theta \in \Theta\}$ belongs to the $L$ parameter exponential family if its density of $X$ can be represented as $f(X|\theta) \triangleq h(X) \exp(\eta(\theta)T(X) - \psi(\theta))$. $\eta : \Theta \to \mathbb{R}^s$ is the natural parametrization for some $s \geq L$. $T : \mathbb{R}^L \to \mathbb{R}^s$ is the sufficient statistic for the natural parameter. $h(\boldsymbol{X})$ is the base density such that $h : \mathbb{R}^L \to [0, \infty)$. Finally, $\psi(\boldsymbol{\theta}) = \log\left(\int_{\mathcal{X}} h(\boldsymbol{X}) \exp(\langle \eta(\boldsymbol{\theta}), T(\boldsymbol{X}) \rangle d\boldsymbol{x})\right)$ is the log-normaliser or log-partition function. Additionally, we assume that the exponential family is non-singular, i.e. $\nabla_\theta^2 \psi(\theta) \geq \beta$ for some $\beta > 0$ and for all $\theta$.*

Note that the natural parameter can have dimension $s$ greater than $L$ but in our case, only the $L$-dimensional mean is unknown. To highlight this, we call the above an $L$-parameter exponential family rather than $s$-parameter. This is a common assumption in BAI for $L = 1$ (Kaufmann et al., 2016; Garivier and Kaufmann, 2016; Degenne and Koolen, 2019). Rather, the Pareto front identification literature has been limited to Gaussians and extending to exponential families has been an open question (Crepon et al., 2024). The non-singular curvature is also a mild assumption for keeping the problem well-defined. For example, this holds true for any Gaussian with non-singular covariance matrix (See Example 1) and Bernoullis with probability of success not equal to zero or one.

### 4.1 Fast Optimisation of Allocations with Frank-Wolfe

The outer optimisation problem in Equation (5) is Linear Program (LP) over a polyhedra with respect to $\boldsymbol{\omega}$. Any PrePEx algorithm solves this LP trying to estimate allocation per step. Unlike typical LP solvers, projection-free methods, like Frank-Wolfe (FW) (Jaggi, 2013), solve this *smooth convex* program more efficiently (Chandrasekaran et al., 2012). The necessary conditions for FW to converge towards the optimal allocation $\boldsymbol{\omega}^\star(M)$ are **1.** the LP under study must be smooth, **2.** the gradient and curvature of the function to be maximised should not blow up at the boundary of the polyhedra.

**1. Smoothness:** For a fixed $\pi_i^\star \in \{\pi_i^\star\}_{i=1}^p$, the function $f_{ij}(\boldsymbol{\omega} \mid M)$ is smooth only at the minima $\pi_j \in \arg\min_{\pi_j \in \mathrm{nbd}(\pi_i^\star)} f_{ij}(\boldsymbol{\omega} \mid M)$. As suggested by Wang et al. (2021), we can adapt FW to cope with non-smooth objective function by constructing $r$-sub-differential set close to the non-smooth points. We define the $r$-sub-differential set as

$$H_M(\boldsymbol{\omega}, r) \triangleq \mathrm{ch}\left\{\nabla_{\boldsymbol{\omega}} f_{ij}(\boldsymbol{\omega}|M) : f_{ij}(\boldsymbol{\omega}|M) < \min_{\pi_i^\star, \pi_j} f_{ij}(\boldsymbol{\omega}|M) + r, \forall \pi_i^\star \in \Pi^P, \pi_j \in \mathrm{nbd}(\pi_i^\star)\right\}. \quad (6)$$

**Algorithm 1 FraPPE- Fr**ugal and F**ast P**reference-Based **P**ure **E**xploration

1: **Input:** Confidence level $\delta$ and sequence $\{r_t\}_{t \geq 1} = t^{-0.9}/K$
2: **Initialise:** For $t \in [K]$, sample each arm once s.t. $\boldsymbol{\omega}_K = (1/K, \cdots, 1/K)$, mean estimate $\hat{M}_K$
3: **while** Equation (8) is **FALSE do**
4:      **if** $\sqrt{t/K} \in \mathbb{N}$ or $\hat{M}_t \notin \mathcal{M}$ **then**
5:          **Forced Exploration:** $\boldsymbol{\omega}_t \leftarrow (1/K, \cdots, 1/K)$
6:      **else**
7:          **Estimate Pareto Indices:** Calculate Pareto indices $\mathcal{P}_t$ based on current estimate $\hat{M}_t$.
8:          **Estimate Set of Pareto Policies:** $\Pi^{\mathcal{P}_t}$ consisting pure policies with $i_t \in \mathcal{P}_t$ as basis.
9:          **Set of Neighbours:** $\Pi \setminus \Pi^{\mathcal{P}_t}$, where $\Pi$ is the set of all pure policies.
10:         **Construct Sub-Differential Set:** $H_{\hat{M}_t}(\boldsymbol{\omega}_t, r_t)$ using Equation (6)
11:         **FW-Update:** $\mathbf{x}_{t+1} \leftarrow \arg\max_{\boldsymbol{\omega} \in \Delta_{K\gamma}} \min_{h \in H_{\hat{M}_t}(\boldsymbol{\omega}_t, r_t)} \langle \mathbf{x} - \boldsymbol{\omega}(t), \mathbf{h} \rangle, \boldsymbol{\omega}_{t+1} \leftarrow \frac{1}{t+1}\mathbf{x}_{t+1} + \frac{t}{t+1}\boldsymbol{\omega}_t$
12:      **end if**
13:     **C-tracking:** Play $A_t \in \arg\min N_{a,t} - \sum_{s=1}^{t+1} \boldsymbol{\omega}_s$ (ties broken arbitrarily)
14:      **Feedback and Parameter Update:** Get feedback $R_t \in \mathbb{R}^L$ and update $\hat{M}_t$ to $\hat{M}_{t+1}$ with $R_t$
15: **end while**
16: **Recommendation Rule:** Recommend $\mathcal{P}_t$ as the Pareto optimal set

---

As computing gradient in the neighbourhood of $\boldsymbol{\omega}$ is expensive, FW further simplifies the outer maximisation and calculates the allocation in two simple steps by linearising as follows

$$\mathbf{x}_{t+1} \triangleq \arg\max_{\mathbf{x} \in \Delta_K} \min_{\mathbf{h} \in H_M(\boldsymbol{\omega}, r)} \langle \mathbf{x} - \boldsymbol{\omega}_t, \mathbf{h} \rangle, \quad \boldsymbol{\omega}_{t+1} \triangleq \frac{t}{t+1}\boldsymbol{\omega}_t + \frac{1}{t+1}\mathbf{x}_{t+1}. \quad (7)$$

We further prove (Appendix D) that $\boldsymbol{\omega} \longmapsto H_M(\boldsymbol{\omega}, r)$ is continuous and continuously differentiable.

**2. Gradient and Curvature.** For FW to converge, boundedness of the gradient and curvature constant is necessary. Lemma 1 ensures that the FW converges to the optimal allocation (ref. Appendix D).

**Lemma 1.** *If Assumption 1 holds true, then for all $M \in \mathcal{M}$: 1.* ***Bounded gradients:*** $\|\nabla_{\boldsymbol{\omega}} f_{ij}(\boldsymbol{\omega}|M)\|_{\infty} \leq D$ *for all $\boldsymbol{\pi}_i^{\star}, \boldsymbol{\pi}_j$, and $\boldsymbol{\omega} \in \Delta_K$. 2.* ***Bounded curvature:*** $C_{f_{ij}(\cdot|M)}(\Delta_{K\gamma}) \leq 8D\alpha^{-1}$ *for all $i, j$, $\gamma \in (0, 1/K)$, and some $\alpha > 0$. Here, $C_f(A)$ is curvature constant of concave differentiable function $f$ in set $A$ (Definition 8) and $\Delta_{K\gamma} \triangleq \{\boldsymbol{\omega} \in \Delta_K : \min_k \boldsymbol{\omega}_k \geq \gamma\}$.*

Lemma 1 allows us to further accelerate the linearised optimisation in Equation (7) in $\mathcal{O}\left(\frac{1}{\text{tol}}\right)$ instead of standard FW complexity $\mathcal{O}\left(\frac{K}{\text{tol}}\right)$ (Jaggi, 2013), where tol is the tolerable error margin for the optimisation. Thus, we propose the *first PrePEX algorithm for general exponential family* whereas existing literature is restricted to Gaussian or Bernoulli (Crepon et al., 2024).

## 4.2   FraPPE: Frugal and Fast PrePEx

We propose FraPPE for *efficient (Frugal and Fast)* identification of *all* the Pareto optimal arms in PrePEx. FraPPE follows the three component-based design from pure exploration literature (Kaufmann et al., 2016; Degenne and Koolen, 2019; Wang et al., 2021).

**Component 1.** The first component of FraPPE is a hypothesis testing scheme based on a *sample statistic* (Line 3 in Algorithm 1) that decides whether the algorithm should stop sampling and recommend the estimated Pareto optimal set as the optimal one. This is called the "**Stopping Rule**". We revisit the stopping rule described by Shukla and Basu (2024): $\min_{\tilde{M} \in \text{ch}(\partial\Lambda(\hat{M}_t))} \min_{\mathbf{z} \in \bar{\mathcal{C}}^+} \sum_{k=1}^{K} N_{k,t} D_{\text{KL}}\left(\mathbf{z}^{\top}\hat{M}_{k,t} \,\Big\|\, \mathbf{z}^{\top}\tilde{M}_k\right) \geq c(t, \delta)$. We note that constructing convex hull around the Alt-set per iteration is *computationally expensive and not really tractable*. Instead, we take advantage of Equation (5) to deploy a *tractable and efficient* stopping rule:

$$\min_{\boldsymbol{\pi}_{i_t}^{\star} \in \Pi^{\mathcal{P}_t}} \min_{\boldsymbol{\pi}_j \in \text{nbd}\left(\boldsymbol{\pi}_{i_t}^{\star}\right)} \inf_{\tilde{M} \in \Lambda_{ij}(\hat{M}_t)} \min_{\mathbf{z} \in \bar{\mathcal{C}}^+} \sum_{k=1}^{K} N_{k,t} D_{\text{KL}}\left(\mathbf{z}^{\top}\hat{M}_{k,t} \,\Big\|\, \mathbf{z}^{\top}\tilde{M}_k\right) \geq c(t, \delta) \quad (8)$$

where $c(t, \delta) \triangleq \sum_{k=1}^{K} 3\ln\left(1 + \ln\left(N_{k,t}\right)\right) + K\mathcal{G}\left(\frac{\ln\left(\frac{1}{\delta}\right)}{K}\right)$. For explicit expression of $\mathcal{G}(\cdot) : \mathbb{R}^+ \to \mathbb{R}^+$, refer to Theorem 13 (Kaufmann and Koolen, 2021). The intuition behind this stopping rule

stems from the *Sticky Track-and-Stop strategy* for multiple correct answers setting (Degenne and Koolen, 2019) that stops as soon as it can identify any **one** of the correct answers (Pareto arms in our case). Though in our setting, we need to rule out the possibility of choosing a confusing instance for all the correct answers, i.e. the Pareto arms. Hence, we take minimum over the finite set of Pareto optimal policies $\{\pi_i^\star\}_{i=1}^p$. Note that, (8) is the first Chernoff-type stopping rule that encapsulates the effect of $\mathcal{C}$ that has been unresolved in the literature (Crepon et al., 2024).

**Component 2.** The next component of FraPPE is a "**Sampling Rule**". It chooses the action to play based on the allocation $\omega_t$ estimated via Equation (7) (cf. Section 4.1). We use "C-tracking" (Line 13, Algorithm 1) as the other variant "D-tracking" fails to converge to $\omega^\star(M)$ for multiple correct answers (Degenne and Koolen, 2019). We refer to Appendix H for convergence and other results.

**Component 3:** Once the stopping rule is fired i.e., flag is **TRUE**, FraPPE recommends the estimated Pareto arms as the set of correct answers. The stopping rule ensures that the Pareto arms given by "**Recommendation rule**" are correct with probability at least $(1 - \delta)$ (Theorem 13).

**Sample Complexity.** Now, we show that FraPPE *is an asymptotically optimal PrePEx algorithm*.

**Lemma 2** (Sample Complexity Upper Bound). *For any $M \in \mathcal{M}$, $\delta \in (0, 1)$, and preference cone $\bar{\mathcal{C}}$, expected stopping time satisfies* $\limsup_{\delta \to 0} \frac{\mathbb{E}[\tau_\delta]}{\log(\frac{1}{\delta})} \leq \mathcal{T}_{M,\bar{\mathcal{C}}}$.

Thus, FraPPE *achieves asymptotic optimality*. We also prove *correctness* (Theorem 6) and derive a *non-asymptotic sample complexity bound* (Lemma 6 in Appendix E.4), which we omit for brevity.

**Computational Complexity.** First, Line 7 suffers worst-case complexity for estimating Pareto set using the algorithm of Kung et al. (1975) is $\mathcal{O}\left(K \log(K)^{\max\{1, L-2\}}\right)$ (Kone et al., 2024). Then, for each $\{\pi_i^\star\}_{i=1}^p$ and $\{\pi_j\}_{j=1}^{|\text{nbd}(\pi_i^\star)|}$, Component 1 and Frank-Wolfe step (Line 11) enjoys time complexity $\mathcal{O}(L)$ due to $K$-independent bound over curvature and gradients (Lemma 1 (Jaggi, 2013)). Thus, FraPPE has the total time complexity $\mathcal{O}\left(\max\left\{K(\log K)^{\max\{1, L-2\}}, KL \min\{K, L\}\right\}\right)$. Note that, for $L \geq 5$ and $K \geq 19$, runtime of FraPPE is $\mathcal{O}\left(K(\log K)^{\max\{1, L-2\}}\right)$, i.e. the Pareto set computation becomes the dominant component. We refer to Appendix G.1 for detailed discussion.

## 5 Experimental Analysis

We perform empirical evaluation of FraPPE on a real-life dataset as well as synthetic environment.

**Benchmark algorithms.** We compare our algorithm with **PSIPS** (Posterior concentration based Bayesian algorithm (Kone et al., 2024)), **APE** (Approximate Pareto set identification (Kone et al., 2023a)), **Oracle** that pulls arms according to $\omega^\star(M)$, i.e., the optimal allocation, **Uniform** sampler, and also **TnS** (Gradient based algorithm in (Crepon et al., 2024)). We consider $c(t, \delta) = \ln(\frac{1+\ln(t)}{\delta})$.

**Experiment 1: Cov-Boost Trial Dataset.** This real-lide inspired data set contains tabulated entries of phase-2 booster trial for Covid-19 (Munro et al., 2021). Cov-Boost has been used as a benchmark dataset for evaluating algorithms for Pareto Set Identification (PSI). It consists bandit instance with 20 vaccines, i.e. arms, and 3 immune responses as objectives, i.e., $K = 20$ and $L = 3$.

**Observation.** (a) **Lower and Stable Sample Complexity.** In Figure 3, we plot the stopping times for $\delta = 0.01$ that validates the frugality of FraPPE in terms of median sample complexity. Additionally, we observe very less variability compared to PSIPS which leverages posterior sampling, which makes FraPPE *a more stable strategy*. For $\delta = 0.1$, Kone et al. (2024) states that PSIPS has an average sample complexity of 20456, while TnS of Crepon et al. (2024) reports it to be 17909. FraPPE exhibits an average sample complexity of 3523 ($\sim$5-6X less) over 100 independent experiments.

(b) **Low Error probability.** In Figure 5, we visualise the evolution of average error $\mathbb{1}(\mathcal{P}_t \neq \mathcal{P}^\star)$ for FraPPE against PSIPS and Uniform explorer. FraPPE reduces the error rate significantly faster.

**Experiment 2: Effect of Correlated Objectives.** We also test FraPPE on the Gaussian instance of Kone et al. (2024) used with 5 arms, 2 objectives, and the covariance matrix with unit variances. Correlation coefficients are varied from $-1$ to $1$ with grid size 0.1. We fix $\delta = 0.01$. We compare the empirical performance with PSIPS as it is the only algorithm tackling correlated objectives.

**Observation: Uniformly Better Performance across Correlations.** We plot the average sample complexity (averaged over 1000 runs for each correlation coefficient) in Figure 4. It shows that

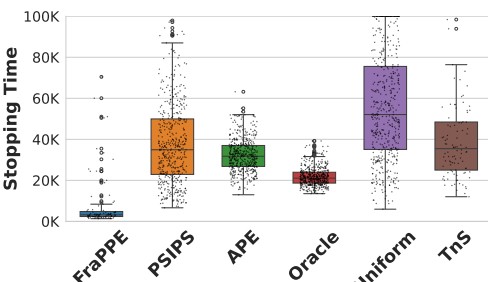

Figure 3: Stopping times for Cov-Boost Trial.

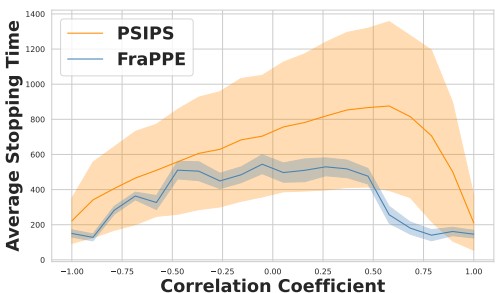

Figure 4: Effect of correlated objectives on Gaussian instance.

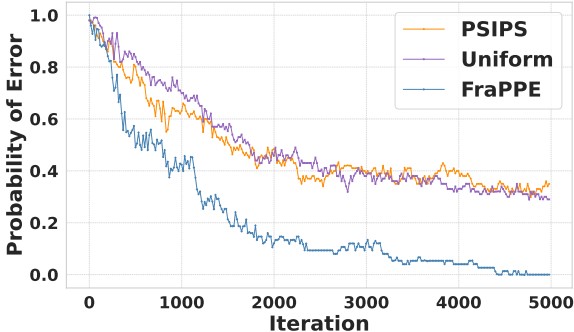

Figure 5: Error probability evolution for Cov-Boost Trial.

FraPPE achieves better sample frugality than PSIPS across all the values of correlation coefficients. Notably, the standard deviation of FraPPE is also narrower indicating its stability across instances.

**Summary.** Thus, based on the results from Experiments 1 and 2 and computational complexity guaranty, we conclude that FraPPE *is the PrePEx algorithm with the lowest empirical stopping time (5X lower), better true positive rate, and lower computational complexity among the optimisation-based baselines*. Further results on runtime analysis of FraPPE are provided in Appendix J.

## 6 Discussions and Future Works

We study the problem of preference based pure exploration with fixed confidence (PrePEx) that aims to identify all the Pareto optimal arms (and policies) for a multi-objective (aka vector-valued) bandit problem with an arbitrary preference cone. We study the existing lower bound for this problem and through three structural observations regarding the Pareto optimal policies, alternating instances, and the Alt-set, we reduce it to a tractable optimisation problem. We further apply Frank-Wolfe based optimisation method and a relaxed stopping rule to propose FraPPE. FraPPE *is the first PrePEx algorithm that is asymptotically optimal, can handle generic exponential family distributions*, and thus, resolving most of the open questions in (Crepon et al., 2024). Experiments show that FraPPE achieves around 5X less sample complexity to identify the exact set of Pareto optimal arms across instances.

Throughout this work, we have assumed to know the exact cone $\mathcal{C}$. Thus, learning the cone simultaneously while solving PrePEx is an interesting future direction of research. Another future work is to scale FraPPE to practical applications of PrePEx, e.g. aligning large language models with RL under Human Feedback (RLHF) (Ji et al., 2023). It would be also interesting to extend our algorithm design from independent arms to structured bandits (e.g., linear, contextual).

## Acknowledgment

DB and UD acknowledge ANR JCJC project REPUBLIC (ANR-22-CE23-0003-01), PEPR project FOUNDRY (ANR23-PEIA-0003), and Inria-ISI Kolkata associate team SeRAI. AS would like to acknowledge the suport from ONR award no. N000142412742 for funding part of this work.

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

# Appendix

## Table of Contents

# A    Notations and Extended Related Works

| Notation | Description |
|---|---|
| $\mathcal{C}, \prec_{\bar{\mathcal{C}}}$ | Given convex cone and induced partial order |
| $\bar{\mathcal{C}}$ | $\triangleq \mathcal{C} \cap \mathbb{B}(1)$ where $\mathbb{B}(1)$ is an unit ball |
| $\bar{\mathcal{C}}^+$ | Dual cone of $\bar{\mathcal{C}}$ |
| $\bar{\mathcal{C}}^\circ$ | Polar cone of a cone $\bar{\mathcal{C}}^+$ |
| $K, L$ | Number of arms and objectives |
| $\mathcal{P}^*, \hat{\mathcal{P}}_t$ | Ground truth Pareto set and estimated Pareto set |
| $M \in \mathbb{R}^{K \times L}$ | matrix with mean reward of $K$ arms |
| $\boldsymbol{\omega}$ | Allocation vector |
| $\Pi^{\mathrm{P}}$ | Family of Pareto optimal policies |
| $\hat{\mu}_{k,t}^{(\ell)}, \mu_k^{(\ell)}$ | Estimated and true of mean rewards |
| $\Lambda_{ij}(M)$ | Set of alternating instances of $M$ for fixed $\boldsymbol{\pi}_i^\star \in \Pi^{\mathrm{P}}$ and $\boldsymbol{\pi}_j \mathrm{nbd}\,(\boldsymbol{\pi}_i^\star)$ |
| $\mathrm{int}(X)$ | Interior of a set $X$ |
| $\mathrm{ch}\,\{X\}$ | Convex Hull of a set $X$ |
| $S_i$ | $S_i \triangleq \left\{ M \in \mathcal{M}, \exists \mathbf{z} \in \mathcal{C} : \mathbf{z}^\top M \boldsymbol{\pi}_i^\star \geq \mathbf{z}^\top M \boldsymbol{\pi}, \forall \boldsymbol{\pi} \in \mathrm{nbd}\,(\boldsymbol{\pi}_i^\star) \right\}$ |
| $\delta$ | Confidence parameter |
| $c(t, \delta)$ | Stopping threshold |
| $c_1(\mathcal{M})$ and $c_2(\mathcal{M})$ | Constants such that $\forall t > c_1(\mathcal{M}), c(t, \delta) \leq \log\left( \frac{c_2(\mathcal{M})t}{\delta} \right)$ |
| $H_M(\boldsymbol{\omega}, r)$ | $r$-subdifferential set (Equation (6)) |
| $\tilde{M}$ | Confusing instance w.r.t $M$ |
| tol | Error tolerance in linear optimisation |
| $N_{a,t}$ | Number of pulls of arm $a$ at time $t \in \mathbb{N}$ |
| $\mathbf{z}$ | Preference vector from the cone $\bar{\mathcal{C}}$ |
| $\boldsymbol{\pi}_i^\star$ | Pareto optimal policy |
| $\boldsymbol{\pi}_j$ | Neighbour of the Pareto optimal policy $\boldsymbol{\pi}_i^\star$ |
| $\Delta_K$ | $K$-dimensional simplex |
| $\Delta_{K\gamma}$ | $\{\boldsymbol{\omega} \in \Delta_K : \min_k \boldsymbol{\omega}_k \geq \gamma\}$ |
| $W$ | Preference Matrix |
| $\tau_\delta$ | $(1 - \delta)$-correct Stopping time |
| $\mathbf{y}$ | Vector from polar cone $\bar{\mathcal{C}}^\circ$ |
| $D$ | Upper bound on infinite norm of gradient $\nabla_{\boldsymbol{\omega}} f_{ij}(\boldsymbol{\omega} \mid M)$ |
| $\mathcal{M}$ | Parameter space of mean matrix |
| $p$ | Number of Pareto optimal arms, i.e cardinality of the set $\mathcal{P}^\star$ |
| $e_k$ | $K$-dimensional vector with 1 at $k$-th index, and 0 elsewhere |

## A.1  Extended Related Works

Pure exploration in multi-armed bandits (MAB) has been extensively studied, particularly in the context of best-arm identification (BAI) under fixed-confidence (Kaufmann et al., 2016; Garivier and Kaufmann, 2016) and fixed-budget (Even-Dar et al., 2006) settings. In this paper we restrict ourselves to fixed-confidence setting where the goal is to recommend the arm with highest mean reward with probability at least $(1 - \delta)$ for a given confidence parameter $\delta \in (0, 1)$. Specially, in the last decade we have witnessed emergence of several algorithmic strategies to tackle the problem of BAI in fixed-confidence setting. To name a few, these include action elimination based approach in (Even-Dar et al., 2006; Kone et al., 2023a), LUCB strategies introduced in (Audibert and Bubeck, 2010), further in (Jamieson et al., 2014), tracking a information theoretic lower bound (Track-and-Stop) (Kaufmann et al., 2016), extended by gamification of the lower bound (Degenne et al., 2020) or acceleration by leveraging projection-free convergence towards optimal allocation (Wang et al., 2021).

**Pure exploration with vector feedback.** Traditionally, pure exploration in BAI focuses on scalarised rewards (Carlsson et al., 2024). In contrast, we consider the problem of preference-based pure exploration (PrePEx), where the agent receives a vector reward upon playing an arm of dimension equal to the number of objectives ($L$) under study partially ordered by a preference cone. Recently, Shukla and Basu (2024) tackle this problem by deriving a novel information-theoretic lower bound on expected sample complexity that captures the influence of the preference cone's geometry. They proposed the Preference-based Track-and-Stop (PreTS) algorithm, which leverages a convex relaxation of the lower bound and demonstrates asymptotic optimality through new concentration inequalities for vector-valued rewards. Kone et al. (2024), on the other hand leverages concentration on priors over reward vectors by posterior sampling to recommend the exact Pareto optimal arms. Other notable works include Auer et al. (2016), who explored Pareto Set Identification (PSI) in MABs, and Ararat and Tekin (2023), who provided gap-based sample complexity bounds under cone-based preferences. Korkmaz et al. (2023) extended these ideas to Gaussian process bandits, while Karagözlü et al. (2024) developed adaptive elimination algorithms for learning the Pareto front under incomplete preferences. Crepon et al. (2024) proposed a gradient-based track-and-stop strategy for exact Pareto front identification with known preference cones.

**Pure exploration with multiple correct answers.** We further connect the premise of PrePEx problem with the literature on pure exploration with multiple correct answers (Degenne and Koolen, 2019; Wang et al., 2021). Unlike standard BAI, this setting assumes existence of multiple optimal answers, though proceeds to identify any one of them. The philosophy of top-$K$ type strategies (Jourdan, 2024; Jourdan et al., 2022; Chen et al., 2017; You et al., 2023) extend this setting by efficiently identifying the top-$k$ optimal answers. On the contrary, in PrePEx we aim to find not k, but all of the correct answers (Pareto optimal arms), being as frugal as possible.

**Duelling bandits.** Preference-based bandit problems have traditionally been studied in the dueling bandit framework, that is limited to pairwise comparisons between arms (Zoghi et al., 2015; Busa-Fekete et al., 2014; Szörényi et al., 2015; Chen and Frazier, 2017). These models focus on learning a global ranking or identifying a *Condorcet* or *Copeland winner* based on binary preference outcomes. While such frameworks capture relative preference information effectively, they typically assume a fixed, often discrete preference structure and are studied under regret minimization. In contrast, PrePEx generalises this idea by considering vector-valued feedback and encoding preferences via convex cones, which enables us to extend towards more complex, continuous, and possibly incomplete preference structures. Unlike dueling bandits, PrePEx targets pure exploration with statistical guarantees, hoping to identify all the Pareto optimal arms with high confidence under a richer preference model.

**Connection to constrained pure exploration and safe RL.** PrePEx also generalises the setting of pure exploration under *known* linear constraints (Carlsson et al., 2024) though the notion of preference cone over objective is redundant for single objective bandit instances. In Safe RL, we consider presence of different risk constraints (e.g., on fairness, resource allocation etc.) (Achiam et al., 2017; Gu et al., 2024). This setting resonates with PrePEx as these constraints can be modelled as conflicting objectives while performing optimisation. The preference cone in PrePEx can also encode these constraint if they are implicit (one objective must not worsen).

# B  Structures of Confusing Instance

In this section, we will first define the Pareto policy set and prove some of its useful properties in B.1. Then, in B.2 we will introduce the novel polar cone characterisation of the alternating instances and finally in B.3 we will show the closed form of the alternating instance under multivariate Gaussian reward vectors with non-diagonal variance-covariance matrix.

## B.1  Pareto Optimal Policies to Pareto Optimal Arms

**Definition 7** (Pareto Policies). *The set of Pareto policies is given by* $\Pi^P \triangleq \left\{ \boldsymbol{\pi} : M^\top \tilde{\boldsymbol{\pi}} \preceq_{\bar{\mathcal{C}}} M^\top \boldsymbol{\pi} \ \forall \ \tilde{\boldsymbol{\pi}} \in \Delta_K \setminus \{\boldsymbol{\pi}\} \right\}$.

We now have some useful results regarding the set of Pareto policies.

**Lemma 3** (Compactness of Pareto Policy set). *The set of Pareto policies $\Pi^P$ is a compact set.*

*Proof.* To prove compactness of $\Pi^P$, we leverage Heine-Borel theorem (refer Theorem 14). It is evident that $\Pi^P$ is an subspace of the Euclidean space $\mathbb{R}^K$. Since complement of $\Pi^P$ is an open set, then $\Pi^P$ is closed. It is also bounded because all elements in $\Pi^P$ are policies which consist of entries bounded in the interval $[0, 1]$. Thus, according to Heine-Borel theorem we can equivalently state that $\Pi^P$ is compact, or in other words every open cover of $\Pi^P$ has a finite sub-cover. $\qquad\square$

**Theorem 2** (Basis of $\Pi^P$). *Pareto optimal policy set $\Pi^P$ is spanned by $p$ pure policies corresponding to $p$ Pareto optimal arms, i.e. $\{\boldsymbol{\pi}_i^\star\}_{i=1}^p$. Here, $\boldsymbol{\pi}_i^\star$ is the pure policy with support on only arm $i$.*

*Proof.* Consider the set of pure policies whose support lies on the Pareto front. For each $i \in \mathcal{P}^*$ define the pure policy $\boldsymbol{\pi}_i^* \in [0, 1]^K$ associated with arm $i$ as:

$$\boldsymbol{\pi}_i^*[j] \quad = \quad \begin{cases} 1, \ j = i \\ 0, \ \text{otherwise} \end{cases}$$

Further, since $i \in \mathcal{P}^*$, $M^\top \boldsymbol{\pi} \preceq_{\bar{\mathcal{C}}} M^\top \tilde{\boldsymbol{\pi}}_i, \ \forall \ \boldsymbol{\pi} \in \Pi \setminus \Pi^P$. With this construction, viewing each policy as a vector in $[0, 1]^K$, we have that the set of pure policies is linearly independent, i.e.,

$$c_1 \tilde{\boldsymbol{\pi}}_1 + c_2 \tilde{\boldsymbol{\pi}}_2 + \ldots + c_{|\mathcal{P}^*|} \tilde{\boldsymbol{\pi}}_{|\mathcal{P}^*|} = 0 \implies c_1 = c_2 = \ldots = c_{|\mathcal{P}^*|} = 0$$

Let $\boldsymbol{\pi} \in \Pi^P \setminus \Pi^{P,\text{basis}}$, where $\Pi^{P,\text{basis}}$ is the set of pure Pareto strategies. Since $\boldsymbol{\pi}$ is a randomized policy (Lemma 3) there exists constants $p_j \geq 0$, $\sum_j p_j = 1$, such that $\boldsymbol{\pi} = p_1 \tilde{\boldsymbol{\pi}}_1 + p_2 \tilde{\boldsymbol{\pi}}_2 + \ldots + p_{|\mathcal{P}^*|} \tilde{\boldsymbol{\pi}}_{|\mathcal{P}^*|}$. By linearity of dot product, $M^\top \boldsymbol{\pi} = p_1 M^\top \tilde{\boldsymbol{\pi}}_1 + p_2 M^\top \tilde{\boldsymbol{\pi}}_2 \ldots + p_{|\mathcal{P}^*|} M^\top \tilde{\boldsymbol{\pi}}_{|\mathcal{P}^*|}$. Since each $\tilde{\boldsymbol{\pi}}_i \in \Pi^P$ and $p_j \geq 0$, $p_1 M^\top \tilde{\boldsymbol{\pi}}_1 + p_2 M^\top \tilde{\boldsymbol{\pi}}_2 + \ldots + p_{|\mathcal{P}^*|} M^\top \tilde{\boldsymbol{\pi}}_{|\mathcal{P}^*|} \not\preceq_{\bar{\mathcal{C}}} M^\top \boldsymbol{\pi}'$ for all $\boldsymbol{\pi}' \in \Pi \setminus \Pi^P$.

Thus, any policy $\boldsymbol{\pi} \in \Pi^P$ can be expressed as a linear combination of the Pareto pure strategies $\boldsymbol{\pi}_i \in \Pi^{P,\text{basis}}, i \in \mathcal{P}^*$. Hence, proved. $\qquad\square$

## B.2  Polar Cone Characterization of Alternative Instances

**Proposition 1** (Polar Cone Representation of Alternating Instances). *For all $M \in \mathcal{M}$ and given $\boldsymbol{\pi}_i^\star \in \Pi^P$ and $\boldsymbol{\pi}_j \in \text{nbd}(\boldsymbol{\pi}_i^\star)$, the set of alternating instances takes the explicit form $\Lambda_{ij}(M) = \{\tilde{M} \in \mathcal{M} \setminus \{M\} : \exists \mathbf{y} \in \text{bd}(\bar{\mathcal{C}}^\circ) \text{ such that } \tilde{M} \boldsymbol{\pi}_j = \tilde{M} \boldsymbol{\pi}_i^\star + \mathbf{y}\}$ where $\mathbf{y}$ are defined with the polar cone of $\bar{\mathcal{C}}$, i.e. $\bar{\mathcal{C}}^\circ \triangleq \{\mathbf{y} : \mathbf{y} \in \mathbb{R}_+^L \text{ s.t. } \langle \mathbf{z}, \mathbf{y} \rangle \leq 0, \forall \mathbf{z} \in \bar{\mathcal{C}}^+\}$.*

*Proof.* Let us remind the definition of the set of confusing instances

$$\Lambda_{ij}(M) \triangleq \left\{ \tilde{M} \in \mathcal{M} \setminus \{M\} : \exists \mathbf{z} \in \bar{\mathcal{C}}^+, \ \langle \text{vect}\left(\mathbf{z}(\boldsymbol{\pi}_j - \boldsymbol{\pi}_i^\star)^\top\right), \text{vect}(\tilde{M}) \rangle = 0 \right\},$$

which implies that $\tilde{M}(\boldsymbol{\pi}_j - \boldsymbol{\pi}_i^\star) \in \text{bd}(\bar{\mathcal{C}}^\circ)$. We characterize the polar cone, using Farkas' lemma. For $\mathbf{x} \in \bar{\mathcal{C}}, \mathbf{x} = \sum_{i=1}^{L'} \alpha_i w_i, \alpha_i > 0 \forall i \in [L]$, where $W^{L \times L} = \{w_1, w_2, ..., w_L\}$, $w_i$'s being the basis rays of the cone $\bar{\mathcal{C}}$.

Using Farkas' lemma, the polar cone can be characterized as $\bar{\mathcal{C}}^\circ \triangleq \{W^\top \boldsymbol{\rho} : \boldsymbol{\rho} \in \mathbb{R}_+^L$ satisfies $\langle \mathbf{z}, W^\top \boldsymbol{\rho} \rangle \le 0, \forall \mathbf{z} \in \bar{\mathcal{C}}\}$. Therefore, by Projection Lemma, $\tilde{M}(\boldsymbol{\pi}_j - \boldsymbol{\pi}_i^\star) = W^\top \boldsymbol{\rho} \in \mathrm{bd}(\bar{\mathcal{C}}^\circ)$. Thus, we can write

$$\Lambda_{ij}(M) \triangleq \left\{ \tilde{M} \in \mathcal{M} \setminus \{M\} : \exists W^\top \boldsymbol{\rho} \in \mathrm{bd}(\bar{\mathcal{C}}^\circ) \text{ such that } \tilde{M}\boldsymbol{\pi}_j = \tilde{M}\boldsymbol{\pi}_i^\star + W^\top \boldsymbol{\rho} \right\}.$$

Defining $\mathbf{y} = W^\top \boldsymbol{\rho}$ concludes the proof. $\qquad\square$

### B.3 Characterisation of Confusing Instances for Multivariate Gaussians

**Lemma 4** (Confusing instances for Multivariate Gaussian rewards). *Let the reward vectors follow Multi-variate Gaussian distributions with diagonal covariance matrix $\Sigma$ with non-zero diagonal entries. Under the polar cone characterisation of the alternating instance, the polar vector has the closed form expression $\mathbf{y} = M\Delta(i,j) - \mathbf{z}^\top M\Delta(i,j)(\mathbf{z}\mathbf{z}^\top)^\dagger \mathbf{z}$, where $\Delta(i,j) \triangleq (\boldsymbol{\pi}_i^\star - \boldsymbol{\pi}_j)$, $\Sigma_0 \triangleq \mathbf{z}^\top \Sigma \mathbf{z}$ for given $\boldsymbol{\pi}_i^\star \in \Pi^P$ and $\boldsymbol{\pi}_j \in \mathrm{nbd}(\boldsymbol{\pi}_i^\star)$. Also, $A^\dagger$ denotes the pseudo-inverse of matrix $A$. The inverse characteristic time is then given by*

$$\mathcal{T}_{\mathcal{M},\bar{\mathcal{C}}}^{-1} = \max_{\boldsymbol{\omega} \in \Delta_K} \min_{\boldsymbol{\pi}_i^\star \in \Pi^P} \min_{\boldsymbol{\pi}_j \in \mathrm{nbd}(\boldsymbol{\pi}_i^\star)} \min_{\mathbf{z} \in \bar{\mathcal{C}}^+} \frac{\left(\mathbf{z}^\top M\Delta(i,j)\right)^2}{2\Sigma_0 \|\Delta(i,j)\|_{\mathrm{Diag}(1/\boldsymbol{\omega}_k)}^2}$$

*Proof.* First, for a fixed $\boldsymbol{\pi}_i^\star \in \Pi^P$ and $\boldsymbol{\pi}_j \in \mathrm{nbd}(\boldsymbol{\pi}_i^\star)$ we proceed by looking at the main optimisation problem under correlated Gaussian assumption,

$$\min_{\tilde{M} \in \Lambda_{ij}(M)} \min_{\mathbf{z} \in \bar{\mathcal{C}}^+} \sum_{k=1}^K \boldsymbol{\omega}_k D_{\mathrm{KL}}\left(\mathbf{z}^\top M_k \,\big\|\, \mathbf{z}^\top \tilde{M}_k\right)$$

$$= \min_{\tilde{M} : \mathbf{z}^\top \tilde{M}(\boldsymbol{\pi}_i^\star - \boldsymbol{\pi}_j) = 0} \min_{\mathbf{z} \in \bar{\mathcal{C}}^+} \sum_{k=1}^K \boldsymbol{\omega}_k \frac{(\mathbf{z}^\top M_k - \mathbf{z}^\top \tilde{M}_k)^2}{2\mathbf{z}^\top \Sigma \mathbf{z}}$$

where the last line holds due to the definition of Alt-set $\bar{\partial}\Lambda_{ij}(M)$. Thus, we incorporate the boundary constraint and write the Lagrangian dual with Lagrangian multiplier $\gamma > 0$ as

$$\mathcal{L}(\tilde{M}, \gamma) = \min_{\mathbf{z} \in \bar{\mathcal{C}}^+} \sum_{k=1}^K \left( \boldsymbol{\omega}_k \frac{(\mathbf{z}^\top M_k - \mathbf{z}^\top \tilde{M}_k)^2}{2\mathbf{z}^\top \Sigma \mathbf{z}} + \gamma \mathbf{z}^\top \tilde{M}_k (\boldsymbol{\pi}_i^\star - \boldsymbol{\pi}_j)_k \right) \qquad (9)$$

We differentiate (9) with respect to $\tilde{M}_k$ and equate it to zero to get the minima

$$\nabla_{\tilde{M}_k} \mathcal{L}(\tilde{M}, \gamma) = 0$$

$$\implies \boldsymbol{\omega}_k \frac{\mathbf{z}^\top M_k - \mathbf{z}^\top \tilde{M}_k}{\mathbf{z}^\top \Sigma \mathbf{z}} = \gamma(\boldsymbol{\pi}_i^\star - \boldsymbol{\pi}_j)_k$$

$$\implies \mathbf{z}^\top \tilde{M}_k = \mathbf{z}^\top M_k - \frac{\gamma(\boldsymbol{\pi}_i^\star - \boldsymbol{\pi}_j)_k \mathbf{z}^\top \Sigma \mathbf{z}}{\boldsymbol{\omega}_k}$$

We plug back the value of $\mathbf{z}^\top \tilde{M}_k$ in (9) to get

$$\mathcal{L}(\gamma) = \sum_{k=1}^K \left( \gamma \mathbf{z}^\top M_k (\boldsymbol{\pi}_i^\star - \boldsymbol{\pi}_j)_k - \frac{\gamma^2 (\boldsymbol{\pi}_i^\star - \boldsymbol{\pi}_j)_k^2 \mathbf{z}^\top \Sigma \mathbf{z}}{2\boldsymbol{\omega}_k} \right) \qquad (10)$$

We again differentiate (10) with respect to $\gamma$ and equate it to zero to get closed form of $\gamma$,

$$\nabla_\gamma \mathcal{L}(\gamma) = 0$$

$$\implies \gamma \sum_{k=1}^K \frac{(\boldsymbol{\pi}_i^\star - \boldsymbol{\pi}_j)_k^2 \mathbf{z}^\top \Sigma \mathbf{z}}{\boldsymbol{\omega}_k} = \sum_{k=1}^K \mathbf{z}^\top M_k (\boldsymbol{\pi}_i^\star - \boldsymbol{\pi}_j)_k$$

$$\implies \gamma = \frac{\sum_{k=1}^K \mathbf{z}^\top M_k (\boldsymbol{\pi}_i^\star - \boldsymbol{\pi}_j)_k}{\mathbf{z}^\top \Sigma \mathbf{z} \sum_{k=1}^K \frac{(\boldsymbol{\pi}_i^\star - \boldsymbol{\pi}_j)_k^2}{\boldsymbol{\omega}_k}}$$

We define $\Delta(i,j) \triangleq (\boldsymbol{\pi}_i^\star - \boldsymbol{\pi}_j)$ and $\Sigma_0 \triangleq \mathbf{z}^\top \Sigma \mathbf{z}$ and lastly $\mathrm{Diag}(1/\boldsymbol{\omega}_k)$ as a $K \times K$ diagonal matrix with $k$-th entry as $\frac{1}{\boldsymbol{\omega}_k}$. Then $\gamma = \frac{\mathbf{z}^\top M \Delta(i,j)}{\Sigma_0 \|\Delta(i,j)\|^2_{\mathrm{Diag}(1/\boldsymbol{\omega}_k)}}$. Thus, we finally have

$$\mathbf{z}^\top \tilde{M}_k = \mathbf{z}^\top M_k - \frac{\mathbf{z}^\top M \Delta(i,j)}{\|\Delta(i,j)\|^2_{\mathrm{Diag}(1/\boldsymbol{\omega}_k)}} \frac{\Delta(i,j)_k}{\boldsymbol{\omega}_k}$$

$$\implies \mathbf{z}^\top \tilde{M} = \mathbf{z}^\top M - \frac{\mathbf{z}^\top M \Delta(i,j)}{\|\Delta(i,j)\|^2_{\mathrm{Diag}(1/\boldsymbol{\omega}_k)}} \Delta_{\boldsymbol{\omega}}$$

$$\implies \tilde{M} = M - \frac{\mathbf{z}^\top M \Delta(i,j)}{\|\Delta(i,j)\|^2_{\mathrm{Diag}(1/\boldsymbol{\omega}_k)}} (\mathbf{z}\mathbf{z}^\top)^\dagger \mathbf{z} \Delta_{\boldsymbol{\omega}}$$

where $\Delta_{\boldsymbol{\omega}}$ is a $K$-dimensional vectors with $k$-th component being $\frac{\Delta(i,j)_k}{\boldsymbol{\omega}_k}$ and $A^\dagger$ denotes the pseudo-inverse of matrix $A$. Now, from the polar cone characterisation discussed in Appendix B.2 we know $\tilde{M}\Delta(i,j) = \mathbf{y} \in \mathrm{bd}(\bar{\mathcal{C}}^\circ)$. Thus we derive the closed form of the polar cone vector as

$$\mathbf{y} = M \Delta(i,j) - \mathbf{z}^\top M \Delta(i,j) (\mathbf{z}\mathbf{z}^\top)^\dagger \mathbf{z},$$

since $\langle \Delta_{\boldsymbol{\omega}}, \Delta(i,j) \rangle = \|\Delta(i,j)\|^2_{\mathrm{Diag}(1/\boldsymbol{\omega}_k)}$.

Therefore, we also get the closed form of the inverse characteristic time for Multivariate Gaussian rewards as well.

$$\mathcal{T}^{-1}_{\mathcal{M},\bar{\mathcal{C}}} = \max_{\boldsymbol{\omega} \in \Delta_K} \min_{\boldsymbol{\pi}_i^\star \in \Pi^{\mathrm{P}}} \min_{\boldsymbol{\pi}_j \in \mathrm{nbd}(\boldsymbol{\pi}_i^\star)} \min_{\mathbf{z} \in \bar{\mathcal{C}}^+} \frac{\left(\mathbf{z}^\top M \Delta(i,j)\right)^2}{2\Sigma_0 \|\Delta(i,j)\|^2_{\mathrm{Diag}(1/\boldsymbol{\omega}_k)}} \mathbf{z}^\top (\mathbf{z}\mathbf{z}^\top)^\dagger \mathbf{z}$$

Note that $\mathbf{z}\mathbf{z}^\top$ is a rank-1 matrix and its pseudo-inverse satisfies all four Moore-Penrose conditions. Thus we can use the identity $(\mathbf{z}\mathbf{z}^\top)^\dagger = \frac{1}{\|\mathbf{z}\|^4} \mathbf{z}\mathbf{z}^\top$. Therefore $\mathbf{z}^\top (\mathbf{z}\mathbf{z}^\top)^\dagger \mathbf{z} = 1$. Hence, the final expression of the inverse characteristic time is given by

$$\mathcal{T}^{-1}_{\mathcal{M},\bar{\mathcal{C}}} = \max_{\boldsymbol{\omega} \in \Delta_K} \min_{\boldsymbol{\pi}_i^\star \in \Pi^{\mathrm{P}}} \min_{\boldsymbol{\pi}_j \in \mathrm{nbd}(\boldsymbol{\pi}_i^\star)} \min_{\mathbf{z} \in \bar{\mathcal{C}}^+} \frac{\left(\mathbf{z}^\top M \Delta(i,j)\right)^2}{2\Sigma_0 \|\Delta(i,j)\|^2_{\mathrm{Diag}(1/\boldsymbol{\omega}_k)}}$$

Hence, we conclude the proof. $\square$

# C Continuity Results

First, we remind the lower bound on the expected sample complexity of any $(1 - \delta)$-correct PrePEx algorithm,

**Theorem 1** (Lower Bound (Shukla and Basu, 2024)). *Given a bandit model $M \in \mathcal{M}$, a preference cone $\bar{\mathcal{C}}$, and a confidence level $\delta \in [0, 1)$, the expected stopping time of any $(1 - \delta)$-correct PrePEx algorithm, to identify the Pareto Optimal Set is*

$$\mathbb{E}[\tau_\delta] \geq \mathcal{T}_{M,\bar{\mathcal{C}}} \log \left( \frac{1}{2.4\delta} \right),  \tag{11}$$

*where, the expectation is taken over the stochasticity of both the algorithm and the bandit instance. Here, $\mathcal{T}_{M,\bar{\mathcal{C}}}$ is called the characteristic time of the PrePEx instance $(\mathcal{M}, \bar{\mathcal{C}})$ and is expressed as*

$$\left( \mathcal{T}_{M,\bar{\mathcal{C}}} \right)^{-1} \triangleq \max_{\boldsymbol{\omega} \in \Delta_K} \min_{\substack{\boldsymbol{\pi}_j \in \mathrm{nbd}(\boldsymbol{\pi}_i^*) \\ \boldsymbol{\pi}_i^* \in \Pi^P}} \min_{\tilde{M} \in \Lambda_{ij}(M)} \underbrace{\min_{\mathbf{z} \in \bar{\mathcal{C}}^+} \sum_{k=1}^{K} \boldsymbol{\omega}_k D_{\mathrm{KL}} \left( \mathbf{z}^\top M_k \,\big\|\, \mathbf{z}^\top \tilde{M}_k \right)}_{f_{ij}(\boldsymbol{\omega}, \tilde{M}|M)},  \tag{12}$$
$$\underbrace{\phantom{\min_{\tilde{M} \in \Lambda_{ij}(M)}}}_{f_{ij}(\boldsymbol{\omega}|M)}$$
$$\underbrace{\phantom{\min_{\substack{\boldsymbol{\pi}_j \in \mathrm{nbd}(\boldsymbol{\pi}_i^*)}}}}_{F(\boldsymbol{\omega}|M)}$$

*where for a fixed $\boldsymbol{\pi}_i^\star \in \Pi^P$ and $\boldsymbol{\pi}_j \in \mathrm{nbd}(\boldsymbol{\pi}_i^\star)$,*

$$\Lambda_{ij}(M) \triangleq \left\{ \tilde{M} \in \mathcal{M} \setminus \{M\} : \exists \, \mathbf{z} \in \mathcal{C}, \, \langle \mathrm{vect}\left( \mathbf{z}(\boldsymbol{\pi}_j - \boldsymbol{\pi}_i^\star)^\top \right), \mathrm{vect}(\tilde{M}) \rangle = 0 \right\}.$$

**Useful Notations:** For maintaining brevity, we state the useful notations related to functions under continuity analysis which are followed throughout this section and beyond.

1. First, we define the following functions. For a fixed $\tilde{M} \in \Lambda_{ij}(M)$,

$$f_{ij}(\boldsymbol{\omega}, \tilde{M}|M) \triangleq \min_{\mathbf{z} \in \bar{\mathcal{C}}^+} \sum_{k=1}^{K} \boldsymbol{\omega}_k D_{\mathrm{KL}} \left( \mathbf{z}^\top M_k \,\big\|\, \mathbf{z}^\top \tilde{M}_k \right)$$

2. Again we define for fixed $\boldsymbol{\pi}_i^\star \in \Pi^P$ and $\boldsymbol{\pi}_j \in \mathrm{nbd}(i)$,

$$f_{ij}(\boldsymbol{\omega}|M) \triangleq \min_{\tilde{M} \in \Lambda_{ij}(M)} f_{ij}(\boldsymbol{\omega}, \tilde{M}|M)$$

3. Finally, $F(\boldsymbol{\omega} \mid M) \triangleq \min_{\substack{\boldsymbol{\pi}_j \in \mathrm{nbd}(\boldsymbol{\pi}_i^*) \\ \boldsymbol{\pi}_i^* \in \Pi^P}} f_{ij}(\boldsymbol{\omega} \mid M)$

**Fact 1.** *Due to convexity of KL divergence with respect to $\mathbf{z} \in \bar{\mathcal{C}}^+$ and convexity of the solid cone $\bar{\mathcal{C}}^+$, $f_{ij}(\boldsymbol{\omega}|M)$ is convex in $\mathbf{z}$ and $\tilde{M}$, but concave with respect to $\boldsymbol{\omega}$ (Minimum over concave functions).*

**Fact 2.** *As a consequence of Fact 1, $F(\boldsymbol{\omega} \mid M)$, being minimum among finite concave function, is concave in $\boldsymbol{\omega}$, but a non-smooth function. This function is smooth only at the points where the minimum is reached for $\boldsymbol{\pi}_i^\star$ and $\boldsymbol{\pi}_j$.*

In C.1, we state and prove continuity results of the peeled objective functions with respect to alternating instance and the preference vector.

## C.1 Continuity w.r.t. Preferences and Alternating Instances: $f_{ij}(\boldsymbol{\omega}, \tilde{M}|M)$ and $f_{ij}(\boldsymbol{\omega}|M)$

**Proposition 2.** *Let $S_i \subset \mathcal{M}$ be the set of mean matrices for which $\boldsymbol{\pi}_i^\star$ is the Pareto optimal policy. For given $\boldsymbol{\pi}_i^\star \in \Pi^P, \boldsymbol{\pi}_j \in \mathrm{nbd}(\boldsymbol{\pi}_i^\star)$, then for all $(\boldsymbol{\omega}, M) \in \Delta_K \times S_i$,*

*(a) For a given $\tilde{M} \in \mathcal{M} \setminus M$, there exists a unique $\mathbf{z}_{\mathrm{inf}} \in \mathrm{int}(\bar{\mathcal{C}}^+)$ such that*

$$\mathbf{z}_{\mathrm{inf}} \triangleq \arg\min_{\mathbf{z} \in \mathrm{int}(\bar{\mathcal{C}}^+)} \sum_{k=1}^{K} \boldsymbol{\omega}_k D_{\mathrm{KL}} \left( \mathbf{z}^\top M_k \,\big\|\, \mathbf{z}^\top \tilde{M}_k \right)$$

*if $\mathbf{z}^\top M_k \neq 0$ and $\mathbf{z}^\top \tilde{M}_k \neq 0$ for all $k = [1, K]$.*

*(b) there exists a unique $\tilde{M}_{\text{inf}}$ such that*

$$\tilde{M}_{\text{inf}} \triangleq \arg\min_{\tilde{M} \in \Lambda_{ij}(M)} \sum_{k=1}^{K} \boldsymbol{\omega}_k D_{\text{KL}} \left( \mathbf{z}_{\text{inf}}^\top M_k \,\middle\|\, \mathbf{z}_{\text{inf}}^\top \tilde{M}_k \right)$$

*(c) As $f_{\boldsymbol{\pi}}$ is continuously differentiable on $\Delta_K \times S_i$, the gradient is expressed as*

$$\nabla_{\boldsymbol{\omega}} f_{ij}(\boldsymbol{\omega}|M) = \min_{\tilde{M}, \mathbf{z} \in \bar{\mathcal{C}}^+} \sum_{k=1}^{K} D_{\text{KL}} \left( \mathbf{z}^\top M_k \,\middle\|\, \mathbf{z}^\top \tilde{M}_k \right) e_k$$

*where, $e_k$ is a $K$-dimensional vector with $1$ in $k$-th index, $0$ elsewhere.*

*Proof.* **Proof of Part (a).** We do this proof in two parts. First we prove the existence of $\mathbf{z}_{\text{inf}}$ and then its uniqueness.

**Existence.** We refer to Theorem 11 to prove the first part. We define $\mathbb{X} \triangleq \Delta_K \times S_i$, $\mathbb{Y} \triangleq \text{int}(\bar{\mathcal{C}}^+)$, $\Phi \triangleq \text{int}(\bar{\mathcal{C}}^+)$ and $u(\boldsymbol{\omega}, M) \triangleq \sum_{k=1}^{K} \boldsymbol{\omega}_k D_{\text{KL}} \left( \mathbf{z}^\top M_k \,\middle\|\, \mathbf{z}^\top \tilde{M}_k \right)$. Now $\Phi : (\Delta_K \times S_i) \rightrightarrows \text{int}(\bar{\mathcal{C}}^+)$ is a *constant correspondence* (as it is defined before computing $\boldsymbol{\omega}$ and $M$) and $u : \Delta_K \times S_i \times \text{int}(\bar{\mathcal{C}}^+) \to \mathbb{R}$ is a continuous mapping. Thus $f_{ij}(\boldsymbol{\omega}, \tilde{M}|M)$ is continuous. Hence, $z_{\text{inf}}$ exists.

**Uniqueness.** Now if we can prove that the KL-divergence is strictly convex in $\text{int}(\bar{\mathcal{C}}^+)$, then $\mathbf{z}_{\text{inf}}$ is unique. For strict convexity, we need to show that the second derivative (Hessian) of the KL divergence w.r.t. $\mathbf{z}$ is positive, which can be tricky if the function has flat regions or degenerate directions (which could arise at the boundary of the polyhedral cone). Here, we leverage Lemma 5 that shows that if $\mathbf{z}^\top M_k \neq 0$ and $\mathbf{z}^\top \tilde{M}_k \neq 0$ for all $k = [1, K]$, then KL is strictly convex on $\mathbf{z}$. That means as $\mathbf{z}_{\text{inf}} \in \text{int}(\bar{\mathcal{C}}^+)$, for any $k \in [K]$, $M_k$ and $\tilde{M}_k$ cannot lie in $\text{int}(\bar{\mathcal{C}}^\circ)$. Thus, $\mathbf{z}_{\text{inf}}$ exists and it is unique. Hence, proved.

**Proof of Part (b).** We again leverage Theorem 11 to prove existence of a unique $\tilde{M}_{\text{inf}}$. We define $\mathbb{X} \triangleq \Delta_K \times S_i$, $\mathbb{Y} = \Lambda_{ij}(M)$, $\Phi = \Lambda_{ij}(M)$ and $u(\boldsymbol{\omega}, M) \triangleq \sum_{k=1}^{K} \boldsymbol{\omega}_k D_{\text{KL}} \left( \mathbf{z}_{\text{inf}}^\top M_k \,\middle\|\, \mathbf{z}_{\text{inf}}^\top \tilde{M}_k \right)$. Now $\Phi : (\Delta_K \times S_i) \rightrightarrows \Lambda_{ij}(M)$ is a *constant correspondence* (as we already fix $\boldsymbol{\pi}_i^\star \in \Pi^P$ and $\boldsymbol{\pi}_j \in \text{nbd}(\boldsymbol{\pi}_i^\star)$, $\Lambda_{ij}(M)$ does not depend on $\boldsymbol{\pi}_i^\star, \boldsymbol{\pi}_j$) and $u : \Delta_K \times S_i \times \Lambda_{ij}(M) \to \mathbb{R}$ is a continuous mapping. Therefore, $\tilde{M}_{\text{inf}}$ is upper-hemicontinuous and $f_{ij}(\boldsymbol{\omega}|M)$ is continuous on $\Lambda_{ij}(M)$. Again, the KL-divergence follows strict convexity in $\Lambda_{ij}(M)$, therefore proving uniqueness of $\tilde{M}_{\text{inf}}$.

**Proof of Part (c).** We leverage Lemma 11 with the following definitions $\mathbb{X} \triangleq \Lambda_{ij}(M) \times \text{int}(\bar{\mathcal{C}}^+)$, $Y \triangleq \Delta_K \times S_i$, $\mathbf{x}^\star(\boldsymbol{\omega}, \mathcal{M}) \triangleq \mathbf{z}_{\text{inf}}^\top (\tilde{M}_{\text{inf}})_k$ and $u(\mathbf{x}^\star, \boldsymbol{\omega}, \mathcal{M}) \triangleq \sum_{k=1}^{K} \boldsymbol{\omega}_k D_{\text{KL}} \left( \mathbf{z}_{\text{inf}}^\top M_k \,\middle\|\, \mathbf{z}_{\text{inf}}(\tilde{M}_{\text{inf}})_k \right)$. This simply proves $u(\mathbf{x}^\star, \boldsymbol{\omega}, M)$ is continuously differentiable by the virtue of continuity of

$\square$

**Lemma 5.** *For all $(\boldsymbol{\omega}, M) \in \Delta_K \times S_i$, $D_{\text{KL}} \left( \mathbf{z}^\top M \,\middle\|\, \mathbf{z}^\top \tilde{M} \right)$ is strictly convex on $\mathbf{z}$ iff $\mathbf{z}^\top \tilde{M}_k \neq 0$ and $\mathbf{z}^\top M_k \neq 0 \forall k = [1, K]$.*

*Proof.* The KL-divergence has the following form

$$D_{\text{KL}} \left( \mathbf{z}^\top M \,\middle\|\, \mathbf{z}^\top \tilde{M} \right) = \sum_{k=1}^{K} \sum_{l=1}^{L} (M_{k,l} \mathbf{z}_l) \log \left( \frac{\sum_{l=1}^{L} M_{k,l} \mathbf{z}_l}{\sum_{l=1}^{L} \tilde{M}_{k,l} \mathbf{z}_l} \right)$$

Partially differentiating the KL with respect to $\mathbf{z}_l$ we get

$$\frac{\partial D_{\mathrm{KL}}\left(\mathbf{z}^\top M \,\middle\|\, \mathbf{z}^\top \tilde{M}\right)}{\partial \mathbf{z}_l} = \sum_{k=1}^{K} \left\{ M_{k,l} \log \left( \frac{\sum_{l=1}^{L} M_{k,l}\mathbf{z}_l}{\sum_{l=1}^{L} \tilde{M}_{k,l}\mathbf{z}_l} \right) + \frac{M_{k,l} \sum_{l=1}^{L} \tilde{M}_{k,l}\mathbf{z}_l - \tilde{M}_{k,l} \sum_{l=1}^{L} M_{k,l}\mathbf{z}_l}{\sum_{l=1}^{L} \tilde{M}_{k,l}\mathbf{z}_l} \right\}$$

The components of Hessian are expressed as

$$\begin{aligned}
H_{l,l'} &\triangleq \frac{\partial^2 D_{\mathrm{KL}}\left(\mathbf{z}^\top M \,\middle\|\, \mathbf{z}^\top \tilde{M}\right)}{\partial \mathbf{z}_l \partial \mathbf{z}_{l'}} \\
&= \frac{\partial}{\partial \mathbf{z}_{l'}} \sum_{k=1}^{K} \left\{ M_{k,l} \log \left( \frac{\sum_{l=1}^{L} M_{k,l}\mathbf{z}_l}{\sum_{l=1}^{L} \tilde{M}_{k,l}\mathbf{z}_l} \right) + M_{k,l} - \tilde{M}_{k,l} \frac{\sum_{l=1}^{L} M_{k,l}\mathbf{z}_l}{\sum_{l=1}^{L} \tilde{M}_{k,l}\mathbf{z}_l} \right\} \\
&= \sum_{k=1}^{K} M_{k,l} \frac{\sum_{l=1}^{L} \tilde{M}_{k,l}\mathbf{z}_l}{\sum_{l=1}^{L} M_{k,l}\mathbf{z}_l} \frac{M_{k,l'} \sum_{l=1}^{L} \tilde{M}_{k,l}\mathbf{z}_l - \tilde{M}_{k,l'} \sum_{l=1}^{L} M_{k,l}\mathbf{z}_l}{(\sum_{l=1}^{L} \tilde{M}_{k,l}\mathbf{z}_l)^2} \\
&\quad - \tilde{M}_{k,l} \frac{M_{k,l'} \sum_{l=1}^{L} \tilde{M}_{k,l}\mathbf{z}_l - \tilde{M}_{k,l'} \sum_{l=1}^{L} M_{k,l}\mathbf{z}_l}{(\sum_{l=1}^{L} \tilde{M}_{k,l}\mathbf{z}_l)^2} \\
&= \sum_{k=1}^{K} M_{k,l} M_{k,l'} \frac{1}{\sum_{l=1}^{L} M_{k,l}\mathbf{z}_l} - \sum_{k=1}^{K} \left( M_{k,l}\tilde{M}_{k,l'} + \tilde{M}_{k,l}M_{k,l'} \right) \frac{1}{\sum_{l=1}^{L} \tilde{M}_{k,l}\mathbf{z}_l} \\
&\quad + \sum_{k=1}^{K} \tilde{M}_{k,l}\tilde{M}_{k,l'} \frac{\sum_{l=1}^{L} M_{k,l}\mathbf{z}_l}{(\sum_{l=1}^{L} \tilde{M}_{k,l}\mathbf{z}_l)^2} \\
&= \sum_{k=1}^{K} \left( \sum_{l=1}^{L} M_{k,l}\mathbf{z}_l \right) \left( \frac{M_{k,l}M_{k,l'}}{(\sum_{l=1}^{L} M_{k,l}\mathbf{z}_l)^2} - \frac{\left( M_{k,l}\tilde{M}_{k,l'} + \tilde{M}_{k,l}M_{k,l'} \right)}{(\sum_{l=1}^{L} M_{k,l}\mathbf{z}_l)(\sum_{l=1}^{L} \tilde{M}_{k,l}\mathbf{z}_l)} + \frac{\tilde{M}_{k,l}\tilde{M}_{k,l'}}{(\sum_{l=1}^{L} \tilde{M}_{k,l}\mathbf{z}_l)^2} \right)
\end{aligned}$$

Thus, the Hessian of KL is positive definite iff for any non-zero $\mathbf{x} \in \mathbb{R}^L$,

$$\begin{aligned}
&\mathbf{x}^\top H x > 0 \\
\implies &\sum_{k=1}^{K} \left( \frac{(\mathbf{x}^\top M_k)^2}{A_k} - 2\frac{(\mathbf{x}^\top M_k)(\mathbf{x}^\top \tilde{M}_k)}{\tilde{A}_k} + \frac{A_k}{\tilde{A}_k^2}(\mathbf{x}^\top \tilde{M}_k)^2 \right) > 0 \\
\implies &\frac{\tilde{A}_k}{A_k}(\mathbf{x}^\top M_k)^2 + \frac{A_k}{\tilde{A}_k}(\mathbf{x}^\top \tilde{M}_k)^2 > 2(\mathbf{x}^\top M_k)(\mathbf{x}^\top \tilde{M}_k) \\
\implies &\left( \sqrt{\frac{A_k}{\tilde{A}_k}}\mathbf{x}^\top \tilde{M}_k - \sqrt{\frac{\tilde{A}_k}{A_k}}\mathbf{x}^\top M_k \right)^2 > 0
\end{aligned}$$

This statement is always true if $A_k = \sum_{l=1}^{L} M_{k,l}\mathbf{z}_l \neq 0$ and $\tilde{A}_k = \tilde{M}_{k,l}\mathbf{z}_l \neq 0$. Hence, we conclude the proof. $\qquad\square$

# D Convergence of FW Algorithms

**Fact 3.** *Frank-Wolfe is a second order method. For it to converge, we must ensure the gradient of the objective with respect to $\boldsymbol{\omega}$ is bounded around the non-smooth points and also the curvature constant must be bounded.*

In this section, we first derive upper bounds on gradient and curvature constant in D.1. Then we move on the state and prove continuity properties of the sub-differential set defined in (6) in D.2. Finally, unifying continuity results from C and results derived in this section, we elaborate on the convergence of Frank-Wolfe in D.3.

## D.1 Boundedness of Gradient and Curvature of Sub-Differentials

Before jumping into the proof of Lemma 1, we first define the curvature constant and a subset of simplex that ensures the minimum index of $\boldsymbol{\omega}$ does not fall on the boundary of the simplex.

**Definition 8.** *For $\boldsymbol{\omega}, \boldsymbol{\omega}' \in \Delta_{K\gamma}$, $\alpha \in (0,1]$ and given the bandit instance $M \in \mathcal{M}$, the curvature constant is expressed as,*

$$C_{f_{ij}(\cdot|M)}(\Delta_{K\gamma}) = \sup_{\substack{\boldsymbol{x}, \boldsymbol{z} \in \Delta_{K\gamma} \\ \alpha \in (0,1] \\ \boldsymbol{y} = \boldsymbol{x} + \alpha(\boldsymbol{z} - \boldsymbol{x})}} \frac{2}{\alpha^2}[\psi(\boldsymbol{x}) - \psi(\boldsymbol{y}) + \langle \boldsymbol{y} - \boldsymbol{x}, \nabla\psi(\boldsymbol{x})\rangle].$$

**Definition 9.** *For any $\gamma \in (0, 1/K)$, we define a subset of the simplex $\Delta_K$ as*

$$\Delta_{K\gamma} \triangleq \{\boldsymbol{\omega} \in \Delta_K : \min_k \boldsymbol{\omega}_k \geq \gamma\}$$

With these definitions in hand, now we show that the space of $f_{ij}(\boldsymbol{\omega}|M)$ had bounded gradient and curvature w.r.t. $\boldsymbol{\omega}$.

**Lemma 1.** *If Assumption 1 holds true, then for all $M \in \mathcal{M}$: 1. **Bounded gradients:** $\|\nabla_{\boldsymbol{\omega}} f_{ij}(\boldsymbol{\omega}|M)\|_\infty \leq D$ for all $\boldsymbol{\pi}_i^\star, \boldsymbol{\pi}_j$, and $\boldsymbol{\omega} \in \Delta_K$. 2. **Bounded curvature:** $C_{f_{ij}(\cdot|M)}(\Delta_{K\gamma}) \leq 8D\alpha^{-1}$ for all $i, j$, $\gamma \in (0, 1/K)$, and some $\alpha > 0$. Here, $C_f(A)$ is curvature constant of concave differentiable function $f$ in set $A$ (Definition 8) and $\Delta_{K\gamma} \triangleq \{\boldsymbol{\omega} \in \Delta_K : \min_k \boldsymbol{\omega}_k \geq \gamma\}$.*

*Proof.* **Proof of 1.** We already have the expression for the gradient from Appendix C as

$$\nabla_{\boldsymbol{\omega}} f_{ij}(\boldsymbol{\omega}|M) = \min_{\tilde{M}, \mathbf{z} \in \mathcal{C}} \sum_{k=1}^{K} D_{\text{KL}}\left(\mathbf{z}^\top M \,\middle\|\, \mathbf{z}^\top \tilde{M}\right) e_k$$

Therefore

$$\|\nabla_{\boldsymbol{\omega}} f_{ij}(\boldsymbol{\omega}|M)\|_\infty = \|d(\mathbf{z}^\top M, \mathbf{z}^\top \tilde{M})\|_\infty = \max_{k \in [1,K]} |D_{\text{KL}}\left(\mathbf{z}^\top M_k \,\middle\|\, \mathbf{z}^\top \tilde{M}_k\right)|$$

We know for exponential families the KL-divergence can be expressed as the Bregman divergence generated by the Cumulant Generating function or Cramer function i.e,

$$D_{\text{KL}}(\boldsymbol{\mu}\|\boldsymbol{\mu}') = A(\boldsymbol{\mu}') - A(\boldsymbol{\mu}) - \nabla A(\boldsymbol{\mu})(\boldsymbol{\mu}' - \boldsymbol{\mu})$$

where $A(.)$ is the CGF and $\boldsymbol{\mu}, \boldsymbol{\mu}'$ belong to $L$-parameter exponential family according to Assumption 1 with natural parameter $\boldsymbol{\theta}, \boldsymbol{\theta}' \in \text{int}(\Theta)$. $D_{\text{KL}}(\boldsymbol{\mu}\|\boldsymbol{\mu}')$ is bounded if support of $\boldsymbol{\theta}$ and $\boldsymbol{\theta}'$ are same, which is true in our case.

**Example 1 : Univariate Gaussian.** Let, $p(x) = \mathcal{N}(\mu_1, \sigma_1^2)$ and $q(x) = \mathcal{N}(\mu_2, \sigma_2^2)$. Any of these densities has the following canonical exponential form $p(x) = \exp\left(\frac{\mu}{\sigma_1^2} x - \frac{1}{2}\theta^2\sigma_1^2 - \frac{x}{2\sigma_1^2} - \frac{1}{2}\log(2\pi\sigma_1^2)\right)$. Therefore for $p(x)$ the natural parameter is $\theta_1 = \begin{bmatrix} \frac{\mu_1}{\sigma_1^2} \\ -\frac{1}{2\sigma_1^2} \end{bmatrix}$ and for the $q(x)$ the natural parameter is $\theta_2 = \begin{bmatrix} \frac{\mu_2}{\sigma_2^2} \\ -\frac{1}{2\sigma_2^2} \end{bmatrix}$ and also the CGF is given by

$$A(\theta_i) = \frac{\theta_{i1}^2}{4\theta_{i2}} + \frac{1}{2}\log\left(-\frac{\pi}{\theta_{i2}}\right)$$

for $i = 1, 2$. Now the KL is always finite, smooth and well-behaved as long as the natural parameters belong to the same domain.

**Example 2: Bernoulli.** Let $p(x) = \text{Ber}(p), p \in (0, 1)$. Then the canonical exponential form can be written as $p(x) = \exp\left(x \log \frac{p}{1-p} + \log(1-p)\right)$. Therefore the natural parameter $\theta = \log \frac{p}{1-p}$, which is the log-odds and the CGF is $A(\theta) = \log(1 + e^\theta)$. So KL-divergence between any two Bernoulli random variable $X$ and $Y$ with mean parameters $p_1$ and $p_2$ is finite and well-behaved iff $p_1, p_2 \in (0, 1)$.

Therefore by the virtue of Assumption 1 we claim that $\exists D > 0$ such that the gradient is upper bounded i.e.,

$$\|\nabla_{\boldsymbol{\omega}} f_{ij}(\boldsymbol{\omega}|M)\|_\infty \leq D \forall i, j, \text{ and } \boldsymbol{\omega} \in \Delta_K.$$

**Proof of 2.** We observe from the proof of part 1, $f_{ij}(\boldsymbol{\omega} \mid M)$ is $D$-smooth, meaning for any $\boldsymbol{\omega}, \boldsymbol{\omega}' \in \Delta_{K\gamma}$

$$| f_{ij}(\boldsymbol{\omega}' \mid M) - f_{ij}(\boldsymbol{\omega} \mid M) | \leq \|\nabla_{\boldsymbol{\omega}} f_{ij}(\boldsymbol{\omega}|M)\|_\infty \|\boldsymbol{\omega}' - \boldsymbol{\omega}\|_1 \leq D \|\boldsymbol{\omega}' - \boldsymbol{\omega}\|_1$$

Then we start with the definition of the curvature constant

$$\begin{aligned}
C_{f_{ij}(\cdot|M)}(\Delta_{K\gamma}) = & \frac{2}{\alpha^2} \left[ f_{ij}(\boldsymbol{\omega}|M) - f_{ij}(\boldsymbol{\omega}''|M) + \langle \boldsymbol{\omega} - \boldsymbol{\omega}'', \nabla f_{ij}(\boldsymbol{\omega}|M) \rangle \right] \\
\leq & \frac{2}{\alpha^2} \left[ f_{ij}(\boldsymbol{\omega} \mid M) - (1-\alpha) f_{ij}(\boldsymbol{\omega} \mid M) + \alpha f_{ij}(\boldsymbol{\omega}'|M) \right. \\
& \left. + \|\nabla_{\boldsymbol{\omega}} f_{ij}(\boldsymbol{\omega}|M)\|_\infty \|\boldsymbol{\omega}' - \boldsymbol{\omega}\|_1 \right] \\
\leq & \frac{2}{\alpha^2} \left[ \alpha D \|\boldsymbol{\omega}' - \boldsymbol{\omega}\|_1 + \alpha D \|\boldsymbol{\omega}' - \boldsymbol{\omega}\|_1 \right] \\
\leq & \frac{4D}{\alpha} \|\boldsymbol{\omega}' - \boldsymbol{\omega}\|_1
\end{aligned}$$

Now, as $\boldsymbol{\omega}, \boldsymbol{\omega}' \in \Delta_{K\gamma}$, $\|\boldsymbol{\omega}' - \boldsymbol{\omega}\|_1 \leq 2$. Therefore, we have the final upper bound on the curvature constant as

$$C_{f_{ij}(\cdot|M)} \leq \frac{8D}{\alpha}$$

Hence, proved. $\qquad\square$

Interestingly, Wang et al. (2021) considered bounded gradients and bounded curvature as assumption. On the other hand, in our PrePEx we get this necessary conditions for convergence automatically leveraging Assumption 1, which is arguably a very generic assumption on the parametric family of the reward vectors and standard in the literature. Additionally, leveraging Lemma 1, Wang et al. (2021) could only show that boundedness of gradient and curvature constant is restricted towards only Bernoulli and Gaussian. Instead, we claim that Lemma 1 holds for any generic exponential family satisfying Assumption 1.

Now we move on the proving some good properties of the subdifferential set that handles the non-smoothness of $f_{ij}(\boldsymbol{\omega} \mid M)$.

## D.2 Continuity of the Frank-Wolfe Iterates: Sub-Differentials

Let's remind the definition of $r$-sub-differential set.

$$H_M(\boldsymbol{\omega}, r) \triangleq \text{ch}\left\{\nabla_{\boldsymbol{\omega}} f_{ij}(\boldsymbol{\omega}|M) : f_{ij}(\boldsymbol{\omega}|M) < \min_{\boldsymbol{\pi}_i^\star, \boldsymbol{\pi}_j} f_{ij}(\boldsymbol{\omega}|M) + r, \forall \boldsymbol{\pi}_i^\star \in \Pi^P, \boldsymbol{\pi}_j \in \text{nbd}\,(\boldsymbol{\pi}_i^\star)\right\}. \quad (13)$$

**Corollary 1.** *The mapping $\boldsymbol{\omega} \mapsto H_M(\boldsymbol{\omega}, r)$ is continuous.*

*Proof.* Let $\{\boldsymbol{\omega}_t, \hat{M}_t, r_t\}_{t=1}^\infty \longrightarrow \{\boldsymbol{\omega}^\star(M), M, r\} \in \Delta_K \times S_i \times (0, 1)$ and $H_M(\boldsymbol{\omega}, r) = \text{ch}\left\{\nabla_{\boldsymbol{\omega}} f_{ij}(\boldsymbol{\omega} \mid \hat{M})\right\}_{j=1}^{|\text{nbd}(\boldsymbol{\pi}_i^\star)|}$ for some $\boldsymbol{\pi}_j \in \text{nbd}\,(\boldsymbol{\pi}_i^\star)$. We can write for any $h \in H_M(\boldsymbol{\omega}, r), \exists$

a sequence of $\{\alpha_i \geq 0\}_{i=1}^{|\mathrm{nbd}(\boldsymbol{\pi}_i^\star)|}$ such that $h$ can be expressed as a convex combination of $\{\nabla_{\boldsymbol{\omega}} f_{ij}(\boldsymbol{\omega} \mid \hat{M})\}_{j=1}^{|\mathrm{nbd}(\boldsymbol{\pi}_i^\star)|}$. Further, leveraging continuity of $f_{ij}(\cdot \mid \hat{M})$ for all $j = [1, |\mathrm{nbd}(\boldsymbol{\pi}_i^\star)|]$(ref. Appendix C), we claim that $\nabla_{\boldsymbol{\omega}} f_{ij}(\boldsymbol{\omega}_t \mid \hat{M}_t) \in H_{\hat{M}_t}(\boldsymbol{\omega}_t, r_t)$ for some $t > N$.

**Lower Hemicontinuity of subdifferential.** Lower Hemicontinuity follows from the continuity result of $\nabla_{\boldsymbol{\omega}} f_{ij}$ derived in Appendix C.

**Upper Hemicontinuity of subdifferentials.** We adapt similar proof structure as Wang et al. (2021) by adding a $\epsilon$-radius (Minkowski addition) to $H_{\hat{M}_t}(\boldsymbol{\omega}_t, r_t)$ to show it is still contained by the open set containing $H_M(\boldsymbol{\omega}^\star(M), r)$. Thus, the subdifferential set is lower and upper hemicontinuous with respect to $\boldsymbol{\omega}$, i.e continuous. Hence, proved. □

Now, we prove the convergence of the Frank-Wolfe iterates. Let us remind the Frank-Wolfe steps once again for brevity,

$$\mathbf{x}_{t+1} \triangleq \arg\max_{\mathbf{x} \in \Delta_K} \min_{\mathbf{h} \in H_M(\boldsymbol{\omega}, r)} \langle \mathbf{x} - \boldsymbol{\omega}_t, \mathbf{h} \rangle \tag{14}$$

$$\boldsymbol{\omega}_{t+1} \triangleq \frac{t}{t+1} \boldsymbol{\omega}_t + \frac{1}{t+1} \mathbf{x}_{t+1} \tag{15}$$

First, we define the following maps

1. $\phi_1 : (\boldsymbol{\omega}, r, \hat{M}, \mathbf{x}) \mapsto \min_{\mathbf{h} \in H_{\hat{M}}(\boldsymbol{\omega}, r)} \langle \mathbf{x} - \boldsymbol{\omega}, \mathbf{h} \rangle$,

2. $\phi_2 : (\boldsymbol{\omega}, r, \hat{M}) \mapsto \max_{\mathbf{x} \in \Delta_K} \phi_1$.

**Corollary 2.** $\phi_1$ is continuous on $\Delta_{K\gamma} \times (0, 1) \times S_i \times \Delta_K$.

*Proof.* We again leverage Theorem 11 with the following definitions, $\mathbb{X} \triangleq \Delta_{K\gamma} \times (0, 1) \times S_i \times \Delta_K$, $\mathbb{Y} \triangleq \mathbb{R}^K$, $\Phi(\boldsymbol{\omega}, r, \hat{M}, \mathbf{x}) \triangleq H_{\hat{M}}(\boldsymbol{\omega}, r)$ and $u(\boldsymbol{\omega}, r, \hat{M}, \mathbf{x}, h) = \langle \mathbf{x} - \boldsymbol{\omega}, \mathbf{h} \rangle$. We are concerned only about continuity of the correspondence $\Phi$, as $u$ is a linear function, hence continuous. Though we have already proven continuity of $\Phi$ in Corollary 1. Thus, we conclude the proof. □

**Corollary 3.** $\phi_2$ is continuous on $\Delta_{K\gamma} \times (0, 1) \times S_i$.

*Proof.* We again apply Theorem 11 with the definitions $\mathbb{X} \triangleq \Delta_{K\gamma} \times (0, 1) \times S_i$, $\mathbb{Y} \triangleq \Delta_K$, $\Phi(\boldsymbol{\omega}, r, \hat{M}, \mathbf{x}) \triangleq \Delta_K$ and $u(\boldsymbol{\omega}, r, \hat{M}, \mathbf{x}, h) = \phi_1(\boldsymbol{\omega}, r, \hat{M}, \mathbf{x})$. As $\Phi$ is a constant corresponding, we claim continuity of $\phi_2$ and conclude the proof. □

**Theorem 3.** *For any $\epsilon > 0$, $\exists$ a constant $\xi_{1,\epsilon} > 0$, such that if $\left\| \hat{M} - M \right\|_{\infty,\infty} < \xi_{1,\epsilon}$, then for all $M \in S_i$*

$$\left| \max_{\mathbf{x} \in \Delta_K} \min_{\mathbf{h} \in H_{\hat{M}}(\boldsymbol{\omega}, r)} \langle \mathbf{x} - \boldsymbol{\omega}, \mathbf{h} \rangle - \max_{\mathbf{x} \in \Delta_K} \min_{\mathbf{h} \in H_M(\boldsymbol{\omega}, r)} \langle \mathbf{x} - \boldsymbol{\omega}, \mathbf{h} \rangle \right| < \frac{\epsilon}{2}, \forall (\boldsymbol{\omega}, r) \in \Delta_{K\gamma} \times (0, 1) \tag{16}$$

*and*

$$\left| \min_{\mathbf{h} \in H_{\hat{M}}(\boldsymbol{\omega}, r)} \langle \mathbf{x} - \boldsymbol{\omega}, \mathbf{h} \rangle - \min_{\mathbf{h} \in H_M(\boldsymbol{\omega}, r)} \langle \mathbf{x} - \boldsymbol{\omega}, \mathbf{h} \rangle \right| < \frac{\epsilon}{2}, \forall (\boldsymbol{\omega}, r, \mathbf{x}) \in \Delta_{K\gamma} \times (0, 1) \times \Delta_K \tag{17}$$

*Proof.* **Proof of claim** (16). We define for $\boldsymbol{\omega} \in \Delta_{K\gamma}$ and $r \in (0, 1)$, $\psi(\hat{M}) \triangleq \min \left\{ - | \phi_2(\boldsymbol{\omega}, r, \hat{M}) - \phi_2(\boldsymbol{\omega}, r, M) | \right\}$. If we apply Theorem 12 with the following definitions $(X) = S_i$, $(Y) = \Delta_{K\gamma} \times (0, 1)$, $\Phi(\hat{M}) = \Delta_{K\gamma} \times (0, 1)$ and finally $u(\hat{M}, \boldsymbol{\omega}, r) = - | \phi_2(\boldsymbol{\omega}, r, \hat{M}) - \phi_2(\boldsymbol{\omega}, r, M) |$, we can claim that $\psi(\hat{M})$ is continuous on $\Delta_{K\gamma} \times (0, 1)$. So, by definition of $\psi$, $\exists \xi_{1,\epsilon} > 0$, such that $\psi(\hat{M}) > -\frac{\epsilon}{2}$ for all $\left\| \hat{M} - M \right\|_{\infty,\infty} < \xi_{1,\epsilon}$.

Hence, proved.

**Proof of claim** (17). This proof is analogous by leveraging Corollary 2.

$\square$

**Corollary 4.** *For any $\epsilon > 0$, $\exists$ a constant $\xi_{2,\epsilon} > 0$, such that if $\left\| \hat{M} - M \right\|_{\infty,\infty} < \xi_{2,\epsilon}$, then*

$$\hat{M} \in S_i \text{ and } \left| F(\boldsymbol{\omega}|M) - F(\boldsymbol{\omega}|\hat{M}) \right| < \epsilon, \forall \boldsymbol{\omega} \in \Delta_{K\gamma}.$$

*Proof.* We define $(X) = S_i$, $(Y) = \Delta_{K\gamma}$, $\Phi(\hat{M}) = \Delta_{K\gamma}$ and $u(\hat{M}, \boldsymbol{\omega}) = - \mid F(\boldsymbol{\omega} \mid \hat{M}) - F(\boldsymbol{\omega} \mid \hat{M}) \mid$. We again leverage Theorem 12 to claim that $\psi(\hat{M}) \triangleq \min_{\boldsymbol{\omega} \in \Delta_{K\gamma}} - \mid F(\boldsymbol{\omega} \mid \hat{M}) - F(\boldsymbol{\omega} \mid \hat{M}) \mid$ is continuous on the open set $S_i$. Also, $u(\hat{M})$ is continuous due to continuity results in Appendix C. Then, by definition of $\psi$, $\exists \xi_{2,\epsilon} > 0$, such that $\psi(\hat{M}) > -\frac{\epsilon}{2}$ for all $\left\| \hat{M} - M \right\|_{\infty,\infty} < \xi_{2,\epsilon}$. Hence, proved.

$\square$

### D.3 Final Convergence Proof

Due to virtue of Lemma 1 and continuity of $f_{ij}$ in $\Delta_{K\gamma}$ (ref. C), we claim that

**Fact 4.** $F(\boldsymbol{\omega}|M)$ *is L-Lipschitz on $\Delta_{K\gamma}$.*

*Proof.* This fact can be simply proved by applying min-value theorem with guarantees from Lemma 1 to get for $\boldsymbol{\omega}, \boldsymbol{\omega}' \in \Delta_{K\gamma}$

$$F(\boldsymbol{\omega}|M) = \min_{\boldsymbol{\pi}_j \in \text{nbd}\left(\boldsymbol{\pi}_i^\star\right)} f_{ij}(\boldsymbol{\omega}|M) \geq \min_{\boldsymbol{\pi}_j \in \text{nbd}\left(\boldsymbol{\pi}_i^\star\right)} \left(f_{ij}(\boldsymbol{\omega}'|M) - D \left\| \boldsymbol{\omega}' - \boldsymbol{\omega} \right\|_{\infty}\right)$$
$$\min_{\boldsymbol{\pi}_j \in \text{nbd}\left(\boldsymbol{\pi}_i^\star\right)} f_{ij}(\boldsymbol{\omega}'|M) - D \left\| \boldsymbol{\omega}' - \boldsymbol{\omega} \right\|_{\infty} = F(\boldsymbol{\omega}'|M) - D \left\| \boldsymbol{\omega}' - \boldsymbol{\omega} \right\|_{\infty}$$

Hence, proved. Additionally, Lipstchitzness of $F$ can be extended from $\Delta_{K\gamma}$ to $\Delta_K$ due to continuity properties proved in Appendix C.1.

$\square$

**Fact 5.** *Let $\gamma \in (0, \frac{1}{K})$, $\boldsymbol{\omega} \in \Delta_{K\gamma}$ and $\mathbf{x} \in \Delta_K$. Under Lemma 1, we have*

$$f_{ij}(\boldsymbol{\omega}|M) + \langle \mathbf{y} - \boldsymbol{\omega}, \nabla f_{ij}(\boldsymbol{\omega}|M) \rangle - f_{ij}(\mathbf{y}|M) \leq \frac{8D\beta}{\gamma}$$

*where $j \in \text{nbd}\,(i)$ and $\mathbf{y} = \boldsymbol{\omega} + \beta(\mathbf{x} - \boldsymbol{\omega})$ for some $\beta \in (0, \frac{1}{2}]$.*

*Proof.*

$$f_{ij}(\boldsymbol{\omega}|M) + \langle \mathbf{y} - \boldsymbol{\omega}, \nabla f_{ij}(\boldsymbol{\omega}|M) \rangle - f_{ij}(\mathbf{y}|M) \leq 2 \left\| \nabla f_{ij}(\boldsymbol{\omega}|M) \right\|_{\infty} \left\| y - \boldsymbol{\omega} \right\|_1$$
$$\leq \frac{8D\beta}{\gamma}$$

where the last inequality holds due to the definition of $\beta$, $\mathbf{x} \in \Delta_{K\gamma/2}$. Hence, proved.

$\square$

We can extend Fact 5 to get similar result on $F(\cdot|M) = \min_{j \in \text{nbd}(i)} f_{ij}(\cdot \mid M)$ as well.

**Fact 6.** *Let $\gamma \in (0, 1/K)$, $r \in (0, 1)$, $\boldsymbol{\omega} \in \Delta_{K\gamma}$ and $x \in \Delta_K$. Then if $\beta < \min\left\{\frac{1}{2}, \frac{r}{D}\right\}$, then we have*

$$F(\mathbf{y} \mid M) \geq F(\boldsymbol{\omega} \mid M) + \min_{\mathbf{h} \in H_M(\boldsymbol{\omega}, r)} \langle \mathbf{y} - \boldsymbol{\omega}, h \rangle - \frac{8D\beta}{\gamma}$$

*where $y = (1 - \beta)\boldsymbol{\omega} + \beta\mathbf{x}$*

*Proof.* Let there are two different neighbours of $\boldsymbol{\pi}_i^\star$ as $\boldsymbol{\pi}_{j_1}$ and $\boldsymbol{\pi}_{j_2} \in \text{nbd}\,(\boldsymbol{\pi}_i^\star)$ such that $F(\boldsymbol{\omega} \mid M) = f_{ij_1}(\boldsymbol{\omega} \mid M) < f_{ij_2}(\boldsymbol{\omega} \mid M)$ and $F(\mathbf{y} \mid M) = f_{ij_2}(\mathbf{y} \mid M) < f_{ij_1}(\mathbf{y} \mid M)$. Then

$$F(\boldsymbol{\omega} \mid M) + \min_{\mathbf{h} \in H_M(\boldsymbol{\omega},r)} \langle \mathbf{y} - \boldsymbol{\omega}, h \rangle - F(\mathbf{y} \mid M) \leq \frac{8D\beta}{\gamma}$$

The last inequality holds due to Fact 5. Hence, proved. □

**Theorem 4.** *Let* $\eta_t \triangleq F(\boldsymbol{\omega}^\star(M) \mid M) - F(\boldsymbol{\omega}_t \mid M)$. *Also, let* $t \in \mathbb{N}$ *satisfying* $\left\lfloor \sqrt{\frac{t}{K}} \right\rfloor \notin \mathbb{N}$, *then under the good events* $G_1(t) \cap G_2(t)$ *and Frank-Wolfe update step defined in Equation* (14) *and* (15) *with* $tr_t > D$, *we have*

$$\eta_t \leq \frac{t-1}{t}\eta_{t-1} + \frac{r_t - \epsilon}{t} + \frac{16D\sqrt{K}}{t}$$

*Proof.* Using Lemma 9 and Fact 6, we have $\gamma = \frac{1}{2\sqrt{tK}}$ and

$$F(\mathbf{y} \mid M) \geq F(\boldsymbol{\omega} \mid M) + \alpha \left( \max_{\boldsymbol{\omega} \in \Delta_K} \min_{\mathbf{h} \in H_M(\boldsymbol{\omega},r)} \langle \mathbf{x} - \boldsymbol{\omega}, \mathbf{h} \rangle - \epsilon \right) - 16D\beta\sqrt{tK}$$
$$\geq F(\boldsymbol{\omega} \mid M) + \alpha(\eta_{t-1} - r - \epsilon) - 16D\beta\sqrt{tK}$$

The second inequality holds due to properties of $H_M(\boldsymbol{\omega}, r)$. Refer Appendix L.2 of Wang et al. (2021) for further details. Now subtracting $F(\boldsymbol{\omega}^\star(M) \mid M)$ from both sides and putting $\beta = \frac{1}{t^{3/2}}$, we get the desired result. Hence, proved. □

Therefore, for $t > 4K$ the optimality gap becomes very small and asymptotically it converges to zero.

**Theorem 5.** *Let* $\{r_t\}_{t \geq 1}$ *be a sequence of positive numbers with the properties* $\lim_{t \to \infty} \frac{1}{t} \sum_{s=1}^t r_s = 0$ *and* $\lim_{t \to \infty} tr_t = \infty$. *Then for* $T \geq \max \left\{ \left(\frac{35D}{\epsilon}\right)^{11}, T_{\epsilon,D}^{\frac{11}{8}}, (4K+1)^{\frac{11}{8}} \right\}$ *where* $\exists T_{\epsilon,D} \in \mathbb{N}$ *such that if* $t \geq T_{\epsilon,D}$ *then* $\sum_{s=1}^t r_s < \epsilon t$ *and* $tr_t > D$ *for any* $\epsilon \in (0,1)$. *Then under the good events defined in Equation* (18) *and* (19) *we have*

$$F(\boldsymbol{\omega}^\star(M) \mid M) - F(\boldsymbol{\omega}_t) \leq 5\epsilon, \forall t = \bar{h}(T), \bar{h}(T) + 1, \ldots, T,$$

*where* $\bar{h}(T) = \min\{t \in \mathbb{N} : t \geq T^{2/11}\underline{h}(T), \sqrt{\frac{t}{K}} \in \mathbb{N}\}$, *and* $\underline{h}(T) = \min\{t \in \mathbb{N} : t \geq T^{8/11} + 2, \sqrt{\frac{t}{K}} \in \mathbb{N}\}$.

Theorem 5 is a by product of Lemma 3 in Wang et al. (2021) with $L = D$.

# E Sample Complexity of FraPPE

In this section, we first derive the stopping criterion in E.1 that makes FraPPE $(1-\delta)$-correct. Then we move on to the almost sure upper bound on sample complexity of FraPPE in E.2. Then we respectively prove asymptotic and non-asymptotic upper bound guaranties on expected sample complexity in E.3 and E.4.

## E.1 Stopping Criterion

**Theorem 6.** *Given any bandit instance $M \in \mathcal{M}$ and a preference cone $\mathcal{C}$, the Chernoff stopping rule to ensure $(1-\delta)$-correctness in identifying the set of Pareto optimal arms is given by*

$$\min_{\boldsymbol{\pi}^{\star}_{i_t} \in \Pi^{\mathcal{P}_t}} \min_{\boldsymbol{\pi}_j \in \mathrm{nbd}\left(\boldsymbol{\pi}^{\star}_{i_t}\right)} \inf_{\tilde{M} \in \Lambda_{ij}(\hat{M}_t)} \min_{\mathbf{z} \in \bar{\mathcal{C}}^+} \sum_{k=1}^{K} N_{k,t} D_{\mathrm{KL}}\left(\mathbf{z}^{\top} \hat{M}_{k,t} \,\big\|\, \mathbf{z}^{\top} \tilde{M}_k\right) \geq c(t,\delta)$$

*where $c(t,\delta) \triangleq \sum_{k=1}^{K} 3 \ln\left(1 + \ln\left(N_{k,t}\right)\right) + K\mathcal{G}\left(\frac{\ln\left(\frac{1}{\delta}\right)}{K}\right)$.*

*Proof.* We begin this proof in two parts. First, we show that the stopping time $\tau_\delta \in \mathbb{N}$ is finite i.e. $\tau_\delta < \infty$ and then apply concentration on the carefully chosen stochastic process by mixture martingale technique (Kaufmann and Koolen, 2021) to achieve $(1-\delta)$-correctness.

**Finiteness of $\tau_\delta$.** We claim that $\tau_\delta < \infty$, if the parameters in the model converges in finite time. Specifically, for FraPPE , we say it stops when the good events defined in E.3 holds with certainty, and additionally the allocation $\boldsymbol{\omega}_t$ has converged to the optimal allocation $\boldsymbol{\omega}^{\star}(M)$. While finiteness of the first event follows directly from tracking results in Appendix H (For $t > 4K$, each arm has been played enough number of times for $\hat{M}_t$ to $M$; finiteness of the second event is due to convergence guaranties of the Frank-Wolfe iterates proven in Appendix D.

**Stopping threshold.** We define the following stochastic process $X_k(t) \triangleq \sum_{k=1}^{K} \max\{0, \min_{\boldsymbol{\pi}^{\star} \in \{\boldsymbol{\pi}^{\star}_i\}_{i=1}^{p}} \min_{\boldsymbol{\pi}_j \in \mathrm{nbd}(\boldsymbol{\pi}^{\star})} \min_{\mathbf{z} \in \bar{\mathcal{C}}^+} N_{k,t} D_{\mathrm{KL}}\left(\mathbf{z}^{\top} M_k \,\big\|\, \mathbf{z}^{\top} \tilde{M}_k\right) - 3\ln(1 + \ln N_{k,t})\}$. Now as $M_K$ belongs to an exponential family of distributions, $\mathbf{z}^{\top}_{\min} M_k$ also belongs to the exponential distribution with linearly projected scalar mean. Thus, we can directly apply the mixture of martingale technique i.e. Theorem 7 of (Kaufmann and Koolen, 2021) (Refer Theorem 13) with our definition of $X_k(t)$ to conclude the proof.

$\square$

## E.2 Almost Sure Upper Bound on Sample Complexity

**Theorem 7.** *For any $M \in \mathcal{M}$, and $\delta \in (0, 1)$, stopping time of the algorithm FraPPE satisfies*

$$\limsup_{\delta \to 0} \frac{\tau_\delta}{\log(\frac{1}{\delta})} \leq \mathcal{T}_{\mathcal{M},\mathcal{C}} \quad and \quad \tau_\delta < \infty,$$

*almost surely.*

*Proof.* This proof structure closely follows Appendix I of (Wang et al., 2021) with adaptations necessary due to vector valued rewards ordered via given preference cone $\mathcal{C}$.

We start the proof defining the event

$$\Xi \triangleq \{F(\boldsymbol{\omega}_t \mid M) \to F(\boldsymbol{\omega}^{\star}(M)) \text{ as } t \to \infty\}$$

We claim that $\Xi$ is a sure event following Theorem 5 and periodic forced exploration used in FraPPE (So every arm is pulled infinite times, thus we leverage theory of large number). Since we have proved continuity of $F(\boldsymbol{\omega} \mid \hat{M}_t)$ with respect to $M$, we can further claim $F(\boldsymbol{\omega}_t \mid \hat{M}_t) \to F(\boldsymbol{\omega}^{\star}(M) \mid M)$ as $t \to \infty$. We again assume existence of a $t_0 \in \mathbb{N}$ such that for $t > t_0$,

$F(\boldsymbol{\omega}_t \mid \hat{M}_t) \geq (1 - \epsilon)F(\boldsymbol{\omega}^\star(M) \mid M)$ for $\epsilon \in (0,1)$. We also assume existence of constants $c_1(\mathcal{M}), c_2(\mathcal{M}) > 0$ such that if $\forall t \geq c_1(\mathcal{M})$, $\beta(t, \delta) \leq \log\left(\frac{c_2(\mathcal{M})t}{\delta}\right)$. Therefore the stopping time

$$
\begin{aligned}
\tau_\delta &= \inf\{t \in \mathbb{N} \cup \{\infty\} : tF(\boldsymbol{\omega}_t \mid \hat{M}_t) \geq \beta(t, \delta)\} \\
&\leq \inf\{t > t_0 : t(1 - \epsilon)F(\boldsymbol{\omega}^\star(M) \mid M) \geq \beta(t, \delta)\} \\
&\leq \inf\left\{t. \max\{t_0, c_1(\mathcal{M})\} : t \geq \frac{\log(c_2(\mathcal{M})t)}{(1 - \epsilon)\delta}\mathcal{T}_{\mathcal{M},\mathcal{C}}\right\} \\
\implies \tau_\delta &\leq c_1(\mathcal{M}) + t_0 + \frac{\mathcal{T}_{\mathcal{M},\mathcal{C}}}{(1 - \epsilon)\delta}\left[\log\left(\frac{ec_2(\mathcal{M})\mathcal{T}_{\mathcal{M},\mathcal{C}}}{(1 - \epsilon)\delta}\right) + \log\log\left(\frac{c_2(\mathcal{M})\mathcal{T}_{\mathcal{M},\mathcal{C}}}{(1 - \epsilon)\delta}\right)\right]
\end{aligned}
$$

where the last inequality holds due to Lemma 12.

This result ensures asymptotic optimality of FraPPE, i.e, for all $\delta \in (0,1)$, $\limsup_{\delta \to 0} \frac{\tau_\delta}{\log(\frac{1}{\delta})} \leq \mathcal{T}_{\mathcal{M},\mathcal{C}}$ and also $\tau_\delta < \infty$ almost surely. $\qquad\square$

### E.3 Expected Upper Bound on Sample Complexity

**Lemma 2** (Sample Complexity Upper Bound). *For any $M \in \mathcal{M}$, $\epsilon, \tilde{\epsilon} \in (0,1)$ and $\delta \in (0,1)$, and preference cone $\bar{\mathcal{C}}$, expected stopping time satisfies* $\limsup_{\delta \to 0} \frac{\mathbb{E}[\tau_\delta]}{\log(\frac{1}{\delta})} \leq (1 + \tilde{\epsilon})\left(\mathcal{T}_{M,\bar{\mathcal{C}}}^{-1} - 6\epsilon\right)^{-1}$.

*Proof.* **Definition of Good Event.** We define our good events as $G_1(T) \triangleq \bigcap_{t=h(t)}^T G_1(t)$ and $G_2(T) \triangleq \bigcap_{t=h(t)}^T G_2(t)$, where

$$
G_1(t) \triangleq \left\{\max_{\mathbf{x} \in \Delta_K} \min_{\mathbf{h} \in H_{F(\cdot|\hat{M}_t)(\boldsymbol{\omega}_{t-1}, r_t)}} \langle \mathbf{x} - \boldsymbol{\omega}_{t-1}, \mathbf{h}\rangle - \epsilon > \max_{\mathbf{x} \in \Delta_K} \min_{\mathbf{h} \in H_{F(\cdot|M)(\boldsymbol{\omega}_{t-1}, r_t)}} \langle \mathbf{x} - \boldsymbol{\omega}_{t-1}, \mathbf{h}\rangle\right\}
\tag{18}
$$

$$
G_2(t) \triangleq \left\{\hat{M}_t \in S_i \wedge |F(\boldsymbol{\omega}|\hat{M}_t) - F(\boldsymbol{\omega}|M)| < \epsilon, \; \forall \boldsymbol{\omega} \in \Delta_{K\gamma}\right\}
\tag{19}
$$

We start by declaring existence of some constants. Let $\exists T_{\epsilon, D} \in \mathbb{N}$ such that if $t \geq T_{\epsilon, D}$ then $\sum_{s=1}^t r_s < \epsilon t$ and $tr_t > D$ for any $\epsilon \in (0,1)$.

Following the concentration results in Appendix F, we can show that $G_1(t) \cap G_2(t)$ holds true, and thus, by Theorem 5, for $t \geq \bar{h}(T)$ :

$$
F(\boldsymbol{\omega}^\star(M)|M) - F(\boldsymbol{\omega_t}|M) < 5\epsilon.
$$

Here, $\bar{h}(T) = \min\{t \in \mathbb{N} : t \geq T^{2/11}\underline{h}(T), \sqrt{\frac{t}{K}} \in \mathbb{N}\}$, $\underline{h}(T) = \min\{t \in \mathbb{N} : t \geq T^{8/11} + 2, \sqrt{\frac{t}{K}} \in \mathbb{N}\}$, and $T \geq \max\{\left(\frac{35D}{\epsilon}\right)^{11}, T_{\epsilon, D}^{\frac{11}{8}}, (4K + 1)^{\frac{11}{8}}\}$.

Now, given $G_1(t) \cap G_2(t)$ holds true, we have $\min(\tau, T) \leq \bar{h}(T) + \sum_{t=\bar{h}(T)}^T \mathbf{1}\{\tau > T\}$, where $\tau$ is the stopping time of FraPPE. Thus,

$$
\min(\tau, T) \leq \bar{h}(T) + \sum_{t=\bar{h}(T)}^T \mathbf{1}\{tF(\boldsymbol{\omega_t} \mid \hat{M}_t) < c(t, \delta)\}.
$$

Now, for $t \geq \bar{h}(T)$, $F(\boldsymbol{\omega_t} \mid \hat{M}_t) < F(\boldsymbol{\omega_t} \mid M) - \epsilon < F(\boldsymbol{\omega}^\star(M)|M) - 6\epsilon$.

Therefore, we have,

$$\min(\tau, T) \leq \bar{h}(T) + \sum_{t=\bar{h}(T)}^{T} \mathbf{1}\{t(F(\boldsymbol{\omega}^\star(M)|M) - 6\epsilon) < c(t, \delta)\} \leq \bar{h}(T) + \frac{c(T, \delta)}{F(\boldsymbol{\omega}_t{}^\star(M)|M) - 6\epsilon}.$$

Finally, we define a constant

$$T_0(\delta) \triangleq \inf\{T \in \mathbb{N} : \bar{h}(T) + \frac{c(T, \delta)}{F(\boldsymbol{\omega}_t{}^\star(M)|M) - 6\epsilon} \leq T\}.$$

Now, we introduce a small constant $\tilde{\epsilon} \in (0, 1)$, such that $T - \bar{h}(T) \geq \frac{T}{1+\tilde{\epsilon}}$, when $T \geq \left(\frac{2}{\tilde{\epsilon}}\right)^{11}$ This choice of $\tilde{\epsilon}$ is reflected in non-asymptotic sample complexity upper bound (Lemma 6 in Appendix E.3).

Now, following the algebra in (Wang et al., 2021), we get

$$\mathbb{E}[\tau_\delta] \leq \left(\frac{35D}{\epsilon}\right)^{11} + T_{\epsilon,D}^{\frac{11}{8}} + (4K+1)^{\frac{11}{8}} + T_0(\delta) + \sum_{T=N+1}^{\infty} \mathbb{P}\left((G_1(T) \cap G_2(T))^c\right) \quad (20)$$

where $N = \max\left\{\left(\frac{35D}{\epsilon}\right)^{11} + T_{\epsilon,D}^{11}, T_{\epsilon,D}^{\frac{11}{8}}, (4K+1)^{\frac{11}{8}}\right\}$ and $T_0(\delta)$ satisfies the following inequality

$$T_0(\delta) \leq \max\left\{\left(\frac{2}{\tilde{\epsilon}}\right)^{11}, c_1(\mathcal{M})\right\}$$
$$+ \frac{1+\tilde{\epsilon}}{F(\boldsymbol{\omega}^\star(M) \mid M) - 6\epsilon}\left[\log\left(\frac{1}{\delta}\right) + \log\log\left(\frac{1}{\delta}\right)\right.$$
$$+ \log\left(\frac{(1+\tilde{\epsilon})c_2(\mathcal{M})e}{(F(\boldsymbol{\omega}^\star(M) \mid M) - 6\epsilon)}\right)$$
$$\left.+ \log\log\left(\frac{(1+\tilde{\epsilon})c_2(\mathcal{M})}{(F(\boldsymbol{\omega}^\star(M) \mid M) - 6\epsilon)}\right)\right]$$

Thus, the asymptotic sample complexity is then given by,

$$\limsup_{\delta \to 0} \frac{\mathbb{E}[\tau_\delta]}{\log(\frac{1}{\delta})} \leq \frac{1+\tilde{\epsilon}}{F(\boldsymbol{\omega}^\star(M) \mid M) - 6\epsilon}$$

Optimality follows if $\epsilon$ and $\tilde{\epsilon}$ are arbitrarily small, that is $\limsup_{\delta \to 0} \frac{\mathbb{E}[\tau_\delta]}{\log(\frac{1}{\delta})} \leq \frac{1}{F(\boldsymbol{\omega}^\star(M)|M)} = \mathcal{T}_{\mathcal{M},\mathcal{C}}$. Hence, proved. $\qquad\square$

### E.4 Non-Asymptotic Sample Complexity Upper Bound

**Lemma 6.** *For any $M \in \mathcal{M}$, $\epsilon, \tilde{\epsilon} \in (0, 1)$ and $\delta \in (0, 1)$, and given preference cone $\mathcal{C}$ stopping time of the algorithm* FraPPE *satisfies*

$$\mathbb{E}[\tau_\delta] \leq 34e^K \sum_{k=1}^{K} \frac{1}{D_{\mathrm{KL}}\left(\mathbf{z}_{\max}^\top M_k - \frac{\epsilon}{2D} \,\middle\|\, \mathbf{z}_{\max}^\top M_k\right)^{19/4}} + \frac{1}{D_{\mathrm{KL}}\left(\mathbf{z}_{\max}^\top M_k + \frac{\epsilon}{2D} \,\middle\|\, \mathbf{z}_{\max}^\top M_k\right)^{19/4}}$$
$$+ \left(\frac{35D}{\epsilon}\right)^{11} + (4K+1)^{\frac{11}{8}} + \max\left\{\left(\frac{2}{\tilde{\epsilon}}\right)^{11}, c_1(\mathcal{M})\right\}$$
$$+ \frac{1+\tilde{\epsilon}}{F(\boldsymbol{\omega}^\star(M) \mid M) - 6\epsilon}\left[\log\left(\frac{(1+\tilde{\epsilon})c_2(\mathcal{M})e}{\delta(F(\boldsymbol{\omega}^\star(M) \mid M) - 6\epsilon)}\right) + \log\log\left(\frac{(1+\tilde{\epsilon})c_2(\mathcal{M})}{\delta(F(\boldsymbol{\omega}^\star(M) \mid M) - 6\epsilon)}\right)\right]$$

*Proof.* We combine Equation (20) and Lemma 7 to write the expression of expected sample complexity and conclude the proof.

$\qquad\square$

# F Concentration Result

**Lemma 7.** *Under the good events defined in* (18) *and* (19),

$$\mathbb{P}\left((G_1(T) \cap G_2(T))^c\right) \leq \infty$$

*Proof.* We first remind the definition of the good events

$$G_1(t) \triangleq \left\{ \max_{\mathbf{x} \in \Delta_K} \min_{\mathbf{h} \in H_{F(.|\hat{M}_t)(\boldsymbol{\omega}_{t-1}, r_t)}} \langle \mathbf{x} - \boldsymbol{\omega}_{t-1}, \mathbf{h} \rangle - \epsilon > \max_{\mathbf{x} \in \Delta_K} \min_{\mathbf{h} \in H_{F(.|M)(\boldsymbol{\omega}_{t-1}, r_t)}} \langle \mathbf{x} - \boldsymbol{\omega}_{t-1}, \mathbf{h} \rangle \right\}$$

$$G_2(t) \triangleq \left\{ \hat{M}_t \in S_i \wedge |F(\boldsymbol{\omega}|\hat{M}_t) - F(\boldsymbol{\omega}|M)| < \epsilon, \ \forall \boldsymbol{\omega} \in \Delta_{K\gamma} \right\}$$

We have $G_1(t) \subset \{\|M - \hat{M}_{t-1}\|_{\infty,\infty} \leq \xi_{1,\epsilon}\}$ and also $G_2(t) \subset \{\|M - \hat{M}_t\|_{\infty,\infty} \leq \xi_{2,\epsilon}\}$, if we apply Theorem 3 and Corollary 4 respectively.

Then we have

$$\mathbb{P}\left((G_1(T) \cap G_2(T))^c\right) \leq \sum_{h(T)}^{T} \mathbb{P}(\|M - \hat{M}_t\|_{\infty,\infty} > \xi_\epsilon)$$

where $\xi_\epsilon = \max(\xi_{1,\epsilon}, \xi_{2,\epsilon})$. Now $\|M - \hat{M}_t\|_{\infty,\infty} = \max_{z \in \bar{\mathcal{C}}^+} \max_{k \in [K]} \mathbf{z}^\top (M - \hat{M}_t)$. Therefore

$$\mathbb{P}\left((G_1(T) \cap G_2(T))^c\right) \leq \sum_{h(t)}^{T} \mathbb{P}(\max_{z \in \bar{\mathcal{C}}^+} \max_{k \in [K]} |\mathbf{z}^\top (M - \hat{M}_t)| > \xi_\epsilon)$$

$$\leq \sum_{h(t)}^{T} \sum_{k=1}^{K} \mathbb{P}(\max_{z \in \bar{\mathcal{C}}^+} |\mathbf{z}^\top (M_k - \hat{M}_{k,t})| > \xi_\epsilon)$$

From tracking results (Lemma 9) we have at any time $t > 4K$ and $\forall k \in [K]$, $N_{t,k} \geq \sqrt{\frac{t}{K}} - K$, ensuring that each arm, is played enough till time $t \in [T]$ to bring $\hat{M}_{k,t}$ close to $M_k$. Now we choose $\mathbf{z}_{\max} \triangleq \arg\max_{z \in \bar{\mathcal{C}}^+} |\mathbf{z}^\top M_k - \hat{M}_{k,t}|$. Therefore

$$\mathbb{P}\left((G_1(T) \cap G_2(T))^c\right) \leq \sum_{h(t)}^{T} \sum_{k=1}^{K} \mathbb{P}\left(|\mathbf{z}_{\max}^\top (M_k - \hat{M}_{k,t})| > \xi_\epsilon\right) \tag{21}$$

Then applying Chernoff inequality on the probability on the R.H.S of the inequality (21), we get

$$\mathbb{P}\left(|\mathbf{z}^\top (M_k - \hat{M}_{k,t})| > \xi_\epsilon\right) \leq e^K \left[\exp(-\sqrt{t}A_k^-) + \exp(-\sqrt{t}A_k^+)\right]$$

where $A_k^- = \frac{D_{\mathrm{KL}}\left(\mathbf{z}_{\max}^\top M_k - \xi_\epsilon \parallel \mathbf{z}_{\max}^\top M_k\right)}{\sqrt{K}}$ and $A_k^+ = \frac{D_{\mathrm{KL}}\left(\mathbf{z}_{\max}^\top M_k + \xi_\epsilon \parallel \mathbf{z}_{\max}^\top M_k\right)}{\sqrt{K}}$. We leverage this upper bound the bad event to get

$$\mathbb{P}\left((G_1(T) \cap G_2(T))^c\right) \leq e^K \sum_{h(t)}^{T} \sum_{k=1}^{K} \left[\exp(-\sqrt{t}A_k^-) + \exp(-\sqrt{t}A_k^+)\right] < \infty$$

$$\implies \sum_{T=N}^{\infty} \mathbb{P}\left((G_1(T) \cap G_2(T))^c\right) < 34 e^K \sum_{k=1}^{K} \frac{1}{D_{\mathrm{KL}}\left(\mathbf{z}_{\max}^\top M_k - \xi_\epsilon \parallel \mathbf{z}_{\max}^\top M_k\right)^{19/4}}$$

$$+ \frac{1}{D_{\mathrm{KL}}\left(\mathbf{z}_{\max}^\top M_k + \xi_\epsilon \parallel \mathbf{z}_{\max}^\top M_k\right)^{19/4}} \tag{22}$$

Where the last line is implied by leveraging Lemma 13 with $\alpha = \frac{8}{11}$, $\beta = \frac{1}{2}$ and $A = A_k^+ = A_k^-$ respectively. $\qquad \square$

Now using Theorem 8 and 9 of Wang et al. (2021), we also get stricter versions of Theorem 3 and Corollary 4 as

**Corollary 5.** *(Stricter version of Corollary 4) For any $M \in \mathcal{M}$ and $\epsilon < (0, \kappa D)$ where $\kappa \triangleq \inf_{\boldsymbol{\pi}_i} \inf_{\mathbf{z} \in \bar{\mathcal{C}}^+} \inf_{\boldsymbol{\pi}_j \in \mathrm{nbd}(\boldsymbol{\pi}_i)} \mathbf{z}^\top M(\boldsymbol{\pi}_i - \boldsymbol{\pi}_j)$, then if $\|M - \hat{M}\|_{\infty,\infty} \leq \frac{\epsilon}{2D}$, then*

$$|F(\boldsymbol{\omega}|M) - F(\boldsymbol{\omega}|\hat{M})| < \epsilon, \quad \forall \boldsymbol{\omega} \in \Delta_{K\gamma}$$

**Theorem 8.** *(Stricter version of Theorem 3) Let $M \in \mathcal{M}$ and $\epsilon \in (0, \kappa D)$. For any $r \in (0,1)$, and $\boldsymbol{\omega} \in \Delta_{K\gamma}$, if $\hat{M} \in S_i$ that is $\|M - \hat{M}\|_{\infty,\infty} \leq \frac{\epsilon}{2D}$, then we get*

$$\left| \max_{\mathbf{x} \in \Delta_K} \min_{\mathbf{h} \in H_{\hat{M}}(\boldsymbol{\omega},r)} \langle \mathbf{x} - \boldsymbol{\omega}, \mathbf{h} \rangle - \max_{\mathbf{x} \in \Delta_K} \min_{\mathbf{h} \in H_M(\boldsymbol{\omega},r)} \langle \mathbf{x} - \boldsymbol{\omega}, \mathbf{h} \rangle \right| < \frac{\epsilon}{2}, \forall (\boldsymbol{\omega}, r) \in \Delta_{K\gamma} \times (0,1)$$

*and*

$$\left| \min_{\mathbf{h} \in H_{\hat{M}}(\boldsymbol{\omega},r)} \langle \mathbf{x} - \boldsymbol{\omega}, \mathbf{h} \rangle - \min_{\mathbf{h} \in H_M(\boldsymbol{\omega},r)} \langle \mathbf{x} - \boldsymbol{\omega}, \mathbf{h} \rangle \right| < \frac{\epsilon}{2}, \forall (\boldsymbol{\omega}, r, \mathbf{x}) \in \Delta_{K\gamma} \times (0,1) \times \Delta_K$$

Therefore putting $\xi_\epsilon = \frac{\kappa}{2D}$ in Equation (22) to get the final non-asymptotic bound on the bad event probabilities as

$$\mathbb{P}\left( (G_1(T) \cap G_2(T))^c \right) \leq 34 e^K \sum_{k=1}^{K} \frac{1}{D_{\mathrm{KL}}\left( \mathbf{z}_{\max}^\top M_k - \frac{\epsilon}{2D} \,\|\, \mathbf{z}_{\max}^\top M_k \right)^{19/4}}$$
$$+ \frac{1}{D_{\mathrm{KL}}\left( \mathbf{z}_{\max}^\top M_k - \frac{\epsilon}{2D} \,\|\, \mathbf{z}_{\max}^\top M_k \right) 19/4}$$

# G  Algorithmic Complexity

## G.1  Time Complexity

**1. Pareto Set Estimation.** Our proposed algorithm FraPPE first computes the set of Pareto arms based on current estimates of the means. We refer to Kone et al. (2024) to state that worst-case complexity for estimating Pareto set is given by $\mathcal{O}\left(K\log(K)^{\max\{1,L-2\}}\right)$.

**2.** Now for each candidate pure Pareto policies and its neighbour,

**(i) Frank-Wolfe Step.** FraPPE again solves a linear optimisation as a part of Frank-Wolfe update step in the sub-differential set. So, it also suffers complexity of $\mathcal{O}\left(\frac{1}{\text{tol}}\right)$ due to $K$-independent upper bound on curvature constant.

**(ii) Stopping Criteria.** To calculate the GLRT metric FraPPE solves another linear optimisation with complexity $\mathcal{O}\left(\frac{L}{\text{tol}}\right)$, since $\mathbf{z}$ and $\mathbf{y}$ respectively come from a convex preference cone and its polar form.

Therefore Step 2 suffers from a total time complexity of $\mathcal{O}\left(\frac{K\min\{K,L\}}{\text{tol}}\right)$ since in the worst case, there can be $K$ Pareto candidates and there can be $\min\{K,L\}$ neighbours.

**3. Parameter Update.** Updating $\hat{M}_{t+1}$ enjoys complexity of order $\mathcal{O}(L)$.

**Total Time Complexity.** Then combining Step 1, 2 and 3 we get the total worst-case time complexity per iteration of FraPPE as $\mathcal{O}\left(K\log(K)^{\max\{1,L-2\}}\right) + \mathcal{O}\left(\frac{KL\min\{K,L\}}{\text{tol}}\right) + \mathcal{O}(L) = \mathcal{O}\left(\max\left\{K\log(K)^{\max\{1,L-2\}}, K\min\{K,L\}\right\}\right)$.

In most real-life instances, the number of available options are greater than objectives, i.e $K > L$. In that case FraPPE enjoys time complexity of $\mathcal{O}\left(KL^2\right)$, which is a significant improvement over existing literature. Notably, the complexity shows that the complexity for calculating the Pareto front dominates the overall complexity when $L \geq 5$ and $K \geq 19$.

## G.2  Space Complexity

We compute the space complexity of FraPPE in three steps,

**1. Mean estimates.** We need to maintain the estimate of $M$ of dimension $K \times L$ per step, so space complexity is of $\mathcal{O}(KL)$.

**2. Estimated Pareto Set.** This is of $\mathcal{O}(K)$ in the worst case.

**3. Neighbour set.** The space for maintaining the neighbour set is of $\mathcal{O}(KL)$.

**Total Sample Complexity.** Therefore the total space complexity of FraPPE is given by $\mathcal{O}(KL)$.

# H  Tracking Results

We leverage similar tracking results as Wang et al. (2021). We reiterate the Tracking Lemmas again with notations used in this paper.

**Lemma 8.** *Let $\{x_s\}_{s\in\mathbb{N}} \in \Delta_K$ be a sequence of vectors such that $x_k$ is $e_k$ for $k = [1, K]$. Then*

$$t = K, N_{k,t} = 1,$$

$$\forall t \geq K + 1, A_t \in \arg\max_{k'} \frac{\sum_{s=1}^{t} x_{k',s}}{N_{k',t-1}} \forall k \in [1, K], \text{ where } N_{k,t} = \sum_{s=1}^{t} \mathbb{1}\{A_s = k\}$$

*where the* $\arg\max$ *breaks ties arbitrarily. Then for all $t \geq K$, and all $k \in [1, K]$*

$$\sum_{s=1}^{t} x_{k,s} - (K - 1) \leq N_{k,t} \leq \sum_{s=1}^{t} x_{k,s} + 1$$

**Lemma 9.** *At any $t \geq 4K$, the sampling rule of* FraPPE *satisfies* $\boldsymbol{\omega}_s \in \Delta_{\frac{1}{2\sqrt{tK}}}$

*Proof.* By forced exploration enforced in FraPPE, we can write

$$t\boldsymbol{\omega}_t = \sum_{s=1}^{t} x_s \geq \frac{1}{K}\mathbb{1}\{x_s = (1/K, 1/K, \ldots, 1/K)\}$$

$$= \sqrt{\left\lfloor \frac{t}{K} \right\rfloor} \geq \sqrt{\frac{t}{K}} - 1 \geq \frac{1}{2}\sqrt{\frac{t}{K}}$$

$$\implies \boldsymbol{\omega}_t \geq \frac{1}{2\sqrt{tK}}$$

$\square$

# I Additional Experimental Details

## I.1 Implementation Details

**Computational Resource.** We run all the algorithms on a 64-bit 13th Gen Intel octa-Core i7-1370P $\times$ 20 processor machine with 32GB RAM.

**Dataset Description: Cov-Boost.** This real-life inspired data set contains tabulated entries of phase-2 booster trial for Covid-19 (Munro et al., 2021). Cov-Boost has been used as a benchmark dataset for evaluating algorithms for Pareto Set Identification (PSI). It consists bandit instance with 20 vaccines, i.e. arms, and 3 immune responses as objectives, i.e., $K = 20$ and $L = 3$. Additionally, Kone et al. (2024) provides detailed description of this dataset in Table 3 (empirical arithmetic mean of the log-transformed immune response for three immunogenicity indicators) and 4 (pooled variances). For entirety, we include the mean and pooled variance table here

Table 3: Mean matrix

| Dose 1/Dose 2 | Dose 3 (booster) | Immune response | | |
| --- | --- | --- | --- | --- |
| | | Anti-spike IgG | $NT_{50}$ | cellular response |
| Prime BNT/BNT | ChAd | 9.50 | 6.86 | 4.56 |
| | NVX | 9.29 | 6.64 | 4.04 |
| | NVX Half | 9.05 | 6.41 | 3.56 |
| | BNT | 10.21 | 7.49 | 4.43 |
| | BNT Half | 10.05 | 7.20 | 4.36 |
| | VLA | 8.34 | 5.67 | 3.51 |
| | VLA Half | 8.22 | 5.46 | 3.64 |
| | Ad26 | 9.75 | 7.27 | 4.71 |
| | m1273 | 10.43 | 7.61 | 4.72 |
| | CVn | 8.94 | 6.19 | 3.84 |
| Prime ChAd/ChAd | ChAd | 7.81 | 5.26 | 3.97 |
| | NVX | 8.85 | 6.59 | 4.73 |
| | NVX Half | 8.44 | 6.15 | 4.59 |
| | BNT | 9.93 | 7.39 | 4.75 |
| | BNT Half | 8.71 | 7.20 | 4.91 |
| | VLA | 7.51 | 5.31 | 3.96 |
| | VLA Half | 7.27 | 4.99 | 4.02 |
| | Ad26 | 8.62 | 6.33 | 4.66 |
| | m1273 | 10.35 | 7.77 | 5.00 |
| | CVn | 8.29 | 5.92 | 3.87 |

Table 4: Pooled variances

| | Immune response | | |
| --- | --- | --- | --- |
| | Anti-spike IgG | $NT_{50}$ | cellular response |
| Pooled sample variance | 0.70 | 0.83 | 1.54 |

**Code.** The anonymous repository link for the implementations - code repository.

## I.2 A Pseudocode of Accelerated PreTS

The algorithm PreTS provided in Shukla and Basu (2024) lacks tractability as discussed before. The allocation computation involved constructing a convex hull over the Alt-set, which is very expensive and makes it non-implementable. So, here we provide an *accelerated* version of PreTS (Algorithm 2) that can be implemented leveraging our observations of the optimisation problem. Note that, this version of PreTS is still expensive to run. Roughly our Algorithm 2 runs 90 times faster than Algorithm 2 when implemented in Python3 with identical resource environment.

---

**Algorithm 2 Accelerated Pre**ference-based **T**rack-and-**S**top (Accelerated PreTS)

---

1: **Input:** Confidence parameter $\delta$, preference cone $\bar{\mathcal{C}}$
2: **Initialise:** For $t \in [K]$, sample each arm once s.t. $\boldsymbol{\omega}_K = (1/K, \cdots, 1/K)$, mean estimate $\hat{M}_K$
3: **while** Equation (8) is **FALSE do**
4:     **Estimate Pareto Indices:** Calculate Pareto indices $\mathcal{P}_t$ based on current estimate $\hat{M}_t$.
5:     **Estimate Set of Pareto Policies:** $\Pi^{\mathcal{P}_t}$ consisting pure policies with $i_t \in \mathcal{P}_t$ as basis.
6:     **Set of Neighbours:** $\Pi \setminus \Pi^{\mathcal{P}_t}$, where $\Pi$ is the set of all pure policies.
7:     **Allocation:** $\boldsymbol{\omega}_t \leftarrow \underset{\boldsymbol{\omega} \in \Delta_K}{\arg\max} \underset{\substack{\boldsymbol{\pi}_j \in \mathrm{nbd}(\boldsymbol{\pi}_i^\star) \\ \boldsymbol{\pi}_i^\star \in \{\boldsymbol{\pi}_i^\star\}_{i=1}^p}}{\min} \underset{\tilde{M} \in \bar{\Lambda}_{ij}(M)}{\min} \underset{\mathbf{z} \in \bar{\mathcal{C}}^+}{\min} \sum_{k=1}^K \boldsymbol{\omega}_k D_{\mathrm{KL}} \left( \mathbf{z}^\top \hat{M}_{k,t} \,\middle\|\, \mathbf{z}^\top \tilde{M}_k \right)$
8:     **C-tracking:** Play $A_t \in \arg\min N_{a,t} - \sum_{s=1}^{t+1} \boldsymbol{\omega}_s$ (ties broken arbitrarily)
9:     **Feedback and Parameter Update:** Get feedback $R_t \in \mathbb{R}^L$ and update $\hat{M}_t$ to $\hat{M}_{t+1}$ with $R_t$
10: **end while**
11: **Recommendation Rule:** Recommend $\mathcal{P}_t$ as the Pareto optimal set

---

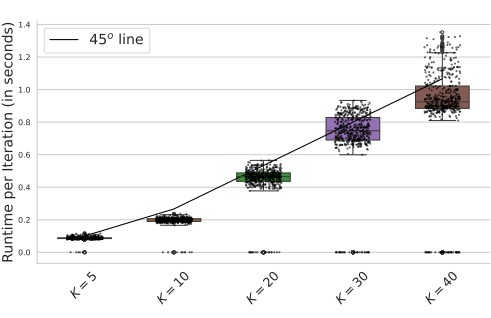 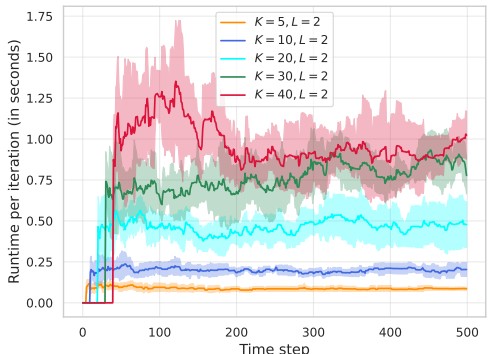

Figure 6: Runtime per iteration (in seconds) of FraPPE for varying number of arms $K$ in the Gaussian instance with $L = 2$ and correlation coefficient $\rho = 0.9$.

## J  Runtime Analysis of FraPPE

In this section, we provide runtime analysis of FraPPE with varying values of $K$ and $L$.

**I. Scaling with $K$.** We consider synthetic environments consisting of $L = 2$, multivariate Gaussian rewards, and positive right orthant as the preference cone. Additionally, we assume the two objectives to be highly positively correlated with correlation coefficient $\rho = 0.9$. We set $\delta = 0.01$. Finally, we vary the number of arms $K = 5, 10, 20, 30, 40$, keeping the Pareto front same and report the corresponding clock time per iteration for each of the instances. The final results are averaged over 10 experiments.

**Observation.** From Figure 6, the trend in runtime is sub-linear with respect to $K$, i.e. the clock runtime complexity scales with at most $\mathcal{O}(K)$.

**I. Scaling with $L$.** To test the runtime of FraPPE with varying values of $L$, we fix $K = 25$. Keeping all other parameters and Pareto front same, we set four environments with $L = 2, 6, 10, 14$ to report the runtimes per iterations. The final results are averaged over 10 experiments.

**Observations:** It is clear from Figure 7 that both mean and median runtime of FraPPE falls below the rate $\mathcal{O}\big((\log(K)^{\max\{1, L-2\}}\big)$ after $L \geq 5$. That means complexity of calculating Pareto front dominates the complexity of lower bound optimisation.

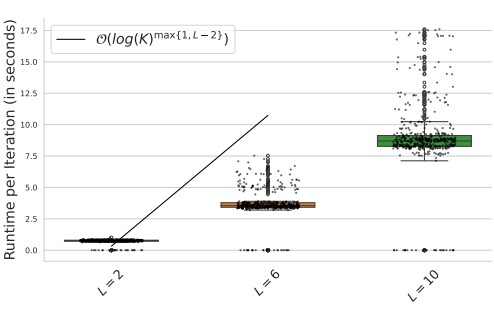 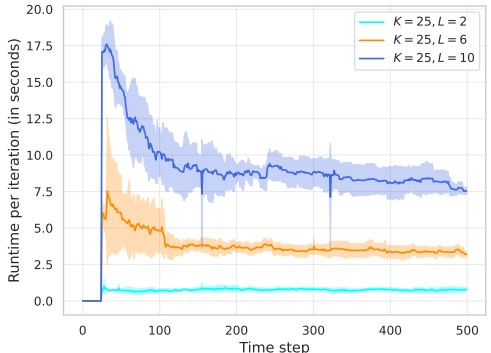

Figure 7: Runtime per iteration (in seconds) of FraPPE for varying number of arms $L$ in the Gaussian instance with $K = 25$ and correlation coefficient $\rho = 0.9$.

# K    Auxiliary Definitions and Results

**Theorem 9** (Berge's Maximum Theorem)**.** *Let $\mathcal{U}$ and $\mathcal{V}$ be topological spaces, $f : \mathcal{U} \times \mathcal{V} \to \mathbb{R}$ and $C : \mathcal{U} \to \mathcal{V}$ be non-empty compact set for all $u \in \mathcal{U}$. Then, if $C$ is continuous at $u$, $f^*(u) = \max_{v \in C(u)} f(u, v)$ is continuous and $C^*(u) = \{v \in C(u) : f^*(u) = f(u, v)\}$ is upper-hemicontinuous.*

**Theorem 10** (Donsker-Vardhan Variational Formula)**.** *For mutual information $KL(P \| Q)$, we have that:*

$$d_{\mathrm{KL}}(P \| Q) = \sup_f \mathbb{E}_P[f] - \ln \mathbb{E}_Q[\exp(f)]$$

**Lemma 10** (Peskun Ordering)**.** *For any two random variables $X, Y$ on $\mathbb{R}^{KL}$ the following are equivalent:*

1. *$X \leq_s Y$*

2. *For all $x \in \mathbb{R}^{KL}$, $\mathbb{P}[X \geq x] \leq \mathbb{P}[Y \geq x]$*

3. *For all non-negative functions $f_1, f_2, \ldots, f_k$, we have that: $\Pi_{i=1}^K f_i \leq \Pi_{i=1}^K f_i$*

**Theorem 11** (Berge (1877))**.** *Let $\mathbb{X}$ and $\mathbb{Y}$ be Hausdorff topological spaces. Assume that*

1. *$\Phi : \mathbb{X} \rightrightarrows \mathbb{K}(\mathbb{X})$ is continuous, where $\mathbb{K}(\mathbb{X}) = \{F \in \mathbb{S}(\mathbb{X}) : F$ is compact.$\}$ (i.e. both upper and lower hemiconituous),*

2. *$u : \mathbb{X} \times \mathbb{Y} \to \mathbb{R}$ is continuous.*

*Then the function $v : \mathbb{X} \to \mathbb{R}$ is continuous and the solution multifunction $\Phi^\star : \mathbb{X} \to \mathbb{S}(\mathbb{Y})$ is upper hemicontinuous and compact valued, where $\mathbb{S}(\mathbb{Y})$ is the set of non-empty subsets of $\mathbb{Y}$.*

**Theorem 12** (Feinberg et al. (2014))**.** *Assume that*

1. *$\mathbb{X}$ is compactly generated,*

2. *$\Phi : \mathbb{X} \rightrightarrows \mathbb{S}(\mathbb{Y})$ is lower hemicontinuous,*

3. *$u : \mathbb{X} \times \mathbb{Y} \to \mathbb{R}$ is $\mathbb{K}$-inf-compact and upper semi-continuous on $\mathrm{Gr}_{\mathbb{X}}(\Phi)$.*

*Then the function $v : \mathbb{X} \to \mathbb{R}$ is continuous and the solution multifunction $\Phi^* : \mathbb{X} \rightrightarrows \mathbb{S}(\mathbb{Y})$ is upper hemicontinuous and compact valued.*

**Lemma 11** (Combes et al. (2017))**.** *Let $\mathbb{X}$ be a metric space and $Y$ be a nonempty open subset in $\mathbb{R}^K$. Let $u : \mathbb{X} \times Y \to \mathbb{R}$ and $\frac{\partial u}{\partial y}$ exists and is continuous in $\mathbb{X} \times Y$. For each $y \in Y$, let $x^\star(y)$ minimises $u(x, y)$ over $x \in \mathbb{X}$. Set*

$$v(y) = u(x^\star(y), y).$$

*Assume that $x^\star : Y \to \mathbb{X}$ is a continuous function. Then $v$ is continuously differentiable and*

$$\frac{d}{dy} v(y) = \frac{\partial u}{\partial y}(x^\star(y), y).$$

**Fact 7** (Existence and Continuity of Minimum)**.** *If $X$ and $Y$ are topological spaces and $Y$ is compact. Then for any continuous $f : X \times Y \to \mathbb{R}$, the function $g(x) \triangleq \inf_{y \in Y} f(x, y)$ is well-defined and continuous. Additionally, $\inf_{y \in Y} f(x, y) = \min_{y \in Y} f(x, y)$.*

**Fact 8** (inf and min over Union of Sets)**.** *Let us consider an ordered universe $S$ and a set $A \subset S$ which is union of $|I|$ sets, i.e. $A = \cup_{i \in I} A_i$ and $A_i \subset S$. If the following statements are true:*

1. *$a \triangleq \inf A$ exists, and*

2. *$a_i \triangleq \inf A_i$ exists for each $i \in I$,*

*they imply that $a = \inf\{a_i : i \in I\}$. The same holds true for $\min$.*

**Theorem 13** (Version of Theorem 7 in Kaufmann and Koolen (2021))**.** *Let $\delta > 0, \nu$ be independent one-parameter exponential families with mean $\boldsymbol{\mu}$ and $S \subset [K]$. Then we have,*

$$\mathbb{P}_\nu \left[ \exists t \in \mathbb{N} : \sum_{a \in S} \tilde{N}_{t,a} d_{KL} \left( \mu_{t,a}, \mu_a \right) \geq \sum_{a \in S} 3 \ln \left( 1 + \ln \left( \tilde{N}_{t,a} \right) \right) + |S| T \left( \frac{\ln \left( \frac{1}{\delta} \right)}{|S|} \right) \right] \leq \delta \, .$$

*Here, $\mathcal{G} : \mathbb{R}^+ \to \mathbb{R}^+$ is such that $\mathcal{G}(x) = 2\tilde{h}_{3/2} \left( \frac{h^{-1}(1+x) + \ln \left( \frac{\pi^2}{3} \right)}{2} \right)$ with*

$$\forall u \geq 1, \quad h(u) = u - \ln(u)$$

$$\forall z \in [1, e], \forall x \geq 0, \quad \tilde{h}_z(x) = \begin{cases} \exp \left( \frac{1}{h^{-1}(x)} \right) h^{-1}(x) & \text{if } x \geq h^{-1} \left( \frac{1}{\ln(z)} \right) \\ z(x - \ln(\ln(z))) & \text{else} \end{cases} .$$

**Theorem 14** (Grinberg (2017))**.** *For a subset $S$ in Euclidean space $\mathbb{R}^n$, the statements $S$ is compact, i.e every open cover of $S$ has a finite sub-cover $\Longleftrightarrow S$ is closed and bounded.*

**Lemma 12** (Lemma 18 in (Garivier and Kaufmann, 2016))**.** *For $\alpha \in [1, e/2]$, any two constants $c_1, c_2$,*

$$x = \frac{1}{c_1} \left[ \log \left( \frac{c_2 e}{c_1^\alpha} \right) + \log \log \left( \frac{c_2}{c_1^\alpha} \right) \right]$$

*is such that $c_1 x \geq \log \left( c_2 x^\alpha \right)$.*

**Lemma 13** (Wang et al. (2021))**.** *Let $\alpha, \beta \in (0, 1)$ and $A > 0$.*

$$\int_0^\infty \left( \int_{T^\alpha}^\infty \exp \left( -At^\beta \right) dt \right) dT = \frac{\Gamma \left( \frac{1}{\alpha \beta} + \frac{1}{\beta} \right)}{\beta A^{\frac{1}{\alpha \beta} + \frac{1}{\beta}}}$$

**Example 1.** *Multivariate Gaussian with mean $\boldsymbol{\mu}$ and covariance $\boldsymbol{\Sigma}$*

*The density function is*

$$f(\mathbf{x}) = (2\pi)^{-\frac{L}{2}} \det(\boldsymbol{\Sigma})^{-\frac{1}{2}} \exp \left( -\frac{1}{2}(\mathbf{x} - \boldsymbol{\mu})^\top \boldsymbol{\Sigma}^{-1}(\mathbf{x} - \boldsymbol{\mu}) \right) \, .$$

*We can rewrite it as*

$$f(\mathbf{x}) = (2\pi)^{-\frac{L}{2}} \exp \left( \langle \boldsymbol{\Sigma}^{-1} \boldsymbol{\mu}, \mathbf{x} \rangle + \left\langle \text{vec} \left( -\frac{1}{2} \boldsymbol{\Sigma}^{-1} \right), \text{vec} \left( \mathbf{x}\mathbf{x}^\top \right) \right\rangle - \frac{1}{2} \left[ \log |\boldsymbol{\Sigma}| + \boldsymbol{\mu}^\top \boldsymbol{\Sigma}^{-1} \boldsymbol{\mu} \right] \right) \, .$$

*Thus, the base density is $h(\boldsymbol{X}) = (2\pi)^{-\frac{L}{2}}$. The natural parameter is $\eta(\boldsymbol{\theta}) = \left( \begin{smallmatrix} \boldsymbol{\Sigma}^{-1} \boldsymbol{\mu} \\ \text{vec}\left(-\frac{1}{2} \boldsymbol{\Sigma}^{-1}\right) \end{smallmatrix} \right)$. The sufficient statistic is $T(\boldsymbol{x}) = \left( \begin{smallmatrix} \mathbf{x} \\ \text{vec}(\mathbf{x}\mathbf{x}^\top) \end{smallmatrix} \right)$. The log-normaliser or log-partition function is $\psi(\boldsymbol{\theta}) = \frac{1}{2} \left[ \log \det(\boldsymbol{\Sigma}) + \boldsymbol{\mu}^\top \boldsymbol{\Sigma}^{-1} \boldsymbol{\mu} \right]$.*

