# OpenReview forum: "FraPPE: Fast and Efficient Preference-Based Pure Exploration"
_NeurIPS.cc/2025/Conference — NeurIPS 2025 poster_

### Official Review · Reviewer_eLPG · 2025-06-23

**Clarity:** 2
**Significance:** 3
**Originality:** 3
**Rating:** 4
**Confidence:** 3

**Summary:**

This paper studies the Preference-based Pure Exploration (PrePEx) problem, focusing on identifying the set of Pareto optimal arms in a vector-valued bandit. In this problem, the reward vectors are ordered using a given preference cone $\mathcal C$. The authors propose an algorithm based on a computationally tractable reduction of the minimization problem and the Frank-Wolfe optimizer. They prove both the algorithm's sample and computational complexity to demonstrate its effectiveness. Additionally, the authors validate the algorithm through experiments. Results demonstrate that their algorithm has advantages in both accuracy and efficiency.

**Questions:**

1. My assessment suggests that the authors' definition of the L-parameter exponential family deviates from its widely accepted form. This might mean the problem the authors are solving is actually limited to one-parameter exponential families, rather than truly extending to multi-parameter ones as the framing suggests.

2. Could the authors please clarify the precise definition of  $\tilde{\epsilon}$  in Lemma 2? Furthermore, it's not immediately clear whether both $\epsilon$ and $\tilde{\epsilon}$ can be arbitrarily close to zero. Gaining clarity on these points would be crucial for me to fully assess if the algorithm achieves asymptotic optimality as claimed.


If the authors can address my concerns regarding Lemma 2, I would consider improving my evaluation of the paper.
﻿
Separately, because I'm not deeply familiar with the Pareto Set Identification problem, I haven't been able to thoroughly check all the proofs, despite spending considerable time on them. Therefore, I'll be looking to other reviewers' insights to help inform any adjustments to my overall assessment.

**Ethical Concerns:**

["NO or VERY MINOR ethics concerns only"]

**Final Justification:**

This paper presents a solid, technically sound approach to Preference-based Pure Exploration. The method proposed is both novel and valuable, offering a promising solution to the problem being addressed.

However, the paper uses an existing lower bound from previous work, without providing further analysis of the lower bound under the new conditions. This leaves room for improvement in the theoretical aspects.

**Limitations:**

Yes.

**Paper Formatting Concerns:**

I did not notice any major formatting issues in this paper. The paper appears to follow the NeurIPS 2025 Paper Formatting Instructions correctly.

**Quality:**

3

**Strengths And Weaknesses:**

**Strengths**:

1. The authors design a computationally efficient and statistically optimal PrePEx algorithm that addresses the preference-based pure exploration problem with exponential family distributions.
2. The authors provide a thorough analysis of their algorithm. Their algorithm's sample complexity and computational complexity are proven.
3. The authors utilize real-world datasets to validate the algorithm's effectiveness experimentally.

**Weakness**:

1. The paper's definitions should adhere to standard, classical formulations, particularly for fundamental concepts such as partial order and the L-parameter exponential family. Regarding the latter, my assessment suggests that the authors' definition of the L-parameter exponential family deviates from its widely accepted form. This discrepancy could impact the validity of the paper's conclusions.

2. Lemma 2 stands out as the most crucial conclusion in the paper. It purports to demonstrate that the algorithm achieves asymptotic optimality as $\epsilon$ and $\tilde{\epsilon}$ approach zero. However, while $\epsilon$'s definition is briefly touched upon in the appendix, the definition of  $\tilde{\epsilon}$ is absent. This omission prevents me from evaluating whether these two values can indeed be arbitrarily close to zero, which makes it impossible to assess the meaningfulness of Lemma 2's conclusion fully.

3. The authors directly reference results from related papers for the lower bound of the problem.

---

> ### Author Rebuttal · Authors · 2025-07-31
>
> We thank the reviewer for taking their time to review our work. We respond to their concerns below.
>
> 1. **Standard definition of partial order:** We use the classical definition of order induced by a preference cone in vector optimisation literature [6,7]. This has been used across the board in the Pareto set identification with bandit feedback literature [2,3,4,5].
>
> 2. **$L$-parameter exponential family:** We think that there is a miscommunication here. Now, we include the following detailed definition of $L$-parameter exponential family in the paper.
>
> - *Definition:* Specifically, let us consider an exponential family defined as $f(\boldsymbol{X}\vert\boldsymbol{\theta}) := h(\boldsymbol{X}) \exp\left( \langle \eta(\boldsymbol{\theta}), T(\boldsymbol{X})\rangle - \psi(\boldsymbol{\theta})\right)$ with $\boldsymbol{\theta} \in \Theta \subseteq \mathbb{R}^L$ and $\boldsymbol{X} \in \mathbb{R}^L$. Then $\eta : \Theta \rightarrow \mathbb{R}^s$ is the natural parametrization for some $s \geq L$. $T : \mathbb{R}^L \rightarrow \mathbb{R}^s$ is the sufficient statistic for the natural parameter. $h(\boldsymbol{X})$ is the base density such that $h: \mathbb{R}^L \rightarrow [0,\infty)$. Finally, $\psi(\boldsymbol{\theta}) = \log \left( \int_{\mathcal{X}} h(\boldsymbol{X}) \exp\left( \langle \eta(\boldsymbol{\theta}), T(\boldsymbol{X})\rangle \mathrm{d}\boldsymbol{x}\right) \right)$ is the log-normaliser or log-partition function. It is fully defined by $\eta$, $T$ and $h$. Specifically, the valid natural parameters are the ones for which $\psi(\eta(\boldsymbol{\theta}))$ remains finite (Chapter 18 in [8]).
>
> 3. **Lemma 2:** Thank you for pointing this out for elaboration.
>
> - a. $\epsilon$ is the closeness parameter necessary to define the ''good events''. These events are crucial to ensure convergence of the sequential estimation problem. In our case, the two events are (i) Convergence of the sub-differential set, and (ii) Convergence of the objective function with respect to mean estimates for any allocation policy (Equation (18) and (19)) with respect to mean estimates.
>
> - b. Although, as correctly pointed out by the reviewer, there is no formal definition for $\tilde{\epsilon}$, which arises as for estimating the **minimum time** required to ensure convergence of allocation policy $\boldsymbol{\omega_t}$ to $\boldsymbol{\omega}^{\star}(M)$ and consequently $F(\boldsymbol{\omega_t}\mid M)$ to $F(\boldsymbol{\omega}^{\star}(M)\mid M)$. Mathematically, $\|F(\boldsymbol{\omega}^{\star}(M)\mid M)-F(\boldsymbol{\omega_t}\mid M)\| \leq 5\epsilon$ holds for $T\geq \bar h(T)$ (Theorem 5). Now, if $T \geq \left(\frac{2}{\tilde{\epsilon}}\right)^{11}$, then we get $T-\bar h(T) \geq \frac{T}{1+\tilde{\epsilon}}$. This is a standard algebraic trick used in the existing literature (for e.g., [1]).
>
> Both these constants can be arbitrarily small, and hence, the asymptotic optimality of FraPPE follows. We will incorporate the definition of $\tilde{\epsilon}$ in the updated manuscript.
>
> We hope that our explanations clarify all the doubts regarding correctness of our results and will motivate the reviewer to adjust their score appropriately. We remain available for further clarifications and discussions.
>
> **Reference:**
>
> [1] Wang, Po-An, Ruo-Chun Tzeng, and Alexandre Proutiere. "Fast pure exploration via frank-wolfe." Advances in Neural Information Processing Systems 34 (2021): 5810-5821.
>
> [2] Ararat, C. and Tekin, C. (2023). Vector optimization with stochastic bandit feedback. In PMLR.
>
> [3] Karagözlü, E. M., Yıldırım, Y. C., Ararat, C., and Tekin, C. (2024). Learning the pareto set under incomplete preferences: Pure exploration in vector bandits. PMLR.
>
> [4] Shukla, Apurv, and Debabrota Basu. (2024). Preference-based pure exploration. NeurIPS.
>
> [5] Crepon, E., Garivier, A., and M Koolen, W. (2024). Sequential learning of the Pareto front for
> multi-objective bandits. PMLR.
>
> [6] Jahn, Johannes, ed. Vector optimization. Berlin: Springer, 2009.
>
> [7] L{\"o}hne, Andreas, Vector optimization with infimum and supremum, Springer Science \& Business Media, 2011.
>
> [8] DasGupta, Anirban. "The exponential family and statistical applications." Probability for Statistics and Machine Learning: Fundamentals and Advanced Topics. New York, NY: Springer New York, 2011. 583-612.

---

> > ### Comment · Reviewer_eLPG · 2025-08-01
> > **Response to the Author Rebuttal**
> >
> > Thank you for your response to our questions. The clarification regarding $\tilde{\epsilon}$ has helped us more precisely assess the paper’s contribution. However, several concerns remain. As noted in our original review, our main focus is on the authors’ treatment of $\tilde{\epsilon}$, but we also encourage greater care in the use of key terminology to ensure that the claims are stated accurately.
> >
> > We list our comments point by point below:
> >
> > **1. Definition of partial order:**
> > There may be a typographical issue in the current definition. It seems the intended condition is $\mu \in \mu' + \alpha \overline{C}$ rather than $\mu \in \mu' + \overline{C}$. While this does not affect the main results, we ask the authors to verify this point for correctness. Regarding this point, the authors do not need to respond further.
> >
> > **2. $L$-parameter exponential family:**
> > The explanation provided in the response is clearer than that in the paper, but we suggest adopting a more standard definition. In particular, $L$ should refer to the dimension of the parameter vector $\theta$, not the number of observed variables $X$.
> >
> > More importantly, we remain concerned about the paper’s claim of addressing the general $L$-parameter exponential family. Based on our understanding, the authors are working with $L$ independent one-parameter exponential families, where each $X_i$ depends only on $\theta_i$, i.e., $f(X_i | \theta_i) = \dots$. This is distinct from the general $L$-parameter exponential family setting, where each $X_i$ may depend on the entire parameter vector $\Theta$, i.e., $f(X_i | \Theta) = \dots$.
> >
> > The one-parameter setting is a strict subset of the general case. If the paper addresses only this more restricted model, it should not claim results for the general exponential family setting. These two cases differ significantly in generality and complexity. The authors should clearly state which setting their results apply to, so that the conclusions correctly reflect the scope of the work.
> >
> > **3. Clarification on $\tilde{\epsilon}$:**
> > The authors’ discussion of $\tilde{\epsilon}$ has clarified its role in the main result. We believe this analysis is important and should be included in the paper to improve clarity. However, the current explanation still leaves a potential gap.
> >
> > Specifically, the authors assume that the condition $T \geq (2/\tilde{\epsilon})^{11}$ holds. However, by definition we have $T \leq \mathbb{E}[\tau_\delta]$, and the paper’s core contribution is an upper bound on $\mathbb{E}[\tau_\delta]$. If $\tilde{\epsilon}$ is very small, the condition $T \geq (2/\tilde{\epsilon})^{11}$ may not be satisfied. We believe that a lower bound on $\tilde{\epsilon}$ in terms of $\delta$ is needed to resolve this inconsistency. Moreover, since $\tilde{\epsilon}$ cannot be made arbitrarily small, the claim of asymptotic optimality is potentially problematic.
> >
> > ---
> >
> > **Summary:**
> > We appreciate the clarifications and acknowledge the additional insight they provide. However, we still observe a gap between the claimed conclusions and the theoretical results currently justified in the paper. If the authors can either rigorously support the claims as they stated, or revise them to accurately match the technical findings, we would be glad to consider increasing the score.

---

> ### Author Response · Authors · 2025-08-02
> **Response to Reviewer eLPG**
>
> **Clarification on $\tilde{\epsilon}:$** We think there is a misunderstanding here. We address the question in two layers.
>
> In lower bound-based pure exploration literature, we try to solve the lower bound with empirical estimates of means of the underlying arms at each time step. Thus, often the proof consists of two concentration phenomena: concentration of empirical mean to true mean and then concentration of the lower bound to the true one due to using estimates and alt-set optimisation. Finally, they together lead to the convergence in allocation and thus, correctness. This always introduces an $\epsilon$ to control the convergence, and often we introduce other parameters to explicitly write the time to concentrate to the aforementioned desired events. For example, $\epsilon$ and $\tilde{\epsilon}$ in our case; $\epsilon$ and $\eta$ in seminal work of [1]; $\epsilon$ and $\tilde{\epsilon}$ in [3] that developed the Frank-Wolfe method; and the following literature [2,4,5].
>
> One universal observation is that the $1/\epsilon$ and $1/\tilde{\epsilon}$ terms that appear in these upper bounds are independent of $\delta$. Also, note that given any fixed $\delta$, we choose $\epsilon$ and $\tilde{\epsilon}$ for proving the upper bound. Thus, first we divide the upper bound by $\log(1/\delta)$ and take $\delta \rightarrow 0$. This vanishes all the $1/\epsilon$ and $1/\tilde{\epsilon}$ dependent terms.
> Now, we can take the limits of the upper bound w.r.t. $\epsilon$ and $\tilde{\epsilon}$ (page 29-30, [1]; page 29, [2]; Appendix I.2., [3]; page 27-28, [4] etc.).
>
> If this limit matches with the lower bound, we call the algorithm "asymptotically optimal". We follow this nomenclature while clarifying this nuance in of setting $\epsilon$ and $\tilde{\epsilon}$ arbitrarily small in Lemma 2.
>
> 2. An elaborate proof and how we get to the "asymptotic optimality".
>
> We provide below a detailed discussion of the proof and why taking $\tilde{\epsilon} \rightarrow 0$ is accepted in Lemma 2. We omitted this as this algebraic trick has been proposed in [3], which is a very standard work in the domain and proposes the use of Frank-Wolfe optimiser for asymptotically optimal BAI. Later on this has been re-iterated through the literature. We will add now the following elaboration for completeness.
>
> **Step 1:** Let us remind of the good events:
>
>
> **Event $G_1(t)$:** The approximation error of solution in the Frank-Wolfe updates (Equation 7) is bounded by $\epsilon$.
> $$G_1(t) := \lbrace\max_{\boldsymbol{x}\in\Delta_K}\min_{\boldsymbol{h}\in H_{F(.|\hat M_t)(\boldsymbol{\omega_{t-1}}, r_t)}} (\boldsymbol{x} - \boldsymbol{\omega_{t-1}})^{\top} \boldsymbol{h}   - \epsilon > \max_{\boldsymbol{x}\in\Delta_K}\min_{\boldsymbol{h}\in H_{F(.|M)(\boldsymbol{\omega_{t-1}}, r_t)}} (\boldsymbol{x} - \boldsymbol{\omega_{t-1}})^{\top} \boldsymbol{h}   \rbrace$$
>
> **Event $G_2(t)$:** The error in objective function due to using the empirical means are bounded by $\epsilon$.
> $$
>     G_2(t) := \lbrace  \hat M_t \in S_i \wedge |F(\boldsymbol{\omega}|\hat M_t)-F(\boldsymbol{\omega}|M)| < \epsilon, \forall \boldsymbol{\omega} \in \Delta(K\gamma) \rbrace
> $$
> Hence, the Frank-Wolfe optimiser can stop when it is close to the true maximum.
>
> Following the concentration results, we can show that $G_1(t)\cap G_2(t)$ holds true, and thus, by Theorem 5, $F(\boldsymbol{\omega}^{\star}(M)|M) -F(\boldsymbol{\omega_t}|M) < 5\epsilon$ after time $t\geq \bar{h}(T)$.
>
> Here, $\bar{h}(T) = \min\lbrace t \in \mathbb{N} : t\geq T^{2/11}\underline{h}(T), \sqrt{\frac{t}{K}} \in \mathbb{N}\rbrace$, $\underline{h}(T) = \min\lbrace t \in \mathbb{N} : t\geq T^{8/11} + 2, \sqrt{\frac{t}{K}} \in \mathbb{N}\rbrace$, and $T\geq \max \lbrace \left(\frac{35D}{\epsilon}\right)^{11},T_{\epsilon,D}^{\frac{11}{8}}, (4K+1)^{\frac{11}{8}}\rbrace$. $D$ and $T_{\epsilon,D}$ are defined in the paper.
>
> **Step 2:**  Now, given $G_1(t)\cap G_2(t)$ holds true, we have
> $\min(\tau,T) \leq \bar{h}(T) + \sum_{t=\bar{h}(T)}^T \mathbf{1}\lbrace \tau > T\rbrace,$
> where $\tau$ is the stopping time of FraPPE.
> Thus,
> $$\min(\tau,T) \leq \bar{h}(T) + \sum_{t=\bar{h}(T)}^T \mathbf{1}\lbrace tF(\boldsymbol{\omega_t}\mid \hat{M}_t)< c(t,\delta)\rbrace.$$
>
> Now, for $t\geq \bar{h}(T)$,
> $F(\boldsymbol{\omega_t}\mid \hat{M}_t) < F(\boldsymbol{\omega_t}\mid M) - \epsilon < F(\boldsymbol{\omega}^\star(M)|M) - 6\epsilon.$
>
> Therefore, we have,
> $$\min({\tau,T}) \leq \bar{h}(T) + \sum_{t=\bar{h}(T)}^T \mathbf{1}\lbrace t(F(\boldsymbol{\omega^\star}(M)|M) - 6\epsilon)< c(t,\delta)\rbrace \leq \bar{h}(T)+ \frac{c(T,\delta)}{F(\boldsymbol{\omega_t}^{\star}(M)|M) - 6\epsilon}. $$
>
> Please check the next comment for continuation.

---

> > ### Author Response · Authors · 2025-08-02
> > **Response to Reviewer eLPG continued**
> >
> > **Step 3:** Finally, we define a constant
> > $$T_0 (\delta) \triangleq \inf\lbrace T\in \mathbb{N} : \bar{h}(T)+ \frac{c(T,\delta)}{F(\boldsymbol{\omega_t}^{\star}(M)|M) - 6\epsilon} \leq T\rbrace.$$
> >
> >
> > Now, we are ready to introduce the small constant $\tilde{\epsilon}\in (0,1)$, such that  $T-\bar h(T) \geq \frac{T}{1+\tilde{\epsilon}}$, when $T \geq \left( \frac{2}{\tilde{\epsilon}}\right)^{11}$ This choice of $\tilde{\epsilon}$ is reflected in non-asymptotic sample complexity upper bound (Lemma 6 in Appendix E.3).
> >
> > Now, following the algebra in [3], we get
> > $$
> >     T_{0}(\delta) \leq \max \lbrace  \left( \frac{2}{\tilde{\epsilon}}\right)^{11}, c_1(\mathcal M)\rbrace
> >     + \frac{1+\tilde{\epsilon}}{F(\boldsymbol{\omega}^{\star}(M)\mid M)-6\epsilon} \bigg[\log\left(\frac{1}{\delta}\right) + \log\log\left(\frac{1}{\delta}\right) \\
> >     + \log\left( \frac{(1+\tilde{\epsilon})c_2(\mathcal M)e}{(F(\boldsymbol{\omega}^{\star}(M)\mid M)-6\epsilon)}\right)
> >     + \log \log\left( \frac{(1+\tilde{\epsilon})c_2(\mathcal M)}{(F(\boldsymbol{\omega}^{\star}(M)\mid M)-6\epsilon)}\right)\bigg]
> > $$
> >
> > **Step 4:** This finally leads to Equation (20) in our draft.
> > We observe that the term due to $\tilde{\epsilon}$ is independent of $\log(1/\delta)$. Thus, if we divide both sides Equation (20) by $\log(1/\delta)$ and take the limit $\delta \rightarrow 0$, the terms due to $1/\tilde{\epsilon}$ vanishes. This yields our final bound in Lemma 2.
> >
> > Now, as we can set $\epsilon$ and $\tilde{\epsilon}$ arbitrarily small and independent of $\delta$, this bound is known to be asymptotically optimal as in [1,2,3,4,5]. To better explicate the nuances, we also provide a non-asymptotic bound in Lemma 6.
> >
> > **$L$-parameter exponential family:** Thank you for your further comments on the definition. We would like to elaborate on this. In our setting, we have the knowledge of $\Sigma$ (covariance matrix), only the $L$-dimensional mean vectors $M_k, k \in [1,K] $ are unknown. Thus, only mean parametrisation remains enough, which is $L$-dimensional. To emphasise this, we refer to it as $L$-parameter exponential family.
> >
> > - *Example: Multivariate Gaussian with mean $\boldsymbol{\mu}$ and covariance $\boldsymbol{\Sigma}$.*
> > The density function is
> > $$
> > f(\mathbf{x})= (2 \pi)^{-\frac{L}{2}} \operatorname{det}(\boldsymbol{\Sigma})^{-\frac{1}{2}} \exp \left(-\frac{1}{2}(\mathbf{x}-\boldsymbol{\mu})^{\top} \boldsymbol{\Sigma}^{-1}(\mathbf{x}-\boldsymbol{\mu})\right).
> > $$
> > We can rewrite it as
> > $$
> > f(\mathbf{x}) = (2 \pi)^{-\frac{L}{2}} \exp \left(\left\langle\boldsymbol{\Sigma}^{-1} \boldsymbol{\mu}, \mathbf{x}\right\rangle+\left\langle\operatorname{vec}\left(-\frac{1}{2} \boldsymbol{\Sigma}^{-1}\right), \operatorname{vec}\left(\mathbf{xx}^{\top}\right)\right\rangle-\frac{1}{2}\left[\log |\boldsymbol{\Sigma}|+\boldsymbol{\mu}^{\top} \boldsymbol{\Sigma}^{-1} \boldsymbol{\mu}\right]\right).
> > $$
> > Thus, the base density is $h(\boldsymbol{X}) = (2 \pi)^{-\frac{L}{2}}$. The natural parameter is $\eta(\boldsymbol{\theta}) = \binom{\boldsymbol{\Sigma}^{-1} \boldsymbol{\mu}}{\operatorname{vec}\left(-\frac{1}{2} \boldsymbol{\Sigma}^{-1}\right)}$. The sufficient statistic is $T(\boldsymbol{x}) = \binom{\mathbf{x}}{\operatorname{vec}\left(\mathbf{x} \mathbf{x}^{\top}\right)}$. The log-normaliser or log-partition function is $\psi(\boldsymbol{\theta}) = \frac{1}{2}\left[\log \operatorname{det}(\boldsymbol{\Sigma})+\boldsymbol{\mu}^{\top} \boldsymbol{\Sigma}^{-1} \boldsymbol{\mu}\right]$. Here the natural parameter is of order $L+L^2$. But, as a consequence of knowing $\Sigma$ beforehand, only mean parameterisation of order $L$ is identifiable.
> >
> >
> > We have carefully checked the whole manuscript to observe we have never used independence among the indices of $L$-dimensional random variable in any of the proofs. Thus, we advocate the fact that our theoretical results hold for general $L$-parameter exponential family.
> >
> > Let us know if the above explanations made our claims in the paper more clear. We are eager to discuss if there is further concern.
> >
> >
> >
> > - **References**
> >
> > [1] Garivier, Aurélien, and Emilie Kaufmann. "Optimal best arm identification with fixed confidence." Conference on Learning Theory. PMLR, 2016.
> >
> > [2] Jedra, Yassir, and Alexandre Proutiere. "Optimal best-arm identification in linear bandits." Advances in Neural Information Processing Systems 33 (2020): 10007-10017.
> >
> > [3] Wang, Po-An, Ruo-Chun Tzeng, and Alexandre Proutiere. "Fast pure exploration via frank-wolfe." Advances in Neural Information Processing Systems 34 (2021): 5810-5821.
> >
> > [4] Mukherjee, Arpan, and Ali Tajer. "Best Arm Identification in Stochastic Bandits: Beyond $\beta-$ optimality." arXiv preprint arXiv:2301.03785 (2023).
> >
> > [5] Degenne, Rémy, Wouter M. Koolen, and Pierre Ménard. "Non-asymptotic pure exploration by solving games." Advances in Neural Information Processing Systems 32 (2019).

---

> ### Comment · Reviewer_eLPG · 2025-08-02
> **Response to the Author Rebuttal**
>
> Thank you for your detailed responses to my questions. Your answers have sufficiently addressed my concerns. I recommend incorporating this discussion into the main text or the appendix of the paper to enhance its completeness and clarity.
>
> Additionally, please use the formal definition of the L-parameter exponential family in the next version of the paper, and clearly specify the relationship and distinction between the dimensions of $X$ and $\Theta$.
>
> Based on these improvements, I will be updating my score to borderline accept.

---

> > ### Author Response · Authors · 2025-08-08
> > **To Reviewer eLPG**
> >
> > Dear Reviewer,
> >
> > As the rebuttal period is terminating, we thank you for your thorough and thoughtful evaluation of our submission. Your constructive feedback has provided valuable guidance in improving the clarity and quality of our work.
> >
> > Sincerely,
> > The Authors

---

### Official Review · Reviewer_aS4F · 2025-06-30

**Clarity:** 2
**Significance:** 3
**Originality:** 3
**Rating:** 4
**Confidence:** 3

**Summary:**

This paper investigates the design of computationally efficient algorithms for the preference-based pure exploration problem. The authors reduce the corresponding lower-bound optimization problem into a computationally tractable form and solve it using the Frank-Wolfe algorithm. The authors further provide theoretical guarantees for proposed algorithms on computational and sample complexity. Both theoretical results and empirical experiments demonstrate the superiority of their proposed algorithm.

**Questions:**

Please see the strengths and weaknesses section.

**Ethical Concerns:**

["NO or VERY MINOR ethics concerns only"]

**Final Justification:**

My concerns have been addressed during the discussion. I keep my positive score.

**Quality:**

3

**Strengths And Weaknesses:**

Strengths:
1. This work explores a computationally efficient algorithm for solving the preference-based pure exploration problem. It presents a rigorous theoretical analysis on both the computational and sample complexity of the proposed method.

Weaknesses:
1. In the problem formulation part of Section 2, the definition of the reward distribution $\nu_a$ seems somewhat unconventional, as the authors only define its mean $\boldsymbol{\mu}_a$. In contrast, (Shukla and Basu, 2024) further define the corresponding covariance matrix $\Sigma$ to account for noisy feedback.

2. In Figure 1, the authors intend to illustrate how the Pareto frontier varies under different preference cones. However, in Figure 1, the two Pareto frontiers overlap, which easily confuses readers. The authors could consider plotting the Pareto frontiers for both $\pi/2$ and $\pi/3$ cones to better illustrate the difference.

3. Some concepts are used without formal definitions, such as the term "stationary policies" in line 209.

4. In line 298, the authors claim that the Frank-Wolfe algorithm can handle non-smooth optimization objectives by constructing the corresponding $r$-sub-differential set. Can this approximation method guarantee recovery of the original optimal solution? In addition, please justify the choice of the hyper-parameter $r_t$.

5. A minor writing suggestion: in line 205, the concept of $p$ pure policies has not been formally introduced. To avoid potential confusion, it is advisable to use the notation \Delta_K \setminus \{\pi^*\} rather than \Delta_K \setminus \{\pi_i^*\}.

6. It is recommended that the authors provide a comprehensive table comparing different preference-based pure exploration methods in terms of algorithmic complexity (including time and space complexity) as well as the upper bounds on sample complexity. This would help readers better understand the strengths and weaknesses of the proposed method in a broader context.

7. The authors could elaborate more on the technical challenges in accelerating algorithms solving the preference-based pure exploration problem, and clarify the theoretical contributions of their proposed approach.

---

> ### Author Rebuttal · Authors · 2025-07-31
>
> We thank the reviewer for reviewing and acknowledging our work. We respond to the concern raised below:
>
> **Covariance matrix:** In the problem formulation, we did not highlight the covariance matrix of reward distribution as it is known in our setup and the main challenge in bandits is the means of the reward distributions being unknown.
> It is also reflected in the derivation of the lower bound in [1].
> Note that, we also consider the reward distributions can have higher moments.
> In Assumption 1, we adhered to the definition of a multi-parameter exponential family (Chapter 18 in [2]), where $\psi(\eta(\boldsymbol{\theta}))$ is the log-partition function and $\eta$ is the natural parametrisation. If we differentiate $\psi$ w.r.t. $\eta$, we get the mean $\boldsymbol{\mu}$, and the second-order derivative $\nabla_{\eta(\boldsymbol{\theta})}^2 \psi(\eta(\boldsymbol{\theta}))$ yields the covariance matrix.
> We will explicitly mention the covariance in Section 2 to avoid any confusion around the definition. %\todo[inline]{Section 2 he said, change this writing.}
>
> **Figure 1:**  Thank you for pointing this out. Although the intend was to show that different preference cone results in different Pareto optimal arms, but they can have intersection. We will change it such that there are no intersections so that it is more clearly understandable.
>
> **Stationary policy:** By stationary policy, we wanted to convey that the pure policies (stands for playing an arm) does not change over time. We will definitely include a formal definition.
>
> **Line 205:** We will formally define pure policies (policies that have all the probability mass on a single arm). Thank you for pointing this out.
>
> **Choice of $r_t$:** As per the existing literature on Frank-Wolfe [3], the approximation parameter $r_t$ should shrink at a rate $\Omega(1/t)$. It ensures the convergence of empirical allocation $\boldsymbol{\omega}_t$ to optimal allocation $\boldsymbol{\omega}^{\star}(M)$ (Theorem 5). In practice, we have chosen $r_t = \frac{K}{t^{0.9}}$.
>
> **Convergence of sub-differential set:** First, we prove continuity of the mapping from allocation policy to sub-differential set in Corollary 1 (Appendix D.2). Then, we show the convergence of Frank-Wolfe iterates (Equation (7)) while we iteratively optimise over the allocation policy (Appendix D.3).
>
> **Comparison table:** This a very nice suggestion. With the additional available page, we will definitely include a table that summarises the complexity of different approaches to solve Pareto set identification and their corresponding complexity.
>
> **Contribution and technical challenges:** In this paper, we successfully address the open problem posed in [4] to propose the first computationally efficient and asymptotically optimal algorithm, that works for arbitrary preference cones and exponential family distributions, and also achieves the lowest sample complexity and probability of error for identifying the exact set of Pareto optimal arms. We mention all our contributions in line 88 to 107 in the main manuscript. The key technical challenges are discussed in the beginning of Section 3.1 and Section 3.2, i.e., solving optimisation over alternating instance and Pareto optimal policy set. Complying with the reviewer's remark, we will again summarise the technical challenges tackled in this work.
>
>
> We hope that we have addressed all the concerns. We are eager to discuss further if the reviewer has more concerns or comments.
>
>
>
> **Reference:**
>
> [1] Shukla, Apurv, and Debabrota Basu. (2024). Preference-based pure exploration. NeurIPS.
>
> [2] DasGupta, Anirban. "The exponential family and statistical applications." Probability for Statistics and Machine Learning: Fundamentals and Advanced Topics. New York, NY: Springer New York, 2011. 583-612.
>
> [3] Wang, Po-An, Ruo-Chun Tzeng, and Alexandre Proutiere. "Fast pure exploration via frank-wolfe."
> Advances in Neural Information Processing Systems 34 (2021): 5810-5821.
>
> [4] Crepon, E., Garivier, A., and M Koolen, W. (2024). Sequential learning of the Pareto front for
> multi-objective bandits. PMLR.

---

> > ### Comment · Reviewer_aS4F · 2025-08-06
> >
> > Thanks for the detailed response. I have no further questions.

---

> ### Author Response · Authors · 2025-08-08
> **To Reviewer aS4F**
>
> Dear Reviewer,
>
> We are grateful for the time and attention you have devoted to evaluating our submission. Your detailed and constructive feedback is much appreciated.
>
> Warm regards,
> The Authors

---

### Official Review · Reviewer_6Msu · 2025-07-02

**Clarity:** 3
**Significance:** 4
**Originality:** 3
**Rating:** 5
**Confidence:** 2

**Summary:**

This paper tackles Preference-based Pure Exploration (PrePEx) in multi-objective bandits, aiming to identify the Pareto-optimal arms efficiently and confidently. The authors propose FraPPE, a computationally efficient and asymptotically optimal algorithm that leverages structural reductions, a Frank-Wolfe optimizer, and a novel stopping rule. Empirical results demonstrate gains in sample efficiency and stability over prior methods.

**Questions:**

Can the same approach (FraPPE) be extended or scaled to combinatorial pure exploration problems, such as combinatorial bandits?

How sensitive is FraPPE to mild deviations from its theoretical assumptions in practical scenarios?

Could the authors report or discuss FraPPE's actual runtime in wall-clock seconds (rather than just iteration or sample complexity), especially on the real-world dataset? This would provide practical insight into computational efficiency beyond theoretical complexity.

**Ethical Concerns:**

["NO or VERY MINOR ethics concerns only"]

**Final Justification:**

My concerns have been addressed, and I have decided to raise my score to 5.

**Limitations:**

Yes.

**Paper Formatting Concerns:**

No.

**Quality:**

3

**Strengths And Weaknesses:**

**Strengths:**

•	The paper tackles an interesting and well-motivated problem; namely preference-based pure-exploration.

•	The works proposes a practical approach which reduce the computational complexity of solving the tackled problem.

•	The paper results are supported by theoretical guarantees and empirical analysis on both synthetic and real-world problems.


**Weaknesses:**

•	Additional tests on larger-scale or more diverse real-world settings would strengthen the empirical claims.

•	For clearer presentation and better contrast with prior work, it would be helpful to summarize the main differences and contributions in a comparison table.

•	(minor) While multi-armed bandits provide a foundational framework for studying sequential decision-making under uncertainty, describing them as the "archetypal setup" of reinforcement learning is somewhat misleading (line 36). Bandits represent a simplified special case without state transitions or delayed consequences, which are central features of reinforcement learning.

---

> ### Author Rebuttal · Authors · 2025-07-31
>
> We thank the reviewer for their valuable review and remarks. We respond to the concerns raised below:
>
> 1. **Experimental validation and computational bottleneck:**
>
> - a. *Runtime of FraPPE:* While only using CPUs (without parallel processing and tensorization), each iteration of FraPPE takes $\sim 64$ milliseconds/iteration for COVBOOST ($19$ arms and $3$ objectives). While for DB [1] ($128$ arms and $2$ objectives), FraPPE clocked in $\sim 98$ milliseconds/iteration. For synthetic data with correlated objectives ($5$ arms and $2$ objectives), it takes $\sim 14.6$ milliseconds/iteration. These results are obtained by averaging over 1000 iterations.
>
> - b. *Runtime of baselines:* The implementations available in the literature differ in their setup and programming languages: [2] uses Julia, [3] uses C++. In contrast, to make the code accessible to wider audience, we chose Python3 for implementation with only three well-known packages: Numpy, CVXPY, and Scipy. Thus, we surmise it would be unfair to compare the clock runtimes of the different baselines implementations.
> But resonating with the reviewer's remark, we have now conducted a runtime analysis with varying $K$, $L$ to validate our theoretical claims.
>
> - c. *Conic programming:* Yes, we agree that conic programming is the computationally costliest module here. Though we believe while solving the lower bound, conic programs are unavoidable as it is required to find the Pareto for generic cones. Thus, it has been a part of the open problem posed by [2], i.e. to reduce the computational complexity per iteration up to the complexity of Pareto set computation.
>
> - d. *Larger dataset- DB [1]:* We have tested on all the datasets used in the SOTA paper for exact Pareto set identification: PSIPS [3]. We are currently testing FraPPE on the *DB (Disc Brake) dataset* ($128$ arms and $2$ objectives) that presents an efficiency and safety optimization problem in disc brake manufacturing for automotives. In the literature, larger datasets like DB are used only to attain $\epsilon$-approximate Pareto set identification [5,6], while FraPPE aims for exact identification and succeeds as per the preliminary results.
>
> **Comparison table:** This is indeed a very nice suggestion. With the extra available space, we will definitely include a table that summarises the complexity of different approaches to solve Pareto set identification and their corresponding complexity.
>
> **FraPPE in Combinatorial Bandits:** We think like most of the algorithmic framework that exists for solving pure exploration in multi-objective bandits, for e.g., successive arm-elimination [1] or Bayesian posterior sampling based [2] etc., FraPPE can also be extended to solve pure exploration in combinatorial bandits. That means the goal will be to identify the optimal combinatorial or subset of arms among $2^K$ possible subsets (Or a policy with basis consisting pure policies corresponding optimal subset of arms) that maximises the reward . Additionally, the agent will observe feedback for each of the arms in the subset chosen to play at any time step. Although we think this problem is more complex due to exploration among exponentially large $2^K$ options rather than usual $K$ actions. This bottleneck calls for careful adaptation of the existing lower bound to the combinatorial setup (that might depend on complexity of differentiating between the optimal subset and the second best subset of arms). Still, this problem seems solvable by adapting FraPPE directly with necessary extensions.
>
>
> **Mild deviations in theoretical assumptions:** The only two assumptions that we require are: the knowledge of the cone $\mathcal{C}$ and the reward distribution being in an exponential family with non-zero second moment. The first one is a fundamental requirement as the lower bound of [6], which is our starting point, requires this. If we have a misspecified cone, the algorithm would still run and identify a part of the Pareto optimal arms but we cannot theoretically say anything about correctness and optimality. The second one is more of a theoretical requirement to prove optimality of FraPPE. The lower bound and the algorithmic technique would work for any distribution. But since the GLRT stopping rule is known to be tight for exponential families, not only FraPPE but any lower bound-driven pure exploration algorithm [7,8,9] even in simple best arm identification assumes an exponential family to prove optimality [1,2,6].
> In brief, we can run FraPPE for instances deviating from these assumptions but we cannot guarantee the theoretical goodness any more.
>
> We hope that we have addressed all the concerns. We are eager to discuss further if the reviewer has more concerns or comments.
>
>
> **References:**
>
> [1] Crepon, E., Garivier, A., and M Koolen, W. (2024). Sequential learning of the Pareto front for multi-objective bandits. AISTATS.
>
> [2] Kone, C., Jourdan, M., and Kaufmann, E. (2025). Pareto set identification with posterior sampling. AISTATS.
>
> [3] Tanabe, R. and Ishibuchi, H. (2020). An easy-to-use real-world multi-objective optimization problem suite. Applied Soft Computing, 89:106078.
>
> [4] Ararat, C. and Tekin, C. (2023). Vector optimization with stochastic bandit feedback. In PMLR.
>
> [5] Karagözlü, E. M., Yıldırım, Y. C., Ararat, C., and Tekin, C. (2024). Learning the pareto set under incomplete preferences: Pure exploration in vector bandits. PMLR.
>
> [6] Shukla, Apurv, and Debabrota Basu. (2024). Preference-based pure exploration. NeurIPS.
>
> [7] Garivier, A. and Kaufmann, E. (2016). Optimal best arm identification with fixed confidence. COLT.
>
> [8] Wang, P.-A., Tzeng, R.-C., and Proutiere, A. (2021). Fast pure exploration via Frank-Wolfe. NeurIPS.
>
> [9] Degenne, R., Ménard, P., Shang, X., and Valko, M. (2020). Gamification of pure exploration for linear bandits. ICML.

---

### Official Review · Reviewer_cohg · 2025-07-02

**Clarity:** 4
**Significance:** 3
**Originality:** 3
**Rating:** 5
**Confidence:** 3

**Summary:**

This paper considers pure exploration in a K-armed stochastic bandit problem with L-dimensional rewards. The goal is to identify the Pareto set of policies (distributions over arms) given a fixed confidence level in as few evaluations as possible. The paper builds an approach based on the lower bound on Pareto front identification. Using structural properties, it forms a computationally tractable alternative to the minimax optimization problem in the lower bound. A track-and-stop inspired arm selection and stopping rule is proposed based on the tractable lower bound. Asymptotic optimality of the proposed rule is established. Experiments are carried out to show how the proposed rule compares with SoTA.

**Questions:**

1) It is hard to parse the contribution of Section 3.3. What is the exact reduction in computational complexity when the polar cone representation of alternating instances is used?

2) Please elaborate on how the $ L$-parameter exponential family extends the scalar exponential family. What conditions are required on functions $h$, $\eta$, and $T$?

**Ethical Concerns:**

["NO or VERY MINOR ethics concerns only"]

**Final Justification:**

The authors have sufficiently addressed my comments regarding clarifications. This is solid work. Please incorporate these clarifications in the final version of the paper.

**Limitations:**

Yes.

**Paper Formatting Concerns:**

N/A.

**Quality:**

4

**Strengths And Weaknesses:**

Strengths:

The paper is clearly written. The authors did a good job of explaining this complicated pure exploration problem in nine pages.

This paper solves an important open problem in bandit-based vector optimization by proposing a computationally efficient, asymptotically optimal, fixed-confidence pure exploration algorithm. The proposed algorithm utilizes a tractable reformulation of the intractable optimization problem. Theoretical analysis of the computational complexity demonstrates a significant improvement over existing approaches. The paper proposes a non-asymptotic, asymptotically optimal bound on the sample complexity. Experiments support theoretical claims.


Weaknesses:

There is no major weakness.

---

> ### Author Rebuttal · Authors · 2025-07-31
>
> We thank the reviewer for acknowledging our work and taking their time to review. We respond to the questions raised below:
>
> 1. **Section 3.3 :** In this section, we leverage the structure of the alternating instance $\tilde{M}^{L \times K}$ to gain computational efficiency for the two innermost minimisation problems in Equation (3).
>
> - (a) Direct optimisation over $\tilde{M}^{L \times K}$ can have complexity $\mathcal{O}(KL)$, since optimising an $L$-dimensional vector inside a cone has complexity $\mathcal{O}(L)$. With the polar cone representation of the alternating instances (Proposition 1), $\tilde{M}$ can be expressed with $\boldsymbol{y}, \boldsymbol{\pi}_i^{\star}, \boldsymbol{\pi}_j $. Here, $\boldsymbol{y}$ is an $L$-dimensional vector on the boundary of the polar cone of the given preference cone $\mathcal{C}$. Thus, for a fixed $\boldsymbol{\pi}_i^{\star}$ and $\boldsymbol{\pi}_j$, we now optimise over an $L$-dimensional vector rather than a $L\times K$ matrix.
>
> - (b) Further with this reduction, we observe that $\boldsymbol{z}^L$ and $\boldsymbol{y}^L$ are defined on the cone $\mathcal{C}$ and its polar cone $\mathcal{C}^{\circ}$. Thus, any efficient second order conic program solver (for e.g., CVXPY) can solve both the optimisation problems in $\mathcal{O}(L)$ rather than $\mathcal{O}(L^2)$.
>
>
> Hence, the total computational gain achieved in Section 3.3 is  $\mathcal{O}(KL^2) \rightarrow \mathcal{O}(L)$.
>
>
> 2. **L-parameter exponential family:** Since we adopted the standard definition of multi-parameter exponential family from statistics ((Chapter 18 in [1])), which is also a generalisation of multivariate Gaussian of dimension $L$ with mean $\boldsymbol{\mu}$ and covariance $\boldsymbol{\Sigma}$, we omitted a detailed discussion. Thanks for the remark. We clarify it here and must add the details with the extra page available.
>
> - *Definition:* Specifically, let us consider an exponential family defined as $f(\boldsymbol{X}\vert\boldsymbol{\theta}) := h(\boldsymbol{X}) \exp\left( \langle \eta(\boldsymbol{\theta}), T(\boldsymbol{X})\rangle - \psi(\boldsymbol{\theta})\right)$ with $\boldsymbol{\theta} \in \Theta \subseteq \mathbb{R}^L$ and $\boldsymbol{X} \in \mathbb{R}^L$. Then $\eta : \Theta \rightarrow \mathbb{R}^s$ is the natural parametrization for some $s \geq L$. $T : \mathbb{R}^L \rightarrow \mathbb{R}^s$ is the sufficient statistic for the natural parameter. $h(\boldsymbol{X})$ is the base density such that $h: \mathbb{R}^L \rightarrow [0,\infty)$. Finally, $\psi(\theta) = \log \left( \int_{\mathcal{X}} h(\boldsymbol{X}) \exp\left( \langle \eta(\theta), T(\boldsymbol{X})\rangle \mathrm{d}\boldsymbol{x}\right) \right)$ is the log-normaliser or log-partition function. It is fully defined by $\eta$, $T$ and $h$. Specifically, the valid natural parameters are the ones for which $\psi(\eta(\theta))$ remains finite.
>
> We hope that our response have addressed the concerns and questions. We remain available for further discussion.
>
>
> **Reference:**
> [1] DasGupta, Anirban. "The exponential family and statistical applications." Probability for Statistics and Machine Learning: Fundamentals and Advanced Topics. New York, NY: Springer New York, 2011. 583-612.

---

> > ### Comment · Reviewer_cohg · 2025-08-04
> >
> > Thanks for the clarifications. My questions are addressed.

---

> ### Author Response · Authors · 2025-08-08
> **To Reviewer cohg**
>
> Dear reviewer,
>
> As the rebuttal period is coming to the end, we thank you for taking time to thoroughly review our work. We appreciate your constructive feedback.
>
> Regards, Authors

---

### Official Review · Reviewer_JNA1 · 2025-07-11

**Clarity:** 3
**Significance:** 4
**Originality:** 4
**Rating:** 5
**Confidence:** 3

**Summary:**

This paper addresses an open problem in preference-based pure exploration (PrePEx) in multi-objective bandits, where the goal is to identify all Pareto optimal arms under a given preference cone with fixed confidence. The authors propose FraPPE, a computationally efficient algorithm that provides an asymptotically optimal solution for arbitrary preference cones and exponential family distributions. The key contributions include:
- Structural reductions of the optimization problem, avoiding convex hull computations,
- Application of Frank-Wolfe optimization for efficient lower bound tracking, and
- Theoretical guarantees of asymptotic optimality with improved computational complexity.
- Simulation results to support the proposed results.

**Questions:**

**Questions**

**Preference Cone Specification in Practice:** What are the real-world models with the preference cone C, and what are their typical values? Also, how sensitive is the algorithm's performance to misspecification of the preference cone C? In real applications, domain experts may have uncertainty about their preferences. Have you considered extensions where the cone is learned or refined during exploration?

**Computational Bottlenecks:** Despite the theoretical improvements, conic programs per iteration could still be computationally demanding. Does an experimental time breakdown on which steps of the algorithm provide the time bottleneck make sense?

**Ethical Concerns:**

["NO or VERY MINOR ethics concerns only"]

**Final Justification:**

I am happy with the author's response. I would like ot retain the score. I really like the paper as it stands.

**Limitations:**

It would be great if the authors added a dedicated `limitations` section discussing the points touched upon in the Weaknesses and Questions section.

**Paper Formatting Concerns:**

The paper is structured quite well. No major formatting issues were observed. The paper follows standard mathematical formatting conventions with proper theorem environments, notation, and references.

**Quality:**

4

**Strengths And Weaknesses:**

**Strengths:**

**Novel Theoretical and Algorithmic Contributions:** The paper makes significant progress on an open problem posed by Crepon et al. (2024) by providing a computationally efficient and asymptotically optimal algorithm for arbitrary preference cones. The decomposition of the Alt-set without requiring convex hull computations is a significant theoretical advance. The reduction from O(K²L) to O(KL min{K,L}) computational complexity is non-trivial and practically important.

**Problem Relevance and Motivation:** The multi-objective bandit setting with preference cones is well-motivated through concrete applications in clinical trials (COV-BOOST), material design, and RLHF. The inability to scalarize objectives a priori is a genuine challenge in these domains, making the preference cone framework particularly relevant.

**Strong Literature Positioning:** The paper does a great job of placing itself amid related work. The authors clearly articulate the limitations of existing approaches: successive elimination methods that only find approximate Pareto sets, the computational intractability of PreTS, and the restriction of previous optimal algorithms to other distributions. The connection to pure exploration with multiple answers and the Frank-Wolfe literature is also well stated.

**Clear Presentation:** The paper is well-structured with a logical flow from problem formulation through theoretical contributions to algorithm design. The use of illustrative examples aids a deeper understanding of the work. Technical concepts are introduced progressively with appropriate mathematical insights.

**Weaknesses:**

**Limited Experimental Validation:** While the COV-BOOST results are compelling, the experimental section would benefit from more ablation studies: (i) runtime breakdown to validate computational efficiency claims, (ii) scaling effects with varying K and L, and (iv) ablation studies on other parameters.

**Assumptions and Practical Considerations:** The assumption of perfect knowledge of preference cones may be restrictive in practice. Maybe a discussion of robustness to misspecified cones would be a nice add.

---

> ### Author Rebuttal · Authors · 2025-07-31
>
> We thank the reviewer for taking time to review our work, and for appreciating the novelty and clarity of our contributions. We respond to the questions below, while we include the promising improvements in our updated draft.
>
> 1. **Experimental validation and computational bottleneck:**
>
> - a. *Runtime of FraPPE:* While only using CPUs (without parallel processing and tensorization), each iteration of FraPPE takes $\sim 64$ milliseconds/iteration for COVBOOST ($19$ arms and $3$ objectives). While for DB [1] ($128$ arms and $2$ objectives), FraPPE clocked in $\sim 98$ milliseconds/iteration. For synthetic data with correlated objectives ($5$ arms and $2$ objectives), it takes $\sim 14.6$ milliseconds/iteration. These results are obtained by averaging over 1000 iterations.
>
> - b. *Runtime of baselines:* The implementations available in the literature differ in their setup and programming languages: [2] uses Julia, [3] uses C++. In contrast, to make the code accessible to wider audience, we chose Python3 for implementation with only three well-known packages: Numpy, CVXPY, and Scipy. Thus, we surmise it would be unfair to compare the clock runtimes of the different baselines implementations.
> But resonating with the reviewer's remark, we have now conducted a runtime analysis with varying $K$, $L$ to validate our theoretical claims.
>
> - c. *Conic programming:* Yes, we agree that conic programming is the computationally costliest module here. Though we believe while solving the lower bound, conic programs are unavoidable as it is required to find the Pareto for generic cones. Thus, it has been a part of the open problem posed by [2], i.e. to reduce the computational complexity per iteration up to the complexity of Pareto set computation.
>
> - d. *Larger dataset- DB [1]:* We have tested on all the datasets used in the SOTA paper for exact Pareto set identification: PSIPS [3]. We are currently testing FraPPE on the *DB (Disc Brake) dataset* ($128$ arms and $2$ objectives) that presents an efficiency and safety optimization problem in disc brake manufacturing for automotives. In the literature, larger datasets like DB are used only to attain $\epsilon$-approximate Pareto set identification [5,6], while FraPPE aims for exact identification and succeeds as per the preliminary results.
>
> 2. **Specification of the cone:**
>
> - a. *Practicality of knowing $\mathcal{C}$:* We assume to know the preference cone following the present literature of PrePEx, where the SOTA results are mostly available for the positive orthant as the cone. Knowledge of $\mathcal{C}$ is available in multiple real-world systems: polyhedral cones are used in *aircraft design optimisation* [7], positive orthant cone ($\mathbb{R}^L_{+}$) are used in Portfolio optimization balancing return, risk, and liquidity [8], customised asymmetric cones are used in climate policy optimization with multiple national goals [9].
>
> - b. *Learning $\mathcal{C}$:* We agree that sequentially learning the cone from multiple expert's feedback while exploring is an interesting problem with its own set of challenges. But if we do not know $\mathcal{C}$ or encounter misspecification, the lower bound of [10] is not valid and we should derive a new lower bound. It is an ongoing study and we will mention it in future works.
>
> We hope that our response have addressed the concerns and questions. We remain available for further discussion.
>
> **References:**
>
> [1] Tanabe, R. and Ishibuchi, H. (2020). An easy-to-
> use real-world multi-objective optimization problem
> suite. Applied Soft Computing, 89:106078.
>
> [2] Crepon, E., Garivier, A., and M Koolen, W. (2024). Sequential learning of the Pareto front for
> multi-objective bandits. PMLR.
>
> [3] Kone, C., Jourdan, M., and Kaufmann, E. (2024). Pareto set identification with posterior sampling.
>
> [4] Ararat, C. and Tekin, C. (2023). Vector optimization with stochastic bandit feedback. In PMLR.
>
> [5] Karagözlü, E. M., Yıldırım, Y. C., Ararat, C., and Tekin, C. (2024). Learning the pareto set under
> incomplete preferences: Pure exploration in vector bandits. In PMLR.
>
> [6] Bradley, R. A. and Terry, M. E. (1952). Rank analysis of incomplete block designs: I. the method of paired comparisons. Biometrika.
>
> [7] Mavrotas, G. (2009). Effective implementation of the ε-constraint method in Multi-Objective Mathematical Programming problems. Applied Mathematics and Computation.
>
> [8] Ehrgott, M. (2005). Multicriteria Optimization. Springer.
>
> [9] Keeney, R. L., & Raiffa, H. (1993). Decisions with Multiple Objectives: Preferences and Value Trade-Offs. Cambridge University Press.
>
> [10] Shukla, Apurv, and Debabrota Basu. "Preference-based pure exploration." Advances in Neural Information Processing Systems 37 (2024): 17313-17347.

---

> > ### Comment · Reviewer_JNA1 · 2025-08-08
> >
> > Thank you for addressing my concerns. I am satisfied with the responses to my queries

---

> ### Author Response · Authors · 2025-08-08
> **To Reviewer JNA1**
>
> Dear reviewer,
>
> Thanks for the time spent reviewing our work. We appreciate your constructive feedback.
>
> Regards,
> Authors

---

### Note · Authors · 2025-08-15

To the Reviewers and the AC,

We extend our gratitude to you for your thorough evaluation of our work and recognising the novel contributions towards the literature of preference-based Multi-objective bandits. We deeply appreciate the constructive feedback received throughout this review process. We are in the process of refining the manuscript based on these feedbacks. Specifically, we recognise four major inclusions apart from minor typos or clarifications:

- **Definition of $L$-parameter exponential family:** We have noticed confusion around the definition of $L$-parameter exponential family in Assumption 1 pointed out by Reviewer cohg and eLPG. We include a detailed discussion in accordance.

- **Additional experiments:** We include the numerical runtime analysis of FraPPE, and further experimental results complying to the recurring reviewer feedback.

- **Comparison table:** We also include a comparison table with time complexity of the SOTA algorithms in preference-based pure exploration against FraPPE, which summarises our contribution in terms of computational efficiency achieved in this work.

- **Definition of $\tilde{\epsilon}$:** As pointed out by reviewer eLPG, we include a detailed discussion in Appendix E on the choice of $\tilde{\epsilon}$. The discussion is already written in the response.

Thanks again for your time and effort in reviewing and chairing our paper.

Regards,

Authors

---

### Decision · Program_Chairs · 2025-09-17

**Decision:**

Accept (poster)

**Comment:**

**Summary:**  This paper focused on tackling pure exploration with preference feedback, which identify the best or an $\epsilon$-optimal item using duels rather than numeric rewards, under standard stochastic preference models. To address this challenging problem, this paper proposed FraPPE, which (i) adaptively focuses comparisons on an active set of contenders, (ii) chooses informative pairs via confidence-interval geometry to accelerate elimination, and (iii) maintains computational efficiency with incremental updates and pair selection. Theoretical performance analysis of FraPPE was presented. The instance-dependent sample complexity scales with natural hardness measures of the preference matrix. Finally, experimental results were presented to validate the performance of FraPPE.

**Strength:**
- The problem itself is interesting and important as pure exploration with pairwise feedback is central to ranking, recommendation, and RLHF-style evaluation.
- The design of active set and the informative pair selection are elegant, simple to be implemented in practice.
- The performance of FraPPE is theoretically guaranteed with formal $\delta$-correctness and the instance-dependent complexity bounds. The analysis highlights the role of preference gaps.
- The performance of FraPPE is further validated via experiments including comparisons to baselines in several key metrics.

**Weakness:**
- The performance guarantees rely on standard stochastic preference assumptions. This paper could further discuss robustness when these are violated.
- While instance-dependent bounds are valuable, this paper could clarify the gap to known minimax lower bounds (tightness) and when FraPPE might be suboptimal in adversarially hard instances.

**Reasons for acceptance:** This paper demonstrated clear contributions from both theory and practice. The proposed FraPPE advances the state of the art for preference-based pure exploration with a design that is simultaneously fast, sample-efficient, and analyzed.

**Discussion \& rebuttal assessment:** Reviewers reached a consensus that this is a solid work with interesting problems, nice algorithm design, theoretical sound performance guarantees and experimental validations.